# DCcluster-Opt: Benchmarking Dynamic Multi-Objective Optimization for Geo-Distributed Data Center Workloads

**Antonio Guillen-Perez**[†], **Avisek Naug**[†], **Vineet Gundecha, Sahand Ghorbanpour,**
**Ricardo Luna Gutierrez**, **Ashwin Ramesh Babu**, **Munther Salim**, **Shubhanker Banerjee**,
**Eoin H. Oude Essink**, **Damien Fay**, **Soumyendu Sarkar**[†*]

Hewlett Packard Enterprise

```
{antonio.guillen, avisek.naug, vineet.gundecha, sahand.ghorbanpour,
   rluna, ashwin.ramesh-babu, msalim, shubhanker, eoin.oude-essink,
                 damien.fay, soumyendu.sarkar}@hpe.com
```

## Abstract

The increasing energy demands and carbon footprint of large-scale AI require intelligent workload management in globally distributed data centers. Yet progress is limited by the absence of benchmarks that realistically capture the interplay of time-varying environmental factors (grid carbon intensity, electricity prices, weather), detailed data center physics (CPUs, GPUs, memory, HVAC energy), and geo-distributed network dynamics (latency and transmission costs). To bridge this gap, we present DCcluster-Opt: an open-source, high-fidelity simulation benchmark for sustainable, geo-temporal task scheduling. DCcluster-Opt combines curated real-world datasets, including AI workload traces, grid carbon intensity, electricity markets, weather across 20 global regions, cloud transmission costs, and empirical network delay parameters with physics-informed models of data center operations, enabling rigorous and reproducible research in sustainable computing. It presents a challenging scheduling problem where a top-level coordinating agent must dynamically reassign or defer tasks that arrive with resource and service-level agreement requirements across a configurable cluster of data centers to optimize multiple objectives. The environment also models advanced components such as heat recovery. A modular reward system enables an explicit study of trade-offs among carbon emissions, energy costs, service level agreements, and water use. It provides a Gymnasium API with baseline controllers, including reinforcement learning and rule-based strategies, to support reproducible ML research and a fair comparison of diverse algorithms. By offering a realistic, configurable, and accessible testbed, DCcluster-Opt accelerates the development and validation of next-generation sustainable computing solutions for geo-distributed data centers.

## 1  Introduction

The rapid growth of large-scale Artificial Intelligence (AI) workloads, driven by the immense computational demands of foundation models, is forcing a paradigm shift towards vast, geo-distributed, and often hybrid computing ecosystems. Initiatives such as the U.S. Department of Energy's American Science Cloud (AmSC) exemplify this future, with the goal of bringing together the nation's leading

---

*Corresponding author. †These authors contributed equally.

39th Conference on Neural Information Processing Systems (NeurIPS 2025) Track on Datasets and Benchmarks.

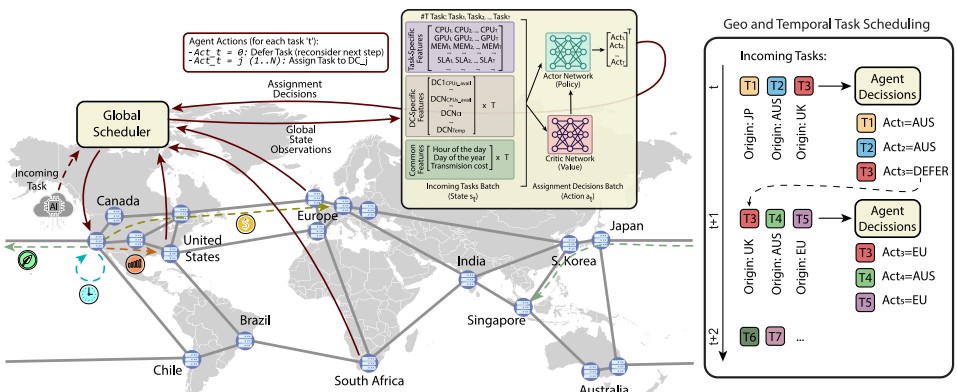

Figure 1: Overview of DCcluster-Opt. **Main Map:** A centralized Global Scheduler manages AI tasks across geo-distributed data centers, considering cost, carbon, network performance, and SLA/deferral. **Agent I/O (Top Center):** Illustrates state $s_t$ (common, DC-specific, task-specific features) input to an Actor-Critic model, producing per-task actions $a_t$. **Scheduling Example (Right):** Shows dynamic task arrivals (T1-T7) over timesteps $t, t+1, t+2$, with agent decisions (assign/defer) and changing task DC execution based on the task origin. DCcluster-Opt integrates real-world data, challenging agents with global and temporal optimization.

supercomputing, data, and experimentation resources into a cohesive scientific instrument [1]. Managing these complex, mission-critical systems presents an unprecedented challenge, demanding a new class of trustworthy, multi-objective AI controllers that can dynamically balance performance, cost, and crucial sustainability goals like carbon emissions and energy efficiency.

Achieving this balance in practice is a formidable operations research problem. These geo-distributed data centers (DCs) encounter heterogeneous computation and a complex interplay of time-varying conditions, fluctuating electricity prices, variable grid carbon intensity, differing network costs and latencies, and varying weather conditions that affect the efficiency of critical components like HVAC systems [2]. Schedulers for these next-generation systems must therefore be capable of managing intricate spatio-temporal trade-offs across this entire spectrum of dynamic signals.

However, the development, auditing, and validation of such controllers are critically hampered by the lack of realistic testbeds. Many existing research studies on task scheduling (see survey by [3]) fail to simultaneously model this full complexity, often employing simplifying assumptions like using abstract physical models [4] or optimizing local controls in isolation [5]. This creates a fundamental gap between algorithmic theory and operational reality, hindering the creation of schedulers that can be trusted in production. To bridge this gap, we present DCcluster-Opt: an open-source, high-fidelity simulation benchmark designed to serve as a configurable testbed where trustworthy AI controllers can be developed, audited, and de-risked before deployment.

To address this gap, we present **DCcluster-Opt**, an open-source benchmark for sustainable spatio-temporal AI task scheduling in globally distributed data centers (Figure 1). DCcluster-Opt combines real-world data streams with physics-informed DC models, offering a high-fidelity, reproducible testbed for developing and comparing intelligent schedulers that reduce AI's environmental impact.

Key contributions of this work include:
- **A high-fidelity configurable simulation benchmark environment for hierarchical real-time data center cluster control and optimization**, modeling compute loads, DC physics, network dynamics, and sustainability signals (cost, carbon, water) for evaluating geo-distributed scheduling along with optimizing control of DC components like cooling.
- **Integration of diverse, real-world cluster-relevant datasets,** including the Alibaba AI workload trace [6], electricity prices, and grid carbon intensity for over 20 global regions (from sources such as Electricity Maps [7] and GridStatus [8]), weather data (from Open-Meteo [9]), and cloud provider transmission costs.
- **A physics-informed extendable datacenter model with plug-in design** accounting for IT power (CPU, GPU, Memory), thermal dynamics, and supporting HVAC control (with EnergyPlus-based [10] and alternative models [11], building on works like [12, 13, 14, 15]), **and supporting optional advanced components like Heat Recovery Units (HRUs) and dynamic RL-based control of HVAC cooling setpoints.**
- **Transmission-aware routing** incorporating monetary costs, carbon emissions, and realistic network delays (based on empirical data from [16]).

- **A modular reward system** that allows for configurable single or multi-objective optimization, facilitating analysis of trade-offs for cost vs. energy vs. carbon vs. water vs. SLA vs. delay.
- **Open-source**[2] with a Gymnasium API [17], configuration files for reproducibility, and baseline controllers (RBCs and example RL agents, including integration with Ray RLlib [18]).

This paper begins with a comparison of related work in Section 2, followed by an overview of DCcluster-Opt's design (Section 3) and the datasets it integrates (Section 4). Section 5 then formally defines the problem as a Markov Decision Process (MDP) and details the environment's Gymnasium-compatible API. An evaluation protocol with extensive results is presented in Section 6, followed by the introduction of our novel agentic AI controller in Section 7. The paper concludes in Section 8 with a discussion of limitations, future research directions, and the benchmark's broader impact. Further details are provided in the supplementary document.

## 2   Related Work

Optimizing DC operations across geo-distributed systems for sustainability and operational efficiency is a diverse research area. DCcluster-Opt contributes a novel and integrated benchmark environment to this evolving landscape.

Early geo-distributed scheduling research often prioritized performance metrics like makespan or data locality [19], sometimes with simplified network or cost models. More recent works incorporate economic factors, such as game theory for cost minimization with time-of-use electricity pricing [20]. Significant efforts focus on sustainable computing via energy- and carbon-aware scheduling, including "follow the renewables/carbon" strategies [21] and applications to workloads with real-time data [22]. Others explore DC-smart grid coordination [23, 24, 25]. While valuable, these often focus on specific optimization facets or use abstracted physical models (e.g., fixed PUE). DCcluster-Opt offers a unified platform to evaluate such strategies under consistent, realistic conditions, using real-world dynamic data and a full-stack energy model (IT, cooling, transmission).

Reinforcement Learning (RL) is increasingly applied to DC management (see surveys [26, 27, 3]), with applications in resource allocation [4], local HVAC control [5, 28, 29], and MARL for task scheduling [22]. DCcluster-Opt serves as a standardized benchmark for these RL approaches, particularly for the complex, global, multi-objective scheduling task. It presents unique RL challenges from dynamic real-world data, physical system inertia, network delays, and multi-objective trade-offs, supported by a Gymnasium API and Ray RLlib integration [18].

Foundational to research in datacenter management are large-scale, public workload traces. Prominent examples include the MIT Supercloud Dataset [30], traces from Google's production clusters [31], and the Azure Public Dataset repository [32]. While these invaluable resources provide the critical *what* (the workload), DCcluster-Opt provides the comprehensive *where* and *when* the dynamic, multi-faceted context in which these workloads are executed. The core contribution of DCcluster-Opt, therefore, is not the provision of a new standalone trace, but the creation of an integrated, high-fidelity testbed that synthesizes such traces with: (i) dynamic, real-world geo-temporal data (carbon intensity, electricity price, weather); (ii) physics-informed energy models for both IT and, critically, HVAC systems; and (iii) realistic network cost and latency models. This enables, for the first time, the holistic study and benchmarking of multi-objective optimization strategies for geo-distributed scheduling under realistic, dynamic conditions.

Existing cloud simulators like CloudSim [33] require extensive customization for detailed, dynamic sustainability studies. Recent sustainability benchmarks include SustainDC for holistic local DC component control [34] and Vessim [35, 36] for general carbon-aware systems co-simulation. DCcluster-Opt is distinct in its end-to-end focus on the global AI/GPU workload scheduling problem across geo-distributed DCs. Its novelty lies in the comprehensive *combination* of: (i) diverse, real-world geo-temporal data (Alibaba GPU trace [6], price, carbon, weather); (ii) physics-informed DC energy models (IT and HVAC); (iii) realistic network cost and delay models [16]; and (iv) a modular multi-objective reward system, packaged for RL research. This integrated approach provides a unique benchmarking capability.

---

[2]Code:      https://github.com/HewlettPackard/sustain-cluster;      Docs:      https://hewlettpackard.github.io/sustain-cluster/

# 3 Benchmark Design

DCcluster-Opt simulates the decision-making environment for a centralized global scheduler that manages AI workloads in a geographically distributed cluster of DCs to optimize sustainability and operational efficiency. This section outlines the benchmark's core concepts and key modeling pillars.

## 3.1 Core Concept: Geo-Distributed Scheduling Scenario

The benchmark models a scenario where a **centralized scheduling agent** manages incoming AI tasks across a **cluster of $N$ geographically distributed data centers**. These simulated sites can represent diverse real-world configurations, such as multiple facilities of a single large organization, a consortium sharing resources (e.g., national labs, federated clouds), or edge nodes interacting with regional clouds. Crucially, each DC is situated in a unique location, subject to distinct, time-varying environmental data (grid carbon intensity, electricity price, weather) and network characteristics (transmission costs, delays [16, 37]). The agent must dynamically assign or defer tasks to optimize multi-objective goals (e.g., global carbon, operational cost, SLAs) under DC resource capacities (CPU, GPU, Memory) and network transfer constraints. See Figure 1 for a visual explanation.

## 3.2 Simulation Loop

DCcluster-Opt progresses in discrete 15-minute timesteps. At each step $t$, the agent interacts with the environment (Figure 2) via the following cycle:

1. **Task Input:** New tasks (with assigned origins explained in Suppl. Mat. Sec. F.5) and previously deferred tasks are created.
2. **Observation ($s_t$):** Agent receives global state and details of $k_t$ pending tasks.
3. **Action ($a_t$):** Agent outputs $k_t$ decisions (defer or assign task $i$ to DC $j$).
4. **Routing:** Remote assignments incur transmission penalties (monetary cost, energy consumption, carbon emissions, and network delay), as detailed in Suppl. G; tasks enter an "in-transit" state for the duration of the calculated delay.
5. **DC Simulation:** DCs process queues, simulate execution, update IT/HVAC power, and log.
6. **Reward ($r_t$):** Global reward calculated based on outcomes via the configured reward function.
7. **State Transition:** Clock advances 15 mins; next state $s_{t+1}$ is generated.

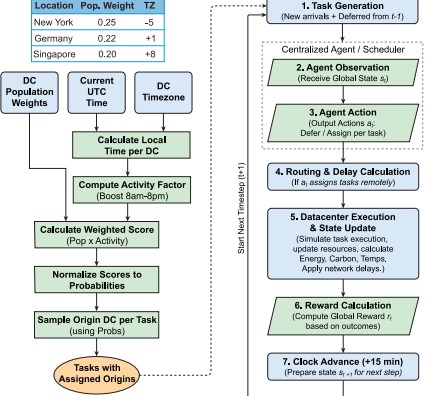

Figure 2: DCcluster-Opt Task Origin Logic (Left) & Simulation Loop (Right). Core 15-min loop: (1) Task Input, (2) Observe $s_t$, (3) Act $a_t$, (4) Route, (5) DC Sim, (6) Reward $r_t$, (7) Transition to $s_{t+1}$.

The 15-minute timestep aligns with real-world data availability, operational cadences, and DC thermal inertia (detailed justification in Suppl. Mat. Sec. C).

## 3.3 Key Modeling Pillars for Realism

DCcluster-Opt integrates several key modeling components for realism:

- **Physics-Informed Datacenter Model:** Each DC simulates IT power (CPU, GPU, Memory varying with load/temperature; server fans) and HVAC system energy. The HVAC model includes components like CRAC units, chillers (using EnergyPlus-derived performance curves [10]), and cooling towers, whose performance adapts to IT load and ambient weather. This captures significant cooling overheads. The framework also supports active control of HVAC cooling setpoints and can include Heat Recovery Units (HRUs), which reuse waste heat to reduce net cooling demand. (Details are in Suppl. Mat. Sec. B, based on [14, 15]).
- **Transmission-Aware Network Model:** Routing tasks remotely incurs: (i) *Monetary Cost* ($/GB from cloud provider rates); (ii) *Energy Consumption* (kWh/GB [38]); (iii) *Carbon Emissions* (Tx

Energy $\times$ Origin Grid CI); and (iv) *Latency (Delay)* (from empirical throughput/RTT data [16]). (Calculation details: Suppl. Mat. Sec. E).
- **Dynamic Task & Environment Model:** Utilizes (i) a real-world AI workload trace [6] with varied demands and realistic probability origin patterns; (ii) SLA constraints; and (iii) real-world, time-varying data streams for electricity prices, grid carbon intensity, and weather.

These models make scheduling decisions that impact real sustainability and performance outcomes.

## 4 Datasets

DCcluster-Opt's realism is critically underpinned by its integration of diverse, real-world datasets that drive the simulation's environmental conditions, workload characteristics, and economic factors. An overview of these datasets, their sources, and their roles is presented in Table 1. Each dataset component has undergone preprocessing and standardization to ensure compatibility and usability within the benchmark environment.

Table 1: Overview of Datasets Integrated into DCcluster-Opt.

| Dataset Component | Source(s) | Description | Coverage |
|---|---|---|---|
| AI Workloads | Alibaba Cluster Trace 2020 [6] | AI job trace (training/inference) | Original: 2 months (2020), >6.5k GPUs. Extended to 1 year. |
| Electricity Prices | Electricity Maps [7], GridStatus [8], ISOs | Time-series electricity prices | >20 global regions, 2020-2024, Sub-hourly. |
| Carbon Intensity | Electricity Maps [7] | Time-series grid carbon intensity | >20 global regions, 2021-2024, Sub-hourly. |
| Weather | Open-Meteo [9] | Ambient air temperature | Locations matching DCs, 2021-2024, Sub-hourly. |
| Transmission Costs | Cloud Providers[a] | Inter-region data transfer cost ($/GB) | Major cloud provider regions. |
| Transmission Delay | Persico et al. [16] | Empirical Throughput and RTT | Between 4 macro-regions (EU, US, AP, SA). |

[a]Based on public pricing from AWS (`aws.amazon.com/ec2/pricing/on-demand/`), GCP (`cloud.google.com/vpc/pricing`), Azure (`azure.microsoft.com/en-us/pricing/details/bandwidth/`).

- **AI Workloads:** The experiments in this paper are driven by the public Alibaba Cluster Trace 2020 [6], a large-scale and detailed GPU job trace. However, the DCcluster-Opt framework is designed to be **workload-agnostic** to ensure long-term relevance. Its data pipeline can be readily adapted to ingest other modern, public traces that specify resource requirements and duration. This allows researchers to model evolving workload patterns, including those from LLMs, by using contemporary datasets such as the MIT Supercloud Dataset [30] or recent traces like Azure [32]. (Details on the Alibaba trace processing are in Suppl. Mat. Sec. D.2).
- **Electricity Prices:** Time-varying prices ($/kWh) for each location are integrated, based on real-world data from sources like [7, 8] and regional operators, enabling simulation of dynamic operational costs. The benchmark's modular design also allows for the implementation of custom, **reactive pricing models**, where the electricity price can respond dynamically to changes in datacenter energy demand. (Details on the current data are in Suppl. Mat. Sec. D.3)
- **Carbon Intensity:** Real-time grid carbon intensity ($gCO_2eq/kWh$) from [7] allows calculation of the location-specific carbon footprint for compute, cooling, and transmission energy, providing a crucial signal for carbon-aware scheduling. (Details: Suppl. Mat. Sec. D.4)
- **Weather Data:** Location-specific ambient temperature data from [9]. It influences the simulated efficiency and energy consumption of the HVAC model. (Details: Suppl. Mat. Sec. D.5)
- **Transmission Costs:** Inter-datacenter data transfer costs ($/GB) are modeled using rates derived from public cloud provider pricing, contributing to operational expense calculations. (Details: Suppl. Mat. Sec. D.6)
- **Transmission Delay Parameters:** Network latency is modeled via delay calculations using empirical throughput and RTT data [16] between geographical macro-regions, impacting task arrival times at remote sites. (Details: Suppl. Mat. Sec. D.7)

Detailed datasheets for each of these core dataset components, following the framework by Gebru et al. [39], are provided in Suppl. Mat. Sec. D.10. These datasheets offer comprehensive information on motivation, composition, collection, preprocessing, uses, distribution, and maintenance. Further specifics on file formats, visualizations, our maintenance plan, and Croissant metadata [40] are also consolidated in Suppl. Mat. Sec. D and Suppl. Mat. Sec. K.

## 5 Problem Formulation and Environment API

We formalize the scheduling problem as a discrete-time Markov Decision Process (MDP). The agent interacts with this MDP via DCcluster-Opt's Gymnasium-compatible API (`TaskSchedulingEnv`). At each 15-minute timestep, the agent observes the state $s_t$, takes an action $a_t$, and receives a reward $r_t$. The simulator's complex transition dynamics then evolve the environment. For a formal, at-a-glance reference, the MDP components are defined in Table 2.

The API exposes these components as follows. The **state** $s_t$, returned by 'env.step()', is a variable-length list of $k_t$ NumPy arrays, one for each pending task. Each per-task feature vector concatenates global time features, task-specific requirements (CPU, GPU, Memory, SLA), and the current state of all N datacenters (resource availability, carbon intensity, electricity price). The **action** $a_t$, passed to 'env.step()', is a corresponding list of $k_t$ integer decisions from $\{0..N\}$, where $a_i = 0$ corresponds to **deferring** a task and $a_i = j > 0$ **assigns** it to datacenter $j$. Finally, the scalar **reward** $r_t$ is computed by a configurable `RewardFunction` that creates a weighted sum of KPIs like total energy cost, carbon emissions, and SLA violations, enabling explicit multi-objective optimization. Full implementation details for handling variable-length inputs and configuring the reward system are provided in the Supplementary Material (Sec. F.2.3, F.4).

Table 2: Formal Definition of the DCcluster-Opt Scheduling MDP.

| Component | Formal Definition |
|---|---|
| **State ($s_t$)** | A list of $k_t$ feature vectors, where each vector concatenates: (1) Global Time Features (4 dims), (2) Task-Specific Features (5 dims), and (3) Per-DC State Features (5×N dims for N datacenters). |
| **Action ($a_t$)** | A sequence of $k_t$ discrete decisions $\{a_{t,i}\}$, where each action $a_{t,i} \in \{0, 1, ..., N\}$ corresponds to either **Defer** ($a_{t,i} = 0$) or **Assign** to datacenter $j$ ($a_{t,i} = j$). |
| **Transition ($P$)** | Simulator dynamics mapping $(s_t, a_t)$ to $s_{t+1}$, including new task arrivals, network routing delays, and physics-informed updates to all DC states. |
| **Reward ($r_t$)** | A scalar value from a configurable, multi-objective function, $r_t = \sum_c w_c \cdot f_c(\cdot)$, where $w_c$ is a weight and $f_c$ is a function for a Key Performance Indicator (KPI) such as total energy cost, carbon emissions, or SLA violations. |

The benchmark's behavior is controlled via YAML files, allowing users to define the cluster topology (`datacenters.yaml`), simulation scenario (`sim_config.yaml`), and multi-objective reward function (`reward_config.yaml`). Comprehensive explanations are in Suppl. Mat. Sec. F.6.

## 6 Experimental Evaluation

To demonstrate the utility of DCcluster-Opt for evaluating and comparing sustainable scheduling strategies, this section outlines our experimental setup, defines the baseline methods compared, presents key performance metrics, and discusses the empirical results.

### 6.1 Experimental Setup and Metrics

**Simulation Configuration.** All evaluations used a simulated 5-DC cluster over a **30-day** period with a 15-minute timestep. To account for stochasticity, each controller was run with **10 random seeds**, and results are reported as **mean ± std. dev**. All RL agents were trained using the same multi-objective reward function. Full configuration details, including DC specifications and workload files, are provided in the Supplementary Material (Sec. G.1.2 and G.1.1).

**Evaluation Metrics.** We assess scheduling strategies using Key Performance Indicators (KPIs) aggregated over the simulation period, encompassing sustainability, economic, and operational aspects. Key metrics include:

- **Sustainability:** Total CO2 Emissions (tonnes, from compute, cooling, and transmission), Total Energy Consumption (MWh), and Total Water Usage (m³).
- **Economic:** Total Operational Cost ($, sum of electricity and transmission costs).
- **Operational Performance:** SLA Violation Rate (%), Average CPU & GPU Utilization (%), Total Transmission (TX) Cost ($), and Total Tasks Deferred.

Lower values are generally better, except for resource utilization, where balanced, high usage is often preferred. These KPIs allow for a nuanced comparison of scheduling policies.

## 6.2 Baseline Controllers and Agents

We compare several scheduling approaches:

- **Rule-Based Controllers (RBCs):** Simple, non-learning heuristics: *Local Only* (no geo-shifting), *Lowest Carbon*, *Lowest Price*, *Most Available* (resource-based), and *Round Robin* [41]. These are invoked via the `strategy` parameter in `sim_config.yaml`. (Detailed logic in Suppl. Mat. Sec. G.2).
- **Custom SAC Implementation:** A Soft Actor-Critic (SAC) [42] agent trained with our specific implementation, capable of both geographical task placement and temporal deferral (*SAC Geo+Time*), and its ablations: *SAC Geo Only* (no deferral) and *SAC Time Only* (local execution with deferral).
- **RLlib Agents:** Agents trained using Ray RLlib [18] to showcase broader compatibility: **PPO** [43], **APPO** [44], and **IMPALA** [45], all with Geo+Time capabilities. These results are included in the Supplemental G.5.
- **Advanced HVAC Control Scenarios:** The *SAC Geo+Time* scheduler is also evaluated with: (1) local RL-controlled HVAC (PPO agent dynamically adjusting cooling setpoints), and (2) RL-controlled HVAC combined with a simulated Heat Recovery Unit (HRU).

Training details and hyperparameters for all RL agents (global schedulers and local HVAC controller) are provided in Suppl. Mat. Sec. G.3.1 and Suppl. Mat. Sec. I.

## 6.3 Results and Analysis

**Comparison of Global Schedulers.** Table 3 presents the aggregated performance for RBCs and our custom SAC agent variants. The results demonstrate DCcluster-Opt's ability to highlight multi-objective trade-offs. While RBCs like `Lowest Price` and `Lowest Carbon` excel in their target metric, they often compromise elsewhere (e.g., `Lowest Price` incurs high CO2). `Round Robin` achieves excellent SLA compliance at moderate costs. Our `SAC (Geo+Time)` agent learns a policy that balances objectives, achieving lower total cost ($92401) than most RBCs and competitive CO2 emissions (308.7 t), but with a higher SLA violation rate (25.47%) due to its use of deferral (474 tasks). Ablations

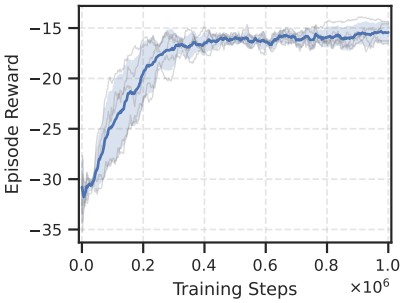

Figure 3: Training progression of SAC (Geo+Time). Mean smoothed episode reward (solid) and ±1 std dev (shaded) over 5 runs. Faded indv. runs. Consistent learning over 1M steps. Reward details in Suppl. Mat. Sec. G.3.2

show that geographical shifting (`SAC Geo Only`) minimizes transmission costs ($4131) and also yields zero SLA violations, while temporal shifting (`SAC Time Only`) operates similarly to `Local Only` but with significant deferral and SLA impact. These findings underscore the challenge of balancing sustainability with operational performance.

To demonstrate the trainability and stability of RL agents within DCcluster-Opt, Figure 3 illustrates the training progression for the `SAC (Geo+Time)` agent, showing consistent improvement in optimizing the composite reward over 1 million steps across five random seeds.

Table 13 compares agents trained using Ray RLlib [18]. The PPO with its current default hyperparameters, generally incurred higher costs and emissions, and exhibited significantly higher SLA violation

Table 3: Performance of scheduling strategies (mean ± std. dev. over 10 seeds).

| Controller | Total Cost ($) ↓ | Total CO2 (t) ↓ | Total Energy (MWh) ↓ | Total Water (m³) ↓ | SLA Viol. (%) ↓ | Avg CPU Util (%) | Avg GPU Util (%) | TX Cost ($) ↓ | Tasks Deferred |
|---|---|---|---|---|---|---|---|---|---|
| RBC (Local Only) | 101349±5162 | 328±7.2 | 1087±2.0 | 7394±64 | 0.02±0.03 | 3.4±0.0 | 11.4±0.2 | 0±0 | 0±0 |
| RBC (Lowest Carbon) | 99845±4765 | 311±7.6 | 1122±8.6 | 7336±70 | 1.32±0.32 | 3.2±0.3 | 12.6±0.8 | 3273±200 | 0±0 |
| RBC (Lowest Price) | 94119±3777 | 325±7.4 | 1097±7.3 | 7355±58 | 1.15±0.54 | 3.8±0.2 | 11.9±0.7 | 3572±213 | 0±0 |
| RBC (Most Available) | 107067±3770 | 325±6.6 | 1165±0.6 | 7373±62 | 1.70±0.00 | 2.4±0.0 | 15.6±0.0 | 2927±38 | 0±0 |
| RBC (Round Robin) | 100896±4450 | 329±7.2 | 1087±1.6 | 7427±62 | **0.00±0.00** | 3.5±0.0 | 11.9±0.0 | 3580±48 | 0±0 |
| RL (Geo Only) | 92618±4139 | **308±7.4** | 1037±2.2 | 7247±67 | 25.42±0.28 | 4.7±0.0 | 6.6±0.3 | 4131±38 | 0±0 |
| RL (Time Only) | 101346±5185 | 327±7.2 | 1086±2.0 | 7394±64 | 23.12±0.05 | 3.4±0.0 | 11.4±0.2 | 0±0 | 432±171 |
| RL (Geo+Time) | **92401±4134** | 308±7.6 | 1037±2.1 | **7253±66** | 25.47±0.30 | 5.0±0.0 | 6.5±0.3 | 4203±37 | 474±165 |

*RL agents trained with default multi-objective reward. Lower is better for all metrics except utilization (%). Bold indicates best performance per column. **Geo Only**: RL agent selects data center location but cannot defer tasks. **Time Only**: Tasks can be deferred but are executed locally. **Geo+Time**: Full agent with both deferral and geographical scheduling capabilities. SAC algorithm was used in this table, and for other RL algorithms trained with RLLib, see Supplemental G.5. TX = Transmission.

Table 4: Comparison of SAC (Geo+Time) Scheduler with Fixed vs. RL-Controlled HVAC (Mean ± Std Dev across 10 seeds over 30 days).

| Controller Configuration | Total Cost ($) | Total CO2 (t) | Total Energy (MWh) | Total Water (m³) |
|---|---|---|---|---|
| SAC (Geo+Time) with Fixed HVAC[†] | 92401±4134 | 308.7±7.6 | 1037.9±2.1 | 7253±66 |
| SAC (Geo+Time) with RL-Controlled HVAC | 81300±3616 | 273.1±6.9 | 921.3±2.3 | 7493±69 |
| SAC (Geo+Time) with RL-Controlled HVAC + HRU | 80225±3666 | 268.9±7.1 | 907.8±3.8 | 7093±69 |

[†]Data from Table 3 (SAC Geo+Time with default fixed HVAC setpoints). Both configurations use the same global task scheduling policy. Lower is better for all metrics.

rates (29.6%) and task deferrals (>3500) than our custom SAC agent. Their resource utilization was also lower. This indicates that while DCcluster-Opt is compatible with standard RL libraries, achieving optimal performance in this complex multi-objective environment requires careful tuning of both algorithmic hyperparameters and the reward structure. The benchmark's capacity to differentiate these advanced algorithms is thus evident. For other results with RL algorithms trained with RLLib, see Supplemental G.5.

**Impact of Advanced Local DC Control.** Table 4 details experiments on holistic system control. Introducing an RL-based HVAC controller (PPO agent dynamically adjusting cooling setpoints, Suppl. Mat. Sec. I) with the `SAC (Geo+Time)` scheduler reduced total energy by 11.2% (to 921.3 MWh) and CO2 by 11.5% (to 273.1 t) over fixed HVAC, with a 12.0% cost saving. Simulating an additional Heat Recovery Unit (HRU) (model in Suppl. Mat. Sec. B.5) further lowered energy to 907.8 MWh and CO2 to 268.9 t, also reducing water use. These results show DCcluster-Opt's utility for quantifying the benefits of hierarchical control and energy efficiency technologies.

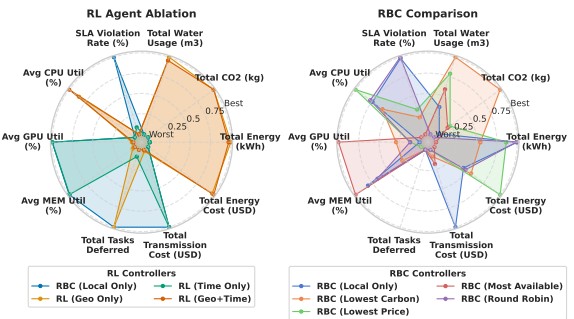

Figure 4: Normalized performance comparison using spider charts (higher/outer values are better). Metrics scaled 0 (worst) to 1 (best) across shown controllers. Left: SAC agent ablations. Right: RBCs. These visualize multi-objective trade-offs. Additional comparisons in Suppl. Mat. Sec. H.

**Visualizing Trade-offs and Dynamics.** To visualize multi-dimensional trade-offs, Figure 4 presents spider charts comparing normalized performance. These plots illustrate how different strategies prioritize objectives. Further time-series visualizations (e.g., Figures in Suppl. Mat. Sec. H) showcase the dynamic behavior of agents in response to changing environmental signals and workload.

## 7 An Agentic AI Controller for Optimized Operations

A core requirement for controllers managing critical, federated infrastructure like the AmSC is trustworthiness: the ability to be audited, understood, and adapted by human operators. Instead of a single "black box" algorithm, we have built an agentic system that addresses this challenge by mimicking the collaborative reasoning of an expert human operations team. Our system is composed of specialized AI agents that work together to:

1. **Sense:** Interpret the complex state of the system.
2. **Analyze:** Formulate high-level strategies to meet multi-objective optimization goals.

3. **Plan:** Translate strategy into a concrete, executable action plan.
4. **Validate:** Ensure the plan is safe and compliant before execution.
5. **Act & Monitor:** Interact with the simulated world and observe outcomes for continuous adaptation.

The result is a scheduler that is not only effective at multi-objective optimization but also transparent and auditable, allowing us to literally "watch it think."

## 7.1 A Multi-Agent Framework for Auditable Control

Our agentic controller is implemented as a team of six specialized LLM-based agents, orchestrated using the LangGraph framework. Each agent has a distinct role in a cognitive workflow that moves from high-level strategy to low-level execution, as depicted in Figure 5.

The agents in the framework are:

- **Sensor Agent:** Translates raw numerical state from the DCcluster-Opt environment into a semantically enriched JSON object, making the system's perception explicit.
- **Analyst Agent:** Receives the structured state and feedback from the previous cycle to formulate a high-level strategic directive. For example: "The global objective is to balance operational cost and carbon footprint. Current state shows DC3 has low carbon and favorable temperatures, making it a prime candidate for workload consolidation."
- **Planner Agent:** Converts the strategic directive into a specific, low-level action list (assign or defer) for every pending task.
- **Validator Agent (Guardrail):** A critical safety component that inspects the action plan for correctness (e.g., valid datacenter IDs, correct format) and compliance with operational rules before execution.
- **Executor Agent:** A simple wrapper that submits the validated plan to DCcluster-Opt environment.
- **Monitor Agent:** Pushes numerical metrics to Prometheus and uses an LLM to generate a qualitative feedback summary for the Analyst Agent's next cycle, enabling reflection and adaptation.

While traditional deep RL agents often act as opaque 'black boxes', the emergence of Large Language Models (LLMs) offers a path toward more interpretable control planes. To demonstrate DCcluster-Opt's utility for exploring these next-generation controllers, we present a case study on distilling the policy of our expert SAC agent into a flexible, text-based Large Language Model (LLM) controller.

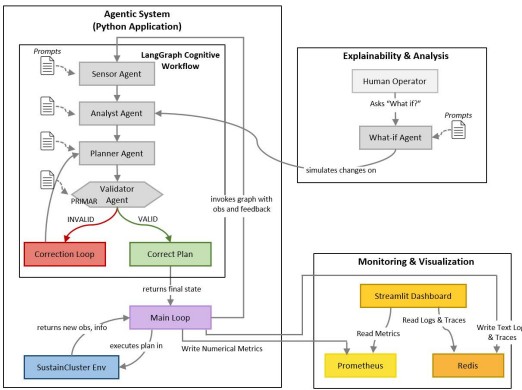

## 7.2 Methodology: Distilling the Centralized RL Policy into an LLM

Our approach is a multi-stage process designed to transfer the optimized scheduling strategy learned by our centralized SAC agent into an LLM. The workflow, illustrated in Figure 6, consists of generating an expert dataset from the trained RL policy, textualizing this data into

Figure 5: The proposed agentic AI framework for optimized scheduling. The system uses a LangGraph Cognitive Workflow where Sensor, Analyst, Planner, and Validator agents collaborate to form a correct plan. The framework also includes a "What-if" agent for operator-driven analysis.

structured prompts, and then fine-tuning a base LLM to create the final, scalable controller.

## 7.3 Explainability of LLM Decisions

**LLM-Generated Decision & Explanation**

**Decision**: DC1 routes to DC3, DC2 routes to DC3, DC3 compute on local.
**SHORT EXPLANATION**: Prioritizing DC3 due to its lower carbon intensity level, which reduces the global environmental impact.
**LONG EXPLANATION**: The primary goal of this decision is to minimize the carbon intensity level of the system... In this case, DC3 has a lower carbon intensity level (112.0 gCO2/kWh) compared to DC1 (194.0 gCO2/kWh) and DC2 (120.0 gCO2/kWh). By routing tasks to DC3, we can reduce the overall carbon intensity... Additionally, DC3 has a lower external temperature (9.3 C) compared to DC1 (25.8 C)... By prioritizing DC3, we can also take advantage of its available CPU and GPU resources (40% and 67%)... This decision is made by considering the trade-offs between the different data centers and their characteristics...

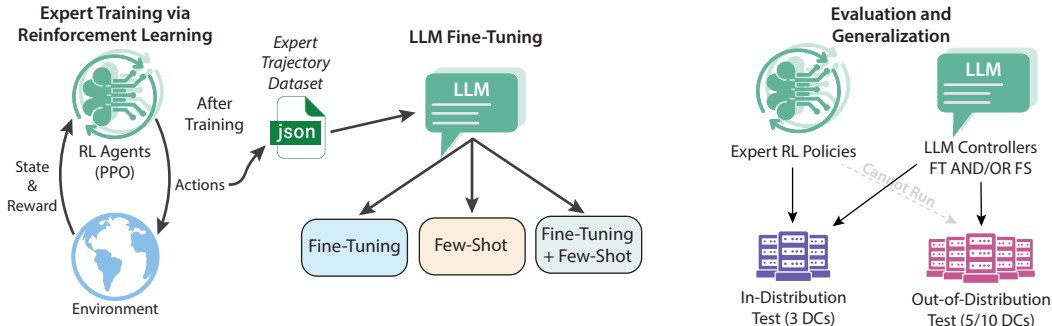

Figure 6: The proposed distillation framework for a centralized scheduler. (1) An expert RL policy ($\pi_{RL}$) is trained using the DCcluster-Opt environment. (2) The expert policy is executed to generate a dataset of state-action trajectories. (3) The numerical state and action data are textualized into a structured, human-readable prompt format. (4) A base LLM is then fine-tuned on this textualized data to create the final, generalizable reasoning controller.

The LLM shows strong multi-objective reasoning, prioritizing carbon intensity while citing input CI values and weighing secondary factors like temperature and resource availability. Its ability to generate auditable, human-readable decision logs marks a crucial step toward building the trustworthy autonomous systems required for production environments. This approach offers three key advantages over traditional RL: **1) Explainability and Trustworthiness,** as the controller can justify any decision with data, acting as a copilot for human operators; **2) Adaptability,** as objectives and constraints can be changed on-the-fly by modifying natural language prompts, allowing it to react instantly to unforeseen events; and **3) Scalability,** as the text-based reasoning generalizes to different numbers of datacenters without requiring costly retraining.

Table 5: Performance comparison. Left (a): In-distribution results on the 3 Geo-Distributed Datacenters. Right (b): Out-of-distribution generalization results on larger, unseen clusters (5/10 Geo-Distributed Datacenters). Values averaged every timestep.

(a) 3-Datacenter (In-Distribution)

| Controller | Carbon (t) ↓ | Cost ($) ↓ | SLA (%) ↓ | Delay (min) ↓ |
|---|---|---|---|---|
| RBC (Lowest Carbon) | 24.63 | 10,131 | 1.33 | 112 |
| RL Expert (SAC) | 22.91 | 8,974 | 1.56 | 1,095 |
| **Our LLM** | **22.51** | **8,817** | **1.15** | 355 |

(b) Generalization (Out-of-Distribution)

| Scenario | Controller | Carbon (t) ↓ | SLA (%) ↓ |
|---|---|---|---|
| 5-DCs | RBC | 62.76 | 1.48 |
| | LLM | **59.55** | **1.08** |
| 10-DCs | RBC | 131.81 | 1.95 |
| | LLM | **125.10** | **1.19** |

# 8 Discussion and Conclusion

We introduced **DCcluster-Opt**, an open-source, high-fidelity benchmark to address the critical need for robust tools in geo-distributed workload optimization. By combining real-world datasets with physics-informed models and a standard Gymnasium API, DCcluster-Opt provides a comprehensive platform for developing and evaluating advanced, multi-objective scheduling algorithms. Our empirical evaluations, comparing various rule-based and reinforcement learning controllers, demonstrate the benchmark's efficacy in capturing complex spatio-temporal trade-offs and differentiating the performance of diverse strategies.

The benchmark's realism is grounded in a **component-wise validation** strategy, where each module is tied to empirical data or established models. However, we acknowledge key limitations that also define our future roadmap. A primary consideration is the evolution of AI workloads; the benchmark's **trace-agnostic** design is a core feature to support modern LLM traces as they become available [30, 32]. Further, while our models are physics-informed, future work can extend them to capture more complex dynamics like Locational Marginal Pricing (LMP), a research avenue our modular architecture is explicitly designed to support.

By offering this realistic, configurable, and extensible testbed, DCcluster-Opt serves as a foundational tool to develop and audit the trusted, autonomous control planes required for next-generation scientific infrastructure like the American Science Cloud. We believe it will accelerate the transition of these advanced, intelligent controllers from simulation to production, ultimately helping to optimize the operational cost and environmental footprint of global computing.

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

# Appendix

## Contents

# A Installation and Dependencies

This section provides detailed instructions for setting up the DCcluster-Opt environment. The code is available at https://github.com/HewlettPackard/sustain-cluster and the documentation can be found at https://hewlettpackard.github.io/sustain-cluster/.

## A.1 System Requirements

The implementation is compatible with various operating systems. All code and dependency installations were tested on macOS 15.4.1 and Ubuntu 22.04. Windows is also supported. The following prerequisites are necessary:

- Python 3.10 or higher
- Git version control system
- Command-line interface: Unix-compatible shell (bash, zsh) or PowerShell on Windows

## A.2 Installation Procedure

### A.2.1 Repository Acquisition

The codebase must be obtained via the following commands:

```
git clone https://github.com/HewlettPackard/sustain-cluster.git
cd sustain-cluster
```

### A.2.2 Virtual Environment Configuration

For Unix-based systems (Linux/macOS):

```
python3 -m venv DCcluster-Opt
source DCcluster-Opt/bin/activate
```

For Windows systems (using PowerShell):

```
python -m venv DCcluster-Opt
.\DCcluster-Opt\Scripts\Activate.ps1
```

Table 6: Python library dependencies for implementation, documentation, and testing

| Category | Package | Version |
|---|---|---|
| | torch | 2.6.0 |
| | pandas | 2.2.3 |
| | matplotlib | 3.10.1 |
| | gymnasium | 1.1.1 |
| Implementation | tqdm | 4.67.1 |
| | tensorboard | 2.19.0 |
| | seaborn | 0.13.2 |
| | PyYAML | 6.0.2 |
| | psychrolib | 2.5.0 |
| | sphinx | 8.2.3 |
| Documentation | furo | 2024.8.6 |
| | sphinx-autodoc-typehints | 3.1.0 |
| Testing | pytest | 8.3.5 |

### A.2.3 Dependency Installation

The libraries delineated in Table 6 correspond to the entries in the `requirements.txt` file and may be installed as follows:

```
pip install --upgrade pip
pip install -r requirements.txt
```

## A.3  Dataset Preparation

The implementation utilizes the Alibaba production cluster dataset 2020[3]. Researchers should download the dataset from the repository and place the processed file in the following default location:

```
data/workload/alibaba_2020_dataset/result_df_full_year_2020.pkl
```

The code supports both the preprocessed `.pkl` file format and the original `.zip` archive. If the `.pkl` file is not detected in the specified location, the system will automatically extract the required data from the corresponding `.zip` archive, assuming it is present in the same directory. For alternative configurations, the dataset may be stored in any accessible directory location, provided that the corresponding path is appropriately specified in the configuration file:

```
configs/env/sim_config.yaml
```

The path should be modified by updating the `workload_path` key in the YAML configuration file. All specified paths should be relative to the project root directory (sustain-cluster).

## A.4  Training

To initiate training, first activate the virtual environment and then execute the training script:

For Unix-based systems (Linux/macOS):

```
source DCcluster-Opt/bin/activate
python train_rl_agent.py --sim-config configs/env/sim_config.yaml \
                         --reward-config configs/env/reward_config.yaml \
                         --dc-config configs/env/datacenters.yaml \
                         --algo-config configs/env/algorithm_config.yaml \
                         --seed 42 \
                         --enable-logger True
```

For Windows (PowerShell):

```
.\DCcluster-Opt\Scripts\Activate.ps1
python train_rl_agent.py --sim-config configs/env/sim_config.yaml \
                         --reward-config configs/env/reward_config.yaml \
                         --dc-config configs/env/datacenters.yaml \
                         --algo-config configs/env/algorithm_config.yaml \
                         --seed 42 \
                         --enable-logger True
```

The training script utilizes the following default configuration files:

- `configs/env/datacenters.yaml`: Defines the data center specifications, including geographical locations, computing resources, and timezone information.
- `configs/env/sim_config.yaml`: Contains simulation parameters such as temporal settings, workload paths, and execution strategies.
- `configs/env/reward_config.yaml`: Configures the reward function components and their respective weights for RL.
- `configs/env/algorithm_config.yaml`: Specifies RL hyperparameters, including learning rates, batch sizes, and neural network configurations.

The architecture supports training with the Ray RLlib framework [18], enabling systematic implementation of RL algorithms across various experimental conditions. This integration provides access

---

[3]Available at: `https://github.com/alibaba/clusterdata`

to a standardized interface for multiple established algorithms, facilitating reproducible research and reducing development time through the utilization of validated policy optimization methods.

## A.5 Evaluation Methodology

To evaluate a trained model's performance, first modify the evaluation script to specify the location of the checkpoint:

```
# Edit eval_agent_notebook.py to update the checkpoint_path variable
# Replace with your specific training timestamp
checkpoint_path = "checkpoints/train_<training_timestamp>/best_checkpoint.pth"
```

Then run the evaluation script:

```
source DCcluster-Opt/bin/activate  # For Unix-based systems
python eval_agent_notebook.py
```

The evaluation procedure generates quantitative metrics such as energy related costs, service-level agreement statistics and hardware resource utilization.

# B  Datacenter Model Details

This section elaborates on the physics-informed models used within each simulated data center (the `SustainDC` environment and its underlying `DatacenterModel`), as introduced in Section 3.4 of the main paper. Refer also to the separate `envs/sustaindc/README_SustainDC.md` in the codebase.

## B.1  Datacenter IT Model

The DC receives a continual stream of computational tasks from the top-level agent. Each task requires varying amounts of computational resources namely CPU processing, GPU acceleration, and memory allocation. As these tasks are processed, they generate heat that must be managed by the DC's cooling systems.

When computational workloads arrive at the DC, they are distributed across the servers. The servers are arranged in racks, with each rack containing multiple servers. Each server houses CPUs, potentially GPUs, memory modules, and cooling fans. The power consumption of these components directly corresponds to the computational load they process.

To model the power consumption accurately, the environmental conditions in which these components operate must be taken into consideration. The inlet temperature, the temperature of air entering a server for cooling is a critical parameter that affects both performance and power consumption.

For a server $i$ in rack $r$ at time $t$, the inlet temperature $T_{inlet,i,t}$ is determined by:

$$T_{inlet,i,t} = \Delta T_{supply,r} + T_{CRAC,supply,t} \tag{1}$$

where $T_{CRAC,supply,t}$ is the supply air temperature from the Computer Room Air Conditioning (CRAC) unit, and $\Delta T_{supply,r}$ represents the temperature rise that occurs as the cool air travels from the CRAC unit to the server inlet. This spatial temperature difference is rack-specific and derived from Computational Fluid Dynamics (CFD) simulations of the DC [46].

With the inlet temperature established, we can now model the power consumption of each component as it processes the workload.

**CPU Power Consumption:**  CPUs form the primary computational engine of the DC, handling a wide range of tasks from basic logic operations to complex calculations. As CPUs execute computational tasks, their power consumption varies with both utilization and temperature. When a CPU is heavily utilized, it draws more power to drive the increased computational activity.

For a CPU in server $i$ at time $t$, the power consumption $P_{CPU,i,t}$ is modeled as:

$$R_{base,i,t} = m_{CPU} \cdot T_{inlet,i,t} + c_{CPU} \tag{2}$$

$$R_{CPU,i,t} = R_{base,i,t} + R_{shift} \cdot \frac{U_{CPU,i,t}}{100} \tag{3}$$

$$P_{CPU,i,t} = P_{full} \cdot R_{CPU,i,t} \tag{4}$$

Here, $P_{idle}$ represents the power consumed when the CPU is idle (typically 110W for the modeled HPE ProLiant servers), while $P_{full}$ is the power at full load (typically 170W). The power ratio $R_{CPU,i,t}$ captures how power scales with temperature and utilization, calculated from a baseline ratio $R_{base,i,t}$ that depends on inlet temperature, plus an adjustment for utilization. The coefficients $m_{CPU}$ and $c_{CPU}$ define the temperature dependence, calculated based on the power ratio bounds and temperature range:

$$m_{CPU} = \frac{CPU\_POWER\_RATIO\_UB[0] - CPU\_POWER\_RATIO\_LB[0]}{INLET\_TEMP\_RANGE[1] - INLET\_TEMP\_RANGE[0]} \tag{5}$$

$$c_{CPU} = CPU\_POWER\_RATIO\_UB[0] - m_{CPU} \cdot INLET\_TEMP\_RANGE[1] \tag{6}$$

The parameter $R_{shift}$ (calculated as $CPU\_POWER\_RATIO\_LB[1] - CPU\_POWER\_RATIO\_LB[0]$) determines how strongly utilization affects power. The CPU utilization $U_{CPU,i,t}$ ranges from 0% (idle) to 100% (fully utilized) [46].

This model ensures that even at zero utilization, the CPU consumes at least its idle power, and as utilization increases, power consumption scales accordingly, influenced by the inlet temperature. The model parameters are derived from server characteristics specified in the `configs/dcs/dc_config.json` file, including:

- CPU_POWER_RATIO_LB: Lower bounds of power ratio at minimum and maximum utilizations [0.01, 1.00]

- CPU_POWER_RATIO_UB: Upper bounds of power ratio at minimum and maximum utilizations [0.03, 1.02]

- INLET_TEMP_RANGE: Operating temperature range for inlet air [16°C, 28°C]

- HP_PROLIANT: Reference power values for the server model [110W, 170W]

**GPU Power Consumption:** GPUs serve as specialized accelerators in modern DCs, handling parallel processing tasks such as machine learning training, inference, and graphics rendering.

For a GPU in server $i$ at time $t$, we model the power consumption $P_{GPU,i,t}$ as given by [47]:

$$\alpha = P_{idle\_GPU} \tag{7}$$

$$\beta = P_{full\_GPU} - P_{idle\_GPU} \tag{8}$$

$$P_{GPU,i,t} = \alpha + \beta \cdot \log_2(1 + U_{GPU,i,t}) \tag{9}$$

Here, $P_{idle\_GPU}$ represents the power consumed when the GPU is idle (typically 25W for our modeled NVIDIA V100 GPUs), while $P_{full\_GPU}$ is the power at full load (typically 250W). The parameters $\alpha$ and $\beta$ represent the idle power and the scaling factor, respectively. The GPU utilization $U_{GPU,i,t}$ ranges from 0% (idle) to 100% (fully utilized).

When both CPUs and GPUs are present in a server, the total server power includes contributions from both components, and the IT fan power responds to the total thermal load between the CPU, Memory and GPU to ensure adequate cooling for all components. The model parameters for our NVIDIA V100 GPUs are derived from specifications in the `configs/dcs/dc_config.json` file, including:

- NVIDIA_V100: Reference power values for the GPU model [25W idle, 250W full load]

**Memory Power Consumption:** Memory modules provide the temporary storage needed for computation, holding both data and instructions. While memory power consumption can vary with access patterns and utilization, a substantial portion is static power that remains relatively constant regardless of utilization [48].

For simplicity, we model static memory power as proportional to the installed capacity, distributed evenly across racks:

$$P_{memory,r,t} = 0.07 \cdot \frac{M_{GB}}{N_{racks}} \tag{10}$$

where $P_{memory,r,t}$ is the memory power consumption of rack $r$ at time $t$, $M_{GB}$ is the total DC memory capacity, and $N_{racks}$ is the number of racks. The coefficient 0.07 reflects typical power characteristics of modern DRAM modules, as established in the GreenDIMM study [48].

This model captures the baseline power consumption of memory systems without explicitly modeling dynamic power variations.

**Server Fan Power Consumption:** As computational components generate heat, server fans must adjust their speed to maintain safe operating temperatures. Higher component utilization and higher inlet temperatures both necessitate increased airflow, which requires higher fan speeds.

For the fans in server $i$ at time $t$, power consumption $P_{fan,i,t}$ is modeled as:

$$R_{base,i,t} = m_{fan} \cdot T_{inlet,i,t} + c_{fan} \tag{11}$$

$$R_{fan,i,t} = R_{base,i,t} + R_{shift,fan} \cdot \frac{U_{eff,i,t}}{100} \tag{12}$$

$$P_{fan,i,t} = P_{ref} \cdot \frac{R_{fan,i,t}}{R_{ref}} \tag{13}$$

Here, $P_{fan,i,t}$ is the fan power consumption, $P_{ref}$ is the reference power, and $R_{fan,i,t}$ is the fan velocity ratio compared to a reference ratio $R_{ref}$. The coefficients $m_{fan}$ and $c_{fan}$ control the baseline relationship between inlet temperature and fan speed, while $R_{shift,fan}$ determines how strongly utilization affects fan speed. The effective load $U_{eff,i,t}$ represents the total of CPU, Memory and GPU load to ensure adequate cooling for all components.

The coefficients are calculated based on the fan characteristics:

$$m_{fan} = \frac{IT\_FAN\_AIRFLOW\_RATIO\_UB[0] - IT\_FAN\_AIRFLOW\_RATIO\_LB[0]}{INLET\_TEMP\_RANGE[1] - INLET\_TEMP\_RANGE[0]} \tag{14}$$

$$c_{fan} = IT\_FAN\_AIRFLOW\_RATIO\_UB[0] - m_{fan} \cdot INLET\_TEMP\_RANGE[1] \tag{15}$$

The parameter $R_{shift,fan}$ is calculated as the difference between the lower bounds of airflow ratio at maximum and minimum utilizations:

$$R_{shift,fan} = IT\_FAN\_AIRFLOW\_RATIO\_LB[1] \tag{16}$$
$$- IT\_FAN\_AIRFLOW\_RATIO\_LB[0]$$

The model parameters are derived from fan characteristics [46] in `configs/dcs/dc_config.json`:

- IT_FAN_AIRFLOW_RATIO_LB: Lower bounds at min/max utilizations [0.01, 0.225]
- IT_FAN_AIRFLOW_RATIO_UB: Upper bounds at min/max utilizations [0.225, 1.0]
- IT_FAN_FULL_LOAD_V: Volumetric flow rate for the IT fan (0.051 m³/s)
- ITFAN_REF_V_RATIO: Reference volumetric flow rate ratio (1.0)
- ITFAN_REF_P: Reference power (10.0W)

**Total Rack and Datacenter Power:** The power consumption of a rack $r$ at time $t$ is the sum of all component powers across its servers:

$$P_{rack,r,t} = \sum_{i=1}^{N_{servers,r}} \left( P_{CPU,i,t} + P_{fan,i,t} + P_{GPU,i,t} \right) + P_{memory,r,t} \tag{17}$$

where $N_{servers,r}$ is the number of servers in rack $r$. The total DC IT power consumption is then:

$$P_{IT,t} = \sum_{r=1}^{N_{racks}} P_{rack,r,t} \tag{18}$$

This power consumption not only determines the energy use of the IT equipment but also generates heat that must be managed by the cooling system, creating a complex interplay between computational load, power consumption, and thermal management.

In practice, the DC model processes each incoming task by calculating its impact on the utilization of CPU, GPU, and memory resources across the servers. These utilization levels then drive the power consumption calculations for each component, ultimately determining the total DC power consumption and thermal load.

The configuration of the DC, including the number of racks, servers per rack, and component specifications, is defined in the `configs/dcs/dc_config.json` file, which allows for customized simulation of different DC architectures and hardware configurations.

### B.2 Thermal Dynamics

**Thermal Dynamics:** In DC thermal modeling, a critical principle is that all power consumed by IT equipment ($P_{IT}$) is eventually converted to heat ($Q_{heat}$). This conversion occurs regardless of the specific components involved, with power becoming heat through various forms of electrical resistance. This equivalence principle forms the basis of our thermal calculations:

$$P_{IT} = Q_{heat} \tag{19}$$

The heat generated by servers warms the cooling air as it passes through the racks. For a rack $i$ at time $t$, the outlet temperature is calculated using an energy balance equation. Given an inlet temperature $T_{in,i,t}$, the outlet temperature $T_{out,i,t}$ is modeled as:

$$T_{out,i,t} = T_{in,i,t} + \frac{c \cdot P_{rack,i,t}^d}{Cp_{air} \cdot \rho_{air} \cdot \dot{V}_{fan,i,t}^e \cdot f} + g \tag{20}$$

Here, $P_{rack,i,t}$ represents the total rack power (including CPU, GPU, fan, and memory power), and $\dot{V}_{fan,i,t}$ is the total volumetric airflow rate through the rack. The constants $c$, $d$, $e$, $f$, and $g$ are empirically derived coefficients that capture the non-linear relationship between power, airflow, and temperature rise. $Cp_{air}$ and $\rho_{air}$ represent the specific heat capacity and density of air, respectively. This thermal model ensures that higher power consumption results in higher outlet temperatures, while increased airflow (from faster fan speeds) reduces the temperature rise. The model includes a correction term $g$ to account for heat bypass and other practical considerations.

The Computer Room Air Conditioning (CRAC) unit receives return air from all racks in the DC. The CRAC return air temperature, $T_{CRAC,t}$, is calculated as the average of all rack outlet temperatures plus their respective return approach temperatures:

$$T_{CRAC,t} = \frac{1}{n} \sum_{i=1}^{n} (T_{out,i,t} + \Delta T_{return,i}) \tag{21}$$

where $n$ is the number of racks, and $\Delta T_{return,i}$ is the return approach temperature for rack $i$ - a parameter that models the temperature increase from the rack outlet to the CRAC return, reflecting the specific DC layout and airflow patterns. These approach temperatures are specified in the `configs/dcs/dc_config.json` file as RACK_RETURN_APPROACH_TEMP_LIST [46].

This multi-stage thermal model captures the complete heat transfer path from IT equipment to the cooling system, enabling accurate prediction of cooling requirements and energy consumption under various workload and environmental conditions.

### B.3 HVAC System Modeling

The DC cooling system follows the journey of heat from its generation in IT equipment to its eventual rejection to the outside environment. This thermal pathway involves multiple stages of heat transfer and several interlinked mechanical systems.

The cooling cycle begins at the server racks, where electrical power consumed by CPUs, GPUs, and other components is converted to heat. As this heat builds up, server fans increase their speed to maintain safe operating temperatures, drawing cool air from the raised floor plenum. This cool air, supplied at a carefully controlled temperature ($T_{setpoint}$), absorbs heat as it passes through the servers and exits at a higher temperature at the rack outlets.

The warmed air then travels through the DC space towards the CRAC units, gaining additional heat along the way due to mixing and recirculation patterns specific to the DC layout (modeled by return approach temperatures). Upon reaching the CRAC units, the total cooling load that must be removed is calculated as:

$$Q_{CRAC} = \dot{m}_{sys} \cdot Cp_{air} \cdot (T_{CRAC} - T_{setpoint}) \tag{22}$$

where $Q_{CRAC}$ is the CRAC cooling load, $\dot{m}_{sys}$ is the mass flow rate of air, $Cp_{air}$ is the specific heat capacity of air, $T_{CRAC}$ is the CRAC return air temperature, and $T_{setpoint}$ is the CRAC supply air temperature setpoint.

Here, $\dot{m}_{sys}$ represents the mass flow rate of air that must be conditioned, determined by the DC's IT load. CRAC units use powerful fans to move this air, with power consumption that follows a cubic relationship with airflow:

$$P_{CRACfan} = P_{ref\_CRAC} \cdot \left( \frac{CRAC\_SUPPLY\_AIR\_FLOW\_RATE\_pu}{CRAC\_REFRENCE\_AIR\_FLOW\_RATE\_pu} \right)^3 \tag{23}$$

where $P_{CRACfan}$ is the CRAC fan power consumption, $P_{ref\_CRAC}$ is the reference CRAC fan power, $CRAC\_SUPPLY\_AIR\_FLOW\_RATE\_pu$ is the supply air flow rate per unit of IT load, and $CRAC\_REFRENCE\_AIR\_FLOW\_RATE\_pu$ is the reference air flow rate.

As the air passes through the CRAC cooling coils, it transfers its heat to chilled water circulating in a closed loop. The chiller, which acts as the heart of the cooling system, removes this heat from the water. The chiller's power consumption varies with both cooling load and ambient conditions:

$$P_{chiller} = \frac{FLP \cdot FPR \cdot AC}{COP} \cdot F \tag{24}$$

where $P_{chiller}$ is the chiller power consumption, $FLP$ is the fractional full load power (how power scales with part-load operation), $FPR$ is the full power ratio (adjustment for ambient conditions), $AC$ is the available capacity (maximum cooling possible at current conditions), $COP$ is the coefficient of performance (cooling effectiveness), and $F$ is the cycling fraction (accounts for efficiency losses during cycling at low loads).

The heat extracted by the chiller must ultimately be rejected to the outside environment, typically through a cooling tower. In the cooling tower, condenser water from the chiller is sprayed over fill material while tower fans draw air through the system. The fan power required depends on the airflow needed to reject the heat:

$$P_{CTfan} = P_{ref\_CT} \cdot \left( \frac{V_{air}}{CT_{ref\_air\_flow}} \right)^3 \tag{25}$$

where $P_{CTfan}$ is the cooling tower fan power, $P_{ref\_CT}$ is the reference cooling tower fan power, $V_{air}$ is the required volumetric air flow rate, and $CT_{ref\_air\_flow}$ is the reference volumetric air flow rate for cooling towers.

Completing the cycle, water pumps circulate both the chilled water and condenser water through their respective loops. The power consumed by these pumps is determined by the hydraulic properties of each system:

$$P_{CW} = \frac{CW\_PRESSURE\_DROP \cdot CW\_WATER\_FLOW\_RATE}{CW\_PUMP\_EFFICIENCY} \tag{26}$$

where $P_{CW}$ is the chilled water pump power, $CW\_PRESSURE\_DROP$ is the pressure drop in the chilled water loop, $CW\_WATER\_FLOW\_RATE$ is the chilled water volumetric flow rate, and $CW\_PUMP\_EFFICIENCY$ is the chilled water pump efficiency.

$$P_{CT} = \frac{CT\_PRESSURE\_DROP \cdot CT\_WATER\_FLOW\_RATE}{CT\_PUMP\_EFFICIENCY} \tag{27}$$

where $P_{CT}$ is the condenser water pump power, $CT\_PRESSURE\_DROP$ is the pressure drop in the condenser water loop, $CT\_WATER\_FLOW\_RATE$ is the condenser water volumetric flow rate, and $CT\_PUMP\_EFFICIENCY$ is the condenser water pump efficiency.

This interconnected system ensures that heat is continuously removed from the DC, maintaining optimal operating conditions for IT equipment while striving for energy efficiency. The efficiency of the overall cooling system depends on careful balance and control of each component, with operating parameters derived from specifications in the `configs/dcs/dc_config.json` file:

- C_AIR: Specific heat capacity of air (1006 J/kg·K)
- RHO_AIR: Air density (1.225 kg/m³)
- CRAC_FAN_REF_P: Reference power for CRAC fans (150 W)
- CRAC_SUPPLY_AIR_FLOW_RATE_pu: Mass air flow rate per unit IT load (0.00005663 kg/s/W)
- CRAC_REFRENCE_AIR_FLOW_RATE_pu: Reference mass air flow rate (0.00009438 kg/s/W)
- CT_FAN_REF_P: Reference power for cooling tower fans (1000 W)
- CT_REFRENCE_AIR_FLOW_RATE: Reference volumetric air flow rate for cooling towers (2.8315 m³/s)
- CW_PRESSURE_DROP: Pressure drop in chilled water loop (300 kPa)
- CT_PRESSURE_DROP: Pressure drop in condenser water loop (300 kPa)
- CW_PUMP_EFFICIENCY: Chilled water pump efficiency (0.87)
- CT_PUMP_EFFICIENCY: Condenser water pump efficiency (0.87)
- CW_WATER_FLOW_RATE: Chilled water volumetric flow rate (0.0011 m³/s)
- CT_WATER_FLOW_RATE: Condenser water volumetric flow rate (0.0011 m³/s)

## B.4  Water Usage Model

In addition to energy consumption, water usage represents a critical environmental consideration for modern DCs, particularly those employing evaporative cooling towers. Our model quantifies this resource demand, enabling operators to balance thermal management needs with water conservation goals.

The cooling tower functions as a heat rejection system where warm condenser water from the chiller system passes through a liquid to air heat exchanger to dump waste heat to the ambient air. If ambient temperatures are too high water can be sprayed on the air side of the heat exchangers to utilize the latent heat of evaporation to enhance the heat exchanger effectiveness. This evaporation, while necessary for cooling, constitutes the as primary source of water consumption in the DC cooling system.

Our water usage model captures this physical process through a set of empirically-validated equations based on research in DC water efficiency management [49] and experimental analysis of cooling tower evaporation [50]. The model begins by calculating the temperature range—the difference between the hot water entering the cooling tower and the cooled water exiting it:

$$T_{range} = T_{hot} - T_{cold} \tag{28}$$

where $T_{hot}$ is the temperature of water returning from the chiller condenser and $T_{cold}$ is the temperature of cooled water returning to the chiller.

This temperature range directly influences the evaporation rate, as larger temperature differences require more evaporative cooling. The relationship is captured in a baseline intercept value derived from prior research:

$$y_{intercept} = 0.3528 \cdot T_{range} + 0.101 \tag{29}$$

The ambient wet-bulb temperature—a measure that combines air temperature and humidity—further modifies the evaporation rate. Higher wet-bulb temperatures reduce the cooling tower's efficiency, requiring more water consumption for the same cooling effect:

$$W_{norm} = 0.044 \cdot T_{wb} + y_{intercept} \tag{30}$$

where $T_{wb}$ is the ambient wet-bulb temperature and $W_{norm}$ is the normalized water usage rate (m³/hr per unit of heat rejection).

Beyond evaporation, cooling towers lose additional water through drift—droplets carried away by airflow. This secondary loss is calculated as:

$$W_{drift} = W_{evap} \cdot D_{rate} \tag{31}$$

where $D_{rate}$ is the drift rate.

The total water consumption combines both evaporative and drift losses:

$$W_{total} = W_{evap} + W_{drift} \tag{32}$$

For operational monitoring, this hourly consumption rate is converted to 15-minute intervals in more familiar units:

$$W_{15min} = \frac{W_{total} \cdot 1000}{4} \tag{33}$$

where $W_{15min}$ is the water usage in liters per 15-minute interval.

This model enables DC operators to predict water consumption under varying workloads and ambient conditions. By understanding these relationships, operators can implement strategies to reduce water usage, such as raising cold water set points during periods of lower wet-bulb temperatures or implementing advanced control systems that optimize the balance between energy and water efficiency.

Furthermore, the model supports sustainability planning by quantifying the water footprint associated with different cooling strategies, facilitating informed decisions about DC design and operation in regions with varying water availability constraints.

## B.5 Heat Recovery Unit

Modern DCs represent not only significant energy consumers but also substantial sources of waste heat. Our model incorporates a heat recovery system that captures a portion of this thermal energy for beneficial reuse, reducing both cooling requirements and external heating demands.

The heat recovery potential is calculated using a temperature-differential approach:

$$T_{delta} = \max(OFFICE\_GUIDE\_TEMP - T_{ambient}, 0) \tag{34}$$

where $T_{delta}$ is the effective temperature difference for heat transfer, $T_{office}$ is the office guide temperature (typically 20-22°C), and $T_{ambient}$ is the current ambient outdoor temperature. The maximum function ensures that heat recovery only occurs when the outdoor temperature is below the desired indoor temperature.

The actual heat recovery capacity is then determined by:

$$Q_{DC\_office} = AVE\_HLP \cdot DC\_AREA\_PU \cdot P_{IT\_max} \cdot T_{delta} \tag{35}$$

$$Q_{ext\_office} = AVE\_HLP \cdot OFFICE\_BUILDING\_AREA \cdot T_{delta} \tag{36}$$

$$Q_{recovery} = Q_{DC\_office} + Q_{ext\_office} \tag{37}$$

where $AVE\_HLP$ is the average heat loss parameter, $A_{DC}$ is the DC office area (normalized per unit of IT load), $A_{office}$ is the adjacent office building area, and $P_{IT\_max}$ is the maximum IT load.

This recovered heat directly reduces the cooling load on the CRAC system:

$$Q_{CRAC\_reduced} = Q_{CRAC} - \min(Q_{recovery}, 0.25 \cdot P_{IT}) \tag{38}$$

The model enforces a practical limitation that no more than 25% of the current IT load can be redirected through heat recovery, representing thermodynamic and implementation constraints. This recovered energy simultaneously reduces cooling energy requirements and offsets heating demands in office spaces, creating a dual efficiency benefit.

The heat recovery system impacts several aspects of the overall energy model:

- Reduced cooling load for the chiller system
- Lower cooling tower fan power due to decreased heat rejection needs
- Decreased water consumption in the cooling tower
- Offset heating energy for office spaces

The model parameters for heat recovery are specified in the DC configuration:

- AVE_HLP: Average heat loss parameter (W/m²·K)
- DC_AREA_PU: DC office area per unit of IT load (m²/W)
- OFFICE_BUILDING_AREA: Adjacent office building area (m²)
- OFFICE_GUIDE_TEMP: Target office temperature (°C)

This approach represents a practical implementation of waste heat recovery that can be deployed in real DC environments without requiring specialized district heating infrastructure.

## C  Time Granularity Rationale (15-Minute Timestep)

DCcluster-Opt operates on a 15-minute discrete timestep. This choice reflects a balance between simulation fidelity, computational tractability, and alignment with real-world operational and data characteristics:

- **Supporting Literature:** Prior research in data center energy optimization, dynamic control, and sustainable computing has often employed similar timesteps (e.g., 5-15 minutes) for modeling and control, demonstrating its efficacy [51, 52, 53, 54, 55].

- **Data Availability:** Key external data feeds, such as real-time grid carbon intensity from Electricity Maps [7] or granular electricity pricing from ISOs and market monitors [8], are often reported at 15-minute or hourly intervals. Aligning the simulation timestep allows direct ingestion of this data.

- **Operational Cadence & Billing:** While individual task durations vary, high-level resource allocation decisions or batch scheduling in large clusters often occur at intervals coarser than per-second [6]. Furthermore, cloud provider billing cycles frequently aggregate usage over minute-level intervals (e.g., 1, 5, or 15 minutes), making decisions at this granularity relevant for cost management.

- **Thermal Inertia:** Datacenter cooling systems (chillers, cooling towers, thermal mass of the building/equipment) exhibit significant inertia [5]. Their response to changes in IT load or control setpoints occurs over minutes, not instantaneously. Simulating at 15-minute intervals captures the relevant timescale for these thermal dynamics and HVAC energy responses without the computational overhead or numerical instability of much finer-grained simulations.

This 15-minute granularity allows DCcluster-Opt to model the essential geo-temporal dynamics relevant to sustainable scheduling while remaining computationally feasible for training RL agents and running extensive evaluations.

## D  Dataset Details

This section provides extended information on the datasets integrated into DCcluster-Opt, supplementing Section 4 of the main paper.

## D.1 Supported Datacenter Locations and Network Mapping

DCcluster-Opt provides integrated real-world environmental data (electricity price, carbon intensity, weather) for a diverse set of global locations, enabling users to simulate geographically distributed data center clusters. The table below lists the currently supported `location` codes (to be used in `datacenters.yaml`), their corresponding geographical region/market, the specific cloud provider region strings they map to for transmission **cost** calculations (via `utils/transmission_region_mapper.py`), and the macro-cluster they map to for transmission **delay** calculations (via `data/network_cost/network_delay.py`).

Table 7: Supported Locations and their Network Region/Cluster Mappings.

| DCcluster-Opt Location Code | Geographical Region / Market Description | Cloud Region Mapping (for Cost) (GCP / AWS / Azure) | Macro-Cluster (for Delay) |
|---|---|---|---|
| US-NY-NYIS | New York, USA (NYISO) | `us-east1` / `us-east-1` / `East US` | US |
| US-CAL-CISO | California, USA (CAISO) | `us-west1` / `us-west-1` / `West US` | US |
| US-TEX-ERCO | Texas, USA (ERCOT) | `us-central1`[a] / `us-east-1-dwf-1` / `South Central US` | US |
| US-MIDA-PJM | Mid-Atlantic, USA (PJM) | `us-east4`[a] / `us-east-1`[a] / `East US 2`[a] | US |
| CA-ON | Ontario, Canada (IESO) | `northamerica-northeast1` / `ca-central-1` / `Canada Central` | US |
| BR-SP | São Paulo, Brazil (ONS) | `southamerica-east1` / `sa-east-1` / `Brazil South` | SA |
| CL-SIC | Norte Grande, Chile (CEN) | `southamerica-west1`[a] / `us-east-1-chl-1`[b] / `Chile Central`[a] | SA |
| DE-LU | Germany+Luxembourg (ENTSO-E) | `europe-west3` / `eu-central-1` / `Germany West Central` | EU |
| FR | France (ENTSO-E) | `europe-west9` / `eu-west-3` / `France Central` | EU |
| ES | Spain (OMIE/ENTSO-E) | `europe-southwest1` / `eu-south-1` / `Spain Central` | EU |
| PT | Portugal (OMIE/ENTSO-E) | `europe-southwest1`[a] / `eu-south-1`[a] / `Portugal North`[a] | EU |
| GB | Great Britain (National Grid ESO) | `europe-west2` / `eu-west-2` / `UK South` | EU |
| BE | Belgium (ENTSO-E) | `europe-west1` / `eu-west-1`[a] / `West Europe` | EU |
| NL | Netherlands (ENTSO-E) | `europe-west4` / `eu-west-1` / `West Europe` | EU |
| AT | Austria (ENTSO-E) | `europe-west3`[a] / `eu-central-1`[a] / `Austria East`[a] | EU |
| CH | Switzerland (ENTSO-E) | `europe-west6` / `eu-central-2` / `Switzerland North` | EU |
| SG | Singapore (USEP/EMA) | `asia-southeast1` / `ap-southeast-1` / `Southeast Asia` | AP |
| JP-TK | Tokyo Area, Japan (JEPX) | `asia-northeast1` / `ap-northeast-1` / `Japan East` | AP |
| KR | South Korea (KPX) | `asia-northeast3` / `ap-northeast-2` / `Korea Central` | AP |
| IN | Mumbai Area, India (POSOCO) | `asia-south1` / `ap-south-1` / `Central India` | AP |
| AU-NSW | New South Wales, Australia (AEMO) | `australia-southeast1` / `ap-southeast-2` / `Australia East` | AP |
| AU-VIC | Victoria, Australia (AEMO) | `australia-southeast2`[a] / `ap-southeast-2`[a] / `Australia Southeast`[a] | AP |
| ZA | South Africa (Eskom) | `africa-south1`[c] / `af-south-1` / `South Africa North` | SA[d] |

[a]Approximation or nearest major region used if a direct 1:1 cloud provider region is not obvious for the specific market area or if the provider has limited presence. Check `utils/transmission_region_mapper.py` for exact mappings.
[b]AWS `us-east-1-chl-1` is a Local Zone in Santiago, Chile.
[c]GCP does not currently have a dedicated "africa-south1"; mapping might use a broader regional approach or proxy.
[d]Based on current configuration in `data/network_cost/network_delay.py`, South African regions map to the `SA` (South America) macro-cluster for delay calculations. This may be revised if specific Africa-to-other-continent delay parameters become available or are modeled.

Users wishing to incorporate additional locations not listed above should refer to Appendix D.8 for guidance on providing the necessary data (price, carbon intensity, weather) and updating the relevant mapping configurations for transmission cost and delay. We aim to continuously expand this list of supported regions in future releases of DCcluster-Opt.

## D.2 AI Workloads (Alibaba GPU Cluster 2020)

The benchmark utilizes the public Alibaba Cluster Trace 2020 [6], capturing real-world GPU usage from a large-scale production AI platform (over 6,500 GPUs across 1800 machines over two months).

- **Preprocessing:** To adapt this trace for continuous, long-term simulation, we apply several steps (detailed in `data/workload/README.md`):
  - Filtering tasks shorter than 15 minutes.
  - Temporally extending the 2-month trace to a full year by analyzing and replicating daily/weekly patterns.

- Assigning a probabilistic origin datacenter to each task based on regional population weights and local time-of-day activity factors (see Section F.5 for logic, implemented in `utils/workload_utils.py::assign_task_origins`).
  - Grouping tasks into 15-minute arrival intervals (UTC).
- **Format & Content:** The processed data is stored as a Pandas DataFrame in a `.pkl` file (e.g., `data/workload/alibaba_2020_dataset/result_df_full_year_2020.pkl`). Each row corresponds to a 15-minute interval and contains a NumPy array (`tasks_matrix`) detailing the tasks arriving in that interval. Task features include duration, resource requests (CPU cores, GPU units, memory GB - normalized from original percentages at runtime), and estimated data bandwidth (GB). The full list of columns in the `tasks_matrix` is: `job_id`, `start_time` (Unix), `end_time` (Unix), `start_dt` (datetime), `duration_min`, `cpu_usage` (%), `gpu_wrk_util` (%), `avg_mem` (GB), `avg_gpu_wrk_mem` (GB), `bandwidth_gb`, `weekday_name`, `weekday_num`.
- **Resource Normalization:** As noted in the main paper, CPU/GPU requirements are converted from percentages to resource units via `utils/workload_utils.py::extract_tasks_from_row` during simulation.
- **Usage:** This dataset drives the simulation by defining when tasks arrive, where they originate, and their computational requirements.
- **Access:** The large processed `.pkl` file is typically distributed within a `.zip` archive in the same directory (`data/workload/alibaba_2020_dataset/`). The simulation code automatically extracts this on first use if the `.pkl` file is not found, but the `.zip` file is present.

The following figures illustrate the characteristics of the processed Alibaba workload trace.

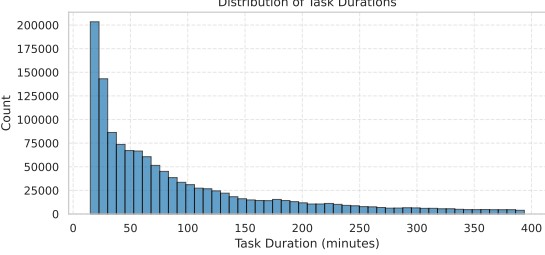

(a) Distribution of task durations (minutes) for jobs $\geq 15$ minutes.

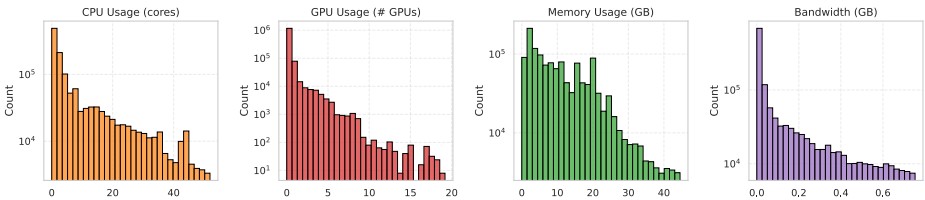

(b) Distribution of resource requests per task (CPU cores, GPU units, Memory GB, Bandwidth GB).

Figure 7: Workload duration and resource request distributions.

## D.3 Electricity Prices

Real-time electricity cost is a major factor in operational expenditure.

- **Sources:** We collect historical price data from public APIs and sources including Electricity Maps [7], GridStatus.io [8], and directly from Independent System Operators (ISOs) like CAISO, NYISO, ERCOT, and European exchanges like OMIE (details and scripts in `data/electricity_prices/`).

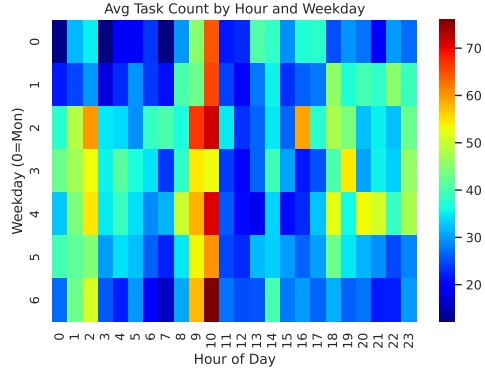

(a) Average task arrival count per hour vs. day of week.

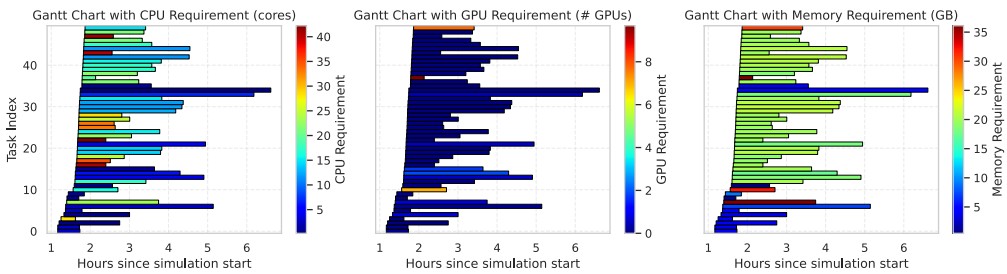

(b) Sample Gantt chart (10hr window) showing tasks colored by resource requests (CPU, GPU, Memory).

Figure 8: Temporal patterns in workload arrivals and resource demands.

- **Coverage & Standardization:** Data spans 2020-2024 for the supported global regions (see Section D.1). Raw data is processed by scripts (`data/electricity_prices/scripts/`) to standardize it into hourly UTC timestamps with prices in USD/MWh. This is then interpolated or forward-filled to match the 15-minute simulation timestep and often normalized to USD/kWh for use.

- **Usage:** Provides the time-varying electricity cost for each DC location, directly impacting the energy cost calculations within the simulation and providing a dynamic signal for cost-aware scheduling agents (via state observation and reward).

- **Storage:** Standardized data is stored in yearly CSV files under `data/electricity_prices/standardized/REGION/YEAR/`.

## D.4 Carbon Intensity

Minimizing the environmental impact requires considering the carbon intensity of the grid powering each data center.

- **Source:** We use historical grid carbon intensity data from the Electricity Maps API [7].

- **Content & Units:** Provides time-series data representing the grams of $CO_2$ equivalent emitted per kWh of electricity consumed ($gCO_2eq/kWh$) for different grid regions.

- **Coverage & Resolution:** Covers supported regions from 2021-2024, typically at hourly or finer resolution, aligned to the 15-minute simulation step.

- **Usage:** This data allows the simulation to calculate the carbon emissions associated with both the operational energy (compute + cooling) consumed at each DC and the energy estimated for data transmission. It provides a critical signal for carbon-aware scheduling.

- **Storage:** Stored in yearly CSV files under `data/carbon_intensity/REGION/YEAR/`.

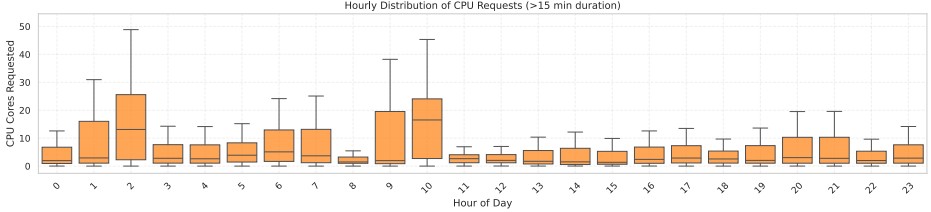

(a) Hourly distribution of CPU core requests.

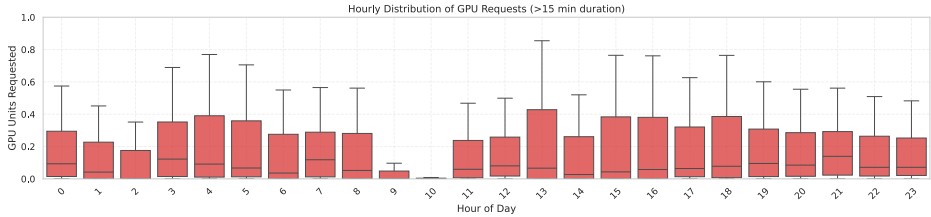

(b) Hourly distribution of GPU unit requests.

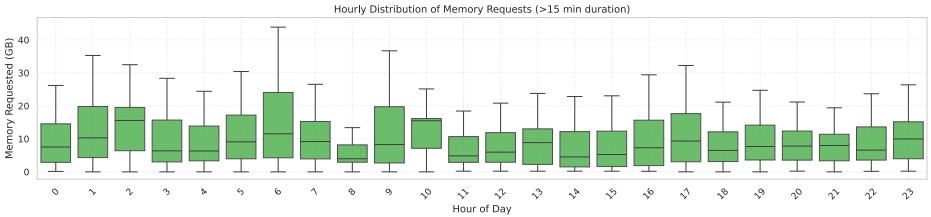

(c) Hourly distribution of Memory (GB) requests.

Figure 9: Hourly variation in resource requests within the workload trace.

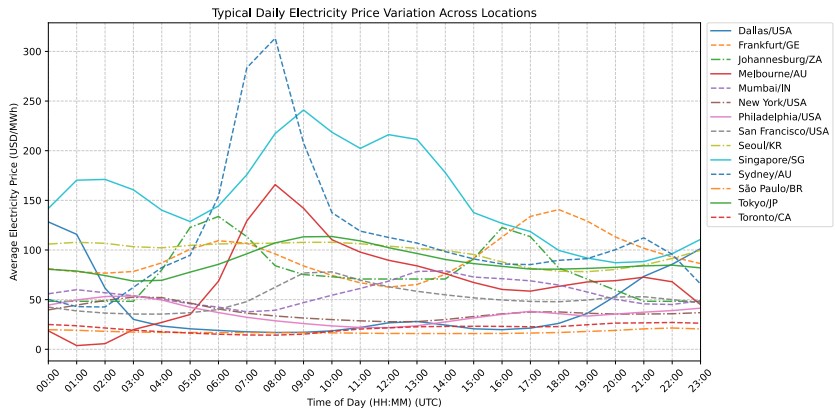

Figure 10: Average hourly electricity price profile over a typical day (UTC time) across selected regions, illustrating temporal cost variations.

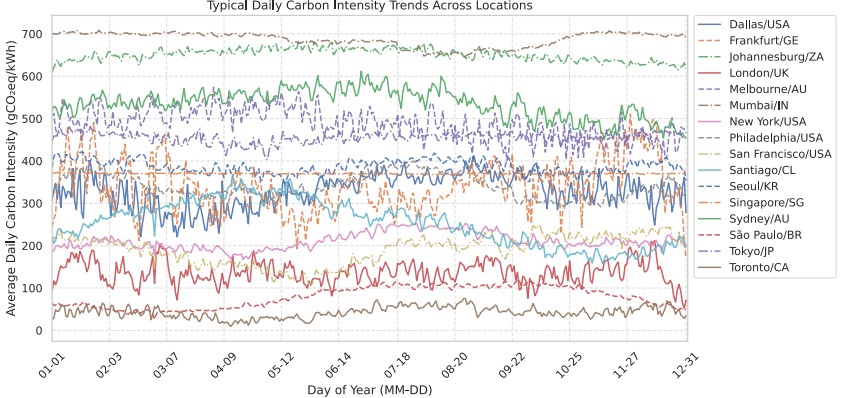

(a) Average daily carbon intensity across selected regions (gCO$_2$eq/kWh), showing geographical differences.

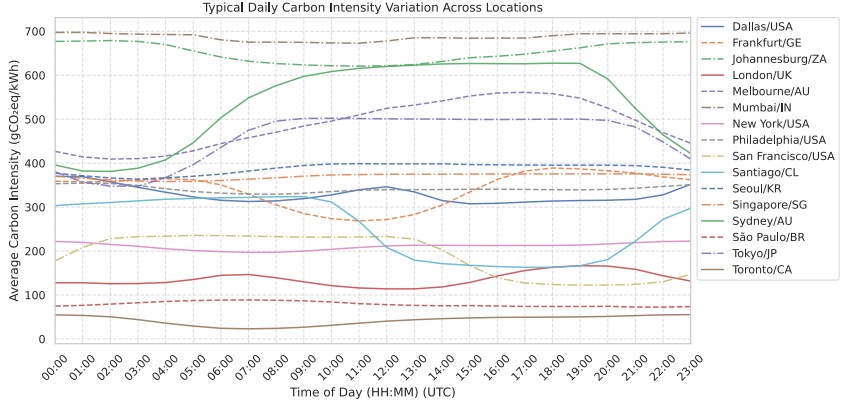

(b) Average hourly carbon intensity profile over a typical day (UTC time), highlighting temporal variations.

Figure 11: Geographical and temporal variations in grid carbon intensity, offering opportunities for carbon-aware scheduling.

### D.5   Weather Data

Ambient weather conditions significantly impact data center cooling efficiency.

- **Source:** Historical weather data is obtained via the Open-Meteo API [9].
- **Content:** Primarily uses ambient air temperature (°C) as input, though other parameters like wet-bulb temperature could be incorporated for more advanced cooling models.
- **Coverage & Resolution:** Covers supported DC locations from 2021-2024, aligned to the 15-minute simulation step.
- **Usage:** The ambient temperature directly influences the performance (Coefficient of Performance - COP) and energy consumption of the simulated HVAC system, particularly the chiller model (Section 3.4).
- **Storage:** Stored in yearly JSON files under `data/weather/REGION/YEAR/`.

### D.6   Transmission Costs (per-GB)

Moving data between geographically dispersed sites incurs direct monetary costs.

- **Sources:** Based on publicly available inter-region data egress/transfer pricing from major cloud providers: AWS (`https://aws.amazon.com/ec2/pricing/on-demand/`), GCP

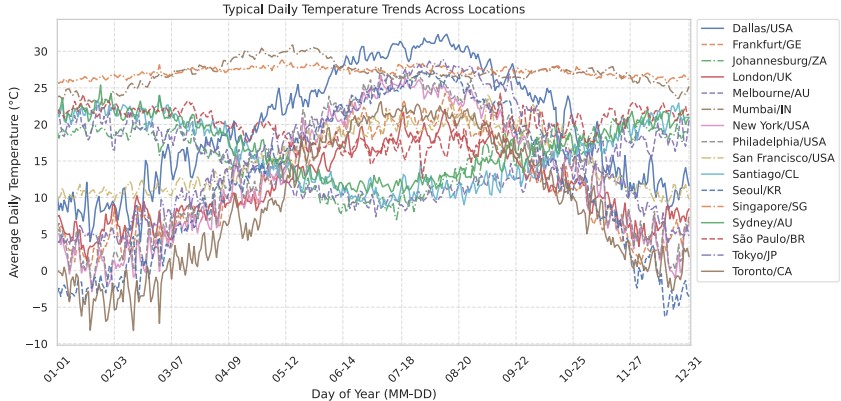

Figure 12: Average daily temperature across selected datacenter regions (°C), showing seasonal variations that impact cooling system load and efficiency.

(`https://cloud.google.com/vpc/pricing`), Azure (`https://azure.microsoft.com/en-us/pricing/details/bandwidth/`)

- **Format:** Compiled into provider-specific cost matrices (CSV format) representing the cost in USD per GB transferred between specific cloud regions.
- **Usage:** When a task is routed remotely, the cost is calculated based on its 'bandwidth_gb' and the relevant entry in the cost matrix corresponding to the origin/destination regions (mapped from DC locations via `utils/transmission_region_mapper.py`). This cost can be factored into the reward signal.
- **Storage:** Stored as CSV files in `data/network_cost/`.

## D.7 Transmission Delay Parameters

Network latency impacts task completion times in distributed systems.

- **Source:** We model delay using empirical network performance data between major geographical macro-regions (EU, US, AP, SA) published by Persico et al. [16].
- **Content:** The key parameters extracted are mean TCP throughput (Mbps) and Round-Trip Time (RTT, ms) between these macro-regions.
- **Integration:** The simulation maps specific DC locations to cloud regions and then to these macro-clusters. The extracted throughput and RTT values are used within the delay calculation formula (see Section E) implemented in `data/network_cost/network_delay.py`.
- **Usage:** Determines the "in-transit" time for remotely assigned tasks, affecting their arrival time at the destination queue.

## D.8 Adding Custom Locations and Regions

DCcluster-Opt is designed to be extensible, allowing users to incorporate new geographical locations beyond the initially provided set. To add a custom location and integrate its associated data for simulation, follow these general steps:

1. **Define a New Location Code and Datacenter Configuration:**
   - Choose a unique string identifier for your new location (e.g., `"My-City-GridOperator"`).
   - Add a new data center entry to your `configs/env/datacenters.yaml` file using this new location code. Specify its `dc_id`, `timezone_shift`, resource capacities (`total_cores`, `total_gpus`, `total_mem`), `population_weight`, and the path to its `dc_config_file` (which can be an existing one like `configs/dcs/dc_config.json` or a custom one). Example entry in `datacenters.yaml`:

```
    - dc_id: 6 # Next available ID
      location: "My-City-GridOperator"
      timezone_shift: 3 # Example: UTC+3
      population_weight: 0.1
      total_cores: 70000
      total_gpus: 500
      total_mem: 64000
      dc_config_file: "configs/dcs/dc_config.json"
```

2. **Provide Environmental Datasets:** For your new location code (e.g., "My-City-GridOperator"), you must provide the corresponding time-series data files in the `data/` directory, following the existing structure:

   - **Electricity Prices:** Create a directory `data/electricity_prices/standardized/My-City-GridOperator/YEAR/` and place CSV files (e.g., `My-City-GridOperator_electricity_prices_YEAR.csv`) containing hourly prices in USD/MWh with UTC timestamps. Refer to existing files for exact format.
   - **Carbon Intensity:** Create `data/carbon_intensity/My-City-GridOperator/YEAR/` and place CSV files (e.g., `My-City-GridOperator_YEAR_hourly.csv`) with hourly carbon intensity in $gCO_2eq/kWh$ and UTC timestamps.
   - **Weather Data:** Create `data/weather/My-City-GridOperator/YEAR/` and place JSON files (e.g., `YEAR.json`) containing hourly weather data (at least ambient temperature) in the format provided by Open-Meteo or compatible with `Weather_Manager`.

   Ensure data covers the simulation years you intend to use. Data managers (`ElectricityPrice_Manager`, `CI_Manager`, `Weather_Manager`) will look for files based on the location code and simulation year.

3. **Map Location to Cloud Provider Region (for Transmission Cost):**

   - Open `utils/transmission_region_mapper.py`.
   - Add your new `location_code` as a key to the relevant dictionary (e.g., `location_to_aws_region`, `location_to_gcp_region`, `location_to_azure_region`) and map it to an existing cloud provider region string (e.g., `"us-west-2"`, `"europe-central2"`). This cloud region string must correspond to a row/column header in the respective transmission cost matrix CSV file (e.g., `data/network_cost/aws_transmission_cost_matrix.csv`).
   - If you are using `cloud_provider: "custom"` in `sim_config.yaml`, add your mapping to `location_to_custom_region` and ensure your custom region name matches your `data/network_cost/custom_transmission_cost_matrix.csv`. Example addition to `location_to_aws_region`:

     ```
     "My-City-GridOperator": "eu-north-1", # Example mapping to AWS Stockholm
     ```

4. **Map Cloud Provider Region to Macro-Cluster (for Transmission Delay):**

   - Open `data/network_cost/network_delay.py`.
   - Ensure the cloud provider region string you used in the previous step (e.g., `"eu-north-1"`) is a key in the appropriate dictionary (`aws_region_to_cluster` or `azure_region_to_cluster`). Map this cloud region to one of the existing macro-clusters: `"US"`, `"EU"`, `"AP"`, or `"SA"`. Example addition to `aws_region_to_cluster`:

     ```
     'eu-north-1':       'EU',
     ```

   - If your new location falls into a geographical area not well represented by the existing four macro-clusters (e.g., Middle East, a distinct part of Africa not covered by "SA" approximation), you would need to:
   - (a) Define a new macro-cluster key (e.g., `"ME"`).
   - (b) Provide inter-macro-cluster throughput (Mbps) and RTT (ms) data for this new cluster to/from all other existing macro-clusters within the `aws_throughput`, `azure_throughput`, `aws_latency`, and `azure_latency` dictionaries in

`network_delay.py`. This would typically require sourcing new empirical data similar to Persico et al. [16] or making informed estimations.

5. **(Optional) Custom Transmission Cost Matrix:** If you need to define entirely custom inter-region transmission costs that do not align with AWS, GCP, or Azure regions:

   - Set `cloud_provider: "custom"` in your `sim_config.yaml`.
   - Create/edit `data/network_cost/custom_transmission_cost_matrix.csv`. The row and column headers in this CSV must exactly match the custom region names you define in the `location_to_custom_region` dictionary within `utils/transmission_region_mapper.py`.
   - Values in the matrix should represent the cost in USD per GB transferred from the origin (row) to the destination (column).

By following these steps, users can extend DCcluster-Opt to simulate custom geographical distributions and network characteristics, further enhancing its flexibility as a research tool. Thoroughly testing any new data integrations and mappings is recommended.

## D.9 Croissant Metadata

To enhance discoverability, interoperability, and reproducibility, DCcluster-Opt provides machine-readable metadata for its core processed datasets using the MLCommons Croissant standard [40]. While this paper is submitted primarily to the benchmark track, we believe in the importance of thorough dataset documentation for the components that drive our benchmark. The Croissant metadata files are provided in JSON-LD format and are located within the `metadata/` directory of our main open-source code repository: `https://github.com/HewlettPackard/sustain-cluster/tree/main/metadata`.

We provide separate Croissant files for each major dataset component, allowing for targeted understanding and use. For example:

- `metadata/workload_trace.jsonld`: Describes the processed Alibaba GPU Workload Trace.
- `metadata/electricity_prices.jsonld`: Describes the collection of standardized electricity price CSV files across all regions and years.
- `metadata/carbon_intensity.jsonld`: Describes the collection of standardized carbon intensity CSV files.
- `metadata/weather_data.jsonld`: Describes the collection of standardized weather JSON files.
- `metadata/transmission_cost_aws.jsonld` (and similar for GCP/Azure): Describes the specific transmission cost matrix.

Each of these `.jsonld` files defines a top-level `sc:Dataset` corresponding to that specific data component. Within each file, the metadata typically includes:

- **General Dataset Properties:**
  - `sc:name`: A human-readable name for the specific dataset component (e.g., "DCcluster-Opt Processed Alibaba GPU Workload Trace").
  - `sc:description`: A textual description of that dataset component.
  - `sc:url`: A URL pointing to the primary data location or relevant documentation for that component.
  - `scc:citation`: Citations to the original data sources relevant to that component.
  - `sc:license`: Information regarding the license of the original data and/or the processed version.
- **Distribution Information** (`sc:distribution`)**:**
  - Describes how to access the file(s) for that specific dataset component (e.g., relative paths if bundled, or details if within an archive).

- – `sc:encodingFormat`: The MIME type of the file(s) (e.g., `"application/vnd.pickle"`, `"text/csv"`, `"application/json"`).
- – `sc:sha256`: The SHA256 checksum for key data files to ensure integrity.
- – `sc:containedIn` (if applicable): Used if files are part of a larger archive referenced by another distribution. For collections of files (like yearly price data per region), a `sc:FileSet` might be used to define a pattern matching multiple files.

- **Record Sets and Fields** (`ml:RecordSet`, `ml:Field`): Describes the structure of the data records within that component.
    - – For **tabular data** (e.g., in Electricity Prices or Carbon Intensity CSVs): An `ml:RecordSet` describes the collection of rows. Each column is an `ml:Field` with attributes like `sc:name`, `sc:description`, `ml:dataType` (e.g., `sc:DateTime`, `sc:Float`), and a `scc:source` linking to the CSV column header.
    - – For the **AI Workload Trace** (`workload_trace.jsonld`): The `ml:RecordSet` describes the list of 15-minute time intervals. An `ml:Field` for `"interval_15m"` and another for `"tasks_matrix"` are defined. The `"tasks_matrix"` field uses `scc:repeated:  true` and `scc:subField` properties to detail the structure of each task within the matrix (e.g., sub-fields for `job_id`, `duration_min`, etc.).
    - – For **Weather Data** (JSON files, described in `weather_data.jsonld`): An `ml:RecordSet` describes hourly entries. `ml:Fields` for `"time"`, `"temperature_2m"`, etc., use `scc:source` with JSONPath expressions to map to the JSON structure.

Users can parse these individual Croissant `.jsonld` files programmatically using libraries such as the `croissant-data` Python package to understand specific dataset schemas, locate data files, and potentially automate data loading and validation. The same library can be used to validate our provided metadata:

```
pip install croissant-data
croissant validate --file metadata/workload_trace.jsonld # Example for one file
```

We believe this structured, component-specific metadata significantly enhances the utility and accessibility of the DCcluster-Opt datasets for the research community.

## D.10 Datasheets for Datasets

This section provides detailed datasheets for the core dataset components integrated into DCcluster-Opt, following the question framework proposed by Gebru et al. [39]. Datasheets are provided for the AI Workload Trace, Electricity Prices, Grid Carbon Intensity, Weather Data, Transmission Cost Matrices, and Transmission Delay Parameters. Each datasheet details aspects such as motivation, composition, collection, preprocessing, intended uses, distribution, and maintenance for the respective data component.

### D.10.1 Datasheet: Processed AI Workload Trace (Alibaba GPU Cluster 2020)

**Motivation (Section 3.1 from Gebru et al.):**

- **For what purpose was the dataset created? Was there a specific task in mind? Was there a specific gap that needed to be filled? Please provide a description.** This processed version of the Alibaba Cluster Trace 2020 [6] was created specifically for the DCcluster-Opt benchmark. The primary purpose is to provide a realistic, year-long stream of AI/GPU-intensive task arrivals with diverse resource requirements (CPU, GPU, memory, bandwidth) and durations. This serves as the core workload input for evaluating sustainable geo-distributed task scheduling algorithms. The specific task is to enable schedulers to make informed assignment or deferral decisions. The gap filled is the lack of publicly available, long-duration, GPU-centric workload traces pre-processed and integrated into a ready-to-use benchmark environment focused on sustainability.
- **Who created the dataset (e.g., which team, research group) and on behalf of which entity (e.g., company, institution, organization)?** The original Alibaba Cluster Trace 2020 was created and released by researchers from Alibaba Group and associated academic

institutions [6]. The specific preprocessing, temporal extension, and geographical origin assignment for use within DCcluster-Opt were performed by the authors of this paper, representing Hewlett Packard Labs.

- **Who funded the creation of the dataset? If there is an associated grant, please provide the name of the grantor and the grant name and number.** The original Alibaba trace was an independent academic/industry contribution. The processing and integration into DCcluster-Opt were supported by Hewlett Packard Enterprise.
- **Any other comments?** None.

**Composition (Section 3.2 from Gebru et al.):**

- **What do the instances that comprise the dataset represent (e.g., documents, photos, people, countries)? Are there multiple types of instances? Please provide a description.** The primary instances are 15-minute time intervals over a one-year period. Each interval contains a collection (a NumPy array called `tasks_matrix`) of individual AI task submissions. Each task within this matrix is a secondary type of instance, characterized by its resource requirements and other attributes.
- **How many instances are there in total (of each type, if appropriate)?** There are $4 \times 24 \times 365 = 35,040$ primary instances (15-minute intervals). The number of task instances (rows within each `tasks_matrix`) varies per interval, reflecting dynamic workload arrivals. The original 2-month trace contained millions of task instances; our filtered and extended yearly trace contains a comparable order of magnitude.
- **Does the dataset contain all possible instances or is it a sample? If the dataset is a sample, then what is the larger set? Is the sample representative? If so, please describe how this representativeness was validated/verified. If it is not representative, please describe why not.** The original Alibaba trace is a sample (two months) from their production AI platform. Our processed version extends this sample to a full year by analyzing and replicating daily/weekly patterns observed in the original trace. While it aims to capture realistic temporal dynamics, it is a synthetic extension and not a direct recording of a full year of Alibaba's operations. Representativeness of the original trace with respect to all AI workloads globally is unknown; it reflects Alibaba's specific platform usage. Our extension aims to preserve the observed workload characteristics over a longer, continuous period for simulation.
- **What data does each instance consist of? "Raw" data or features? In either case, please provide a description.** Each 15-minute interval instance contains a `tasks_matrix`. Each task (row) within this matrix consists of features derived from the original trace, including: `job_id` (string), `start_time` (Unix timestamp of original arrival, used for pattern analysis), `end_time` (Unix), `start_dt` (original datetime), `duration_min` (float), `cpu_usage` (float, percentage), `gpu_wrk_util` (float, percentage), `avg_mem` (float, GB), `avg_gpu_wrk_mem` (float, GB), `bandwidth_gb` (float, estimated), `weekday_name` (string), and `weekday_num` (int). These can be considered processed features from the raw logs.
- **Is there a label or target associated with each instance? If so, please provide a description.** No, this is an input workload trace. There are no pre-assigned labels or targets for scheduling decisions; the goal of the benchmark is for an agent to *learn* or *apply* optimal scheduling actions.
- **Is any information missing from individual instances? If so, please provide a description, explaining why this information is missing.** The original trace anonymized job/user identifiers. Specific task inter-dependencies beyond job grouping are not explicitly provided. The 'bandwidth_gb' feature was estimated and added during our preprocessing as it was not in the original trace.
- **Are relationships between individual instances made explicit?** Tasks are implicitly grouped by their original arrival patterns within 15-minute intervals. Tasks belonging to the same original job might share a common prefix in their 'job_id', but explicit job structures or DAGs are not part of this processed dataset.
- **Are there recommended data splits (e.g., training, development/validation, testing)? If so, please provide a description of these splits, explaining the rationale behind them.** Not explicitly defined within the dataset itself. For RL agent training, users typically simulate using a portion of the year (e.g., several months) for training and evaluate on unseen subsequent portions or different years (if multiple years of environmental data are used). The benchmark design encourages evaluating generalization across different time periods.

- **Are there any errors, sources of noise, or redundancies in the dataset? If so, please provide a description.** The original trace may contain inherent noise typical of real-world logs. Our temporal extension process, while pattern-based, introduces a synthetic element. The probabilistic origin assignment also introduces variability. Redundancies might exist if identical tasks (in terms of resource request and duration) appear, though this is typical of real workloads.
- **Is the dataset self-contained, or does it link to or otherwise rely on external resources?** [...] The processed `.pkl` file, once extracted from its `.zip` archive, is self-contained for the workload data itself. The original trace from which it was derived is an external resource (Alibaba GitHub). a) Guarantees for the original trace: Subject to Alibaba's hosting. Our processed version within DCcluster-Opt is stable. b) Archival versions: The specific version of the original trace used is documented (2020 release). We will version our processed dataset with DCcluster-Opt releases. c) Restrictions: The original Alibaba trace license typically requires non-commercial use and attribution. Users of DCcluster-Opt must also respect these terms for the underlying workload data.
- **Does the dataset contain data that might be considered confidential?** [...] No. The original Alibaba trace was anonymized by its creators. Our processing does not re-identify individuals or reveal confidential operational details.
- **Does the dataset contain data that, if viewed directly, might be offensive, insulting, threatening, or might otherwise cause anxiety? If so, please describe why.** No. It contains anonymized task execution metadata.
- **Does the dataset identify any subpopulations?** [...] No. Individual user or group identities are not present.
- **Is it possible to identify individuals [...] from the dataset? If so, please describe how.** No, not from the processed data provided within DCcluster-Opt.
- **Does the dataset contain data that might be considered sensitive in any way [...eg, race, sexual orientations...]? If so, please provide a description.** No.
- **Any other comments?** The primary purpose of this processed trace is to provide diverse and temporally realistic AI/GPU workload patterns for the scheduling benchmark.

**Collection Process (Section 3.3 from Gebru et al.):**

- **How was the data associated with each instance acquired? [...] If the data was reported by subjects or indirectly inferred/derived from other data, was the data validated/verified? If so, please describe how.** The original data was directly observed system logs from Alibaba's production clusters [6]. Our 'bandwidth_gb' feature was indirectly inferred based on typical data sizes for AI tasks and added during preprocessing; this is an estimation. The temporal extension and origin assignment are derived/simulated.
- **What mechanisms or procedures were used to collect the data (e.g., hardware apparatuses or sensors, manual human curation, software programs, software APIs)? How were these mechanisms or procedures validated?** Original data: Alibaba's internal logging infrastructure. Our processing: Python scripts using Pandas/NumPy. Validation of processing is through code review and checking statistical properties of the output against the input.
- **If the dataset is a sample from a larger set, what was the sampling strategy?** The original 2-month trace is a contiguous period from Alibaba's operations. Our temporal extension replicates patterns; it's not a statistical re-sampling of a larger unobserved set.
- **Who was involved in the data collection process (e.g., students, crowdworkers, contractors) and how were they compensated?** Original data: Alibaba employees/researchers. Our processing: Authors of this paper. No external compensation involved for this processing phase.
- **Over what timeframe was the data collected? Does this timeframe match the creation timeframe of the data associated with the instances?** [...] Original trace: July-August 2020. Our processed data synthetically extends this to represent a generic full year, with timestamps adjusted to match the simulation year (e.g., 2023) but retaining the 2020 day-of-week and seasonal patterns.
- **Were any ethical review processes conducted?** [...] For the original Alibaba trace, refer to [6]. Our processing of this public, anonymized data for creating a synthetic benchmark input did not undergo a separate IRB review as no new human subject data was collected.

- **Did you collect the data from the individuals in question directly, or obtain it via third parties or other sources?** Obtained via a public third-party release (Alibaba GitHub).
- **Were the individuals in question notified about the data collection? [...]** N/A for our processing. Refer to original trace for their practices.
- **Did the individuals in question consent to the collection and use of their data? [...]** N/A for our processing. Refer to original trace.
- **If consent was obtained, were the consenting individuals provided with a mechanism to revoke their consent [...]** N/A for our processing.
- **Has an analysis of the potential impact of the dataset and its use on data subjects [...] been conducted?** N/A for our processing of the already anonymized trace. The original trace providers would have considered this.
- **Any other comments?** None.

**Preprocessing/Cleaning/Labeling (Section 3.4 from Gebru et al.):**

- **Was any preprocessing/cleaning/labeling of the data done? [...]** Yes. As described above and in Appendix D.2:
    - Filtering of short tasks.
    - Temporal extension of the 2-month trace to 1 year.
    - Probabilistic assignment of origin data centers.
    - Grouping tasks into 15-minute intervals.
    - Estimation and addition of a 'bandwidth_gb' feature.
    - Runtime normalization of CPU/GPU percentage values to resource units.

    No explicit labeling for ML tasks was performed on the workload itself.
- **Was the "raw" data saved in addition to the preprocessed/cleaned/labeled data? [...]** The original Alibaba trace files (.tar.gz) are publicly available from their repository. Our intermediate processing stages are not typically archived with each DCcluster-Opt release, but the scripts to perform the processing are provided. We distribute the final processed '.pkl' (within a '.zip').
- **Is the software that was used to preprocess/clean/label the data available? [...]** Yes, Python scripts for preprocessing are part of the DCcluster-Opt codebase (primarily in `utils/workload_utils.py` and data analysis scripts in `data/workload/alibaba_2020_dataset/`).
- **Any other comments?** None.

**Uses (Section 3.5 from Gebru et al.):**

- **Has the dataset been used for any tasks already? If so, please provide a description.** This processed version is primarily used within the DCcluster-Opt benchmark to drive the simulations for evaluating the baseline controllers and RL agents presented in this paper.
- **Is there a repository that links to any or all papers or systems that use the dataset? If so, please provide a link or other access point.** Papers using DCcluster-Opt (and thus this processed workload) will cite this benchmark paper. The DCcluster-Opt GitHub repository will be the central point.
- **What (other) tasks could the dataset be used for?** Beyond driving DCcluster-Opt, the processed year-long trace could be useful for other research on long-term resource management, capacity planning simulations for GPU clusters, or developing workload forecasting models for AI tasks, keeping in mind its synthetic extension.
- **Is there anything about the composition of the dataset or the way it was collected and preprocessed/cleaned/labeled that might impact future uses? [...]** The primary impact is that the temporal extension is synthetic, based on patterns from two months. While designed to be realistic, it's not a true year-long trace. The origin assignment is probabilistic. The 'bandwidth_gb' is an estimation. These factors should be considered if using the data for tasks highly sensitive to precise, real-world future forecasting. It's suitable for evaluating adaptive scheduling policies within a simulated year.
- **Are there tasks for which the dataset should not be used?** It should not be used for direct financial forecasting related to Alibaba's operations or for making definitive claims about future specific workload trends at Alibaba beyond what the original 2-month trace

represented. It's a tool for benchmarking scheduling algorithms in a *realistic but simulated* environment.
- **Any other comments?** None.

**Distribution (Section 3.6 from Gebru et al.):**

- **Will the dataset be distributed to third parties outside of the entity [...]** Yes, it is distributed publicly as part of the open-source DCcluster-Opt benchmark via GitHub.
- **How will the dataset will be distributed? Does the dataset have a digital object identifier (DOI)?** Distributed as a `.zip` file (containing the `.pkl`) within the GitHub repository (`https://github.com/HewlettPackard/sustain-cluster`). We will pursue assigning a DOI for the benchmark and its core datasets via a service like Zenodo upon acceptance or public release.
- **When will the dataset be distributed?** It is currently available with the DCcluster-Opt codebase on GitHub.
- **Will the dataset be distributed under a copyright or other intellectual property (IP) license, and/or under applicable terms of use (ToU)? [...]** The DCcluster-Opt codebase, including our processing scripts, is under the MIT License. The underlying Alibaba Cluster Trace 2020 data is subject to its own license terms (typically non-commercial research use with attribution), which users must respect.
- **Have any third parties imposed IP-based or other restrictions on the data associated with the instances? [...]** Yes, the original Alibaba trace terms as mentioned above.
- **Do any export controls or other regulatory restrictions apply to the dataset or to individual instances? [...]** Not to our knowledge for the processed, anonymized data. Users should verify for their own jurisdictions if using the original trace.
- **Any other comments?** None.

**Maintenance (Section 3.7 from Gebru et al.):**

- **Who will be supporting/hosting/maintaining the dataset?** The authors and maintainers of the DCcluster-Opt project (Hewlett Packard Labs).
- **How can the owner/curator/manager of the dataset be contacted?** Via GitHub issues on the DCcluster-Opt repository or through corresponding author contact.
- **Is there an erratum? If so, please provide a link or other access point.** Errata, if any, will be documented in the GitHub repository's issues or release notes.
- **Will the dataset be updated? [...]** The core processed Alibaba trace (based on 2020 data) is unlikely to change unless significant errors are found or a new version of the original trace is released and reprocessed. Updates would be communicated via GitHub releases.
- **If the dataset relates to people, are there applicable limits on the retention of the data [...]** N/A, data is anonymized task metadata.
- **Will older versions of the dataset continue to be supported/hosted/maintained? [...]** Major versions of the processed dataset will be versioned alongside DCcluster-Opt releases. Older versions will remain accessible via Git history.
- **If others want to extend/augment/build on/contribute to the dataset, is there a mechanism for them to do so? [...]** Contributions to preprocessing scripts or suggestions for new workload integrations can be made via GitHub pull requests and issues. Validation would involve code review and statistical comparison by the core maintainers.
- **Any other comments?** Our primary focus for data maintenance will be on the environmental datasets (price, carbon, weather), which we aim to update periodically (annually) as new data becomes available from their respective sources (see Appendix K for overall benchmark maintenance).

### D.10.2   Datasheet: Electricity Prices

**Motivation (Section 3.1 from Gebru et al.):**

- **For what purpose was the dataset created? Was there a specific task in mind? Was there a specific gap that needed to be filled? Please provide a description.** This curated collection of electricity price data was assembled for the DCcluster-Opt benchmark. Its purpose is to provide realistic, time-varying electricity costs for over 20 global regions,

enabling the simulation and evaluation of cost-aware and multi-objective (e.g., cost vs. carbon) scheduling algorithms. The specific task is to allow the simulated data centers to incur actual monetary costs for their energy consumption, creating a dynamic economic signal for scheduling agents. The gap filled is the lack of readily available, standardized, and integrated multi-year price datasets across diverse global energy markets for use in DC sustainability research.

- **Who created the dataset (e.g., which team, research group) and on behalf of which entity (e.g., company, institution, organization)?** The raw data was originally published by various entities (Electricity Maps, GridStatus.io, regional ISOs/energy exchanges). The collection, cleaning, standardization (to UTC, USD/MWh), interpolation to 15-minute intervals, and organization for DCcluster-Opt were performed by the authors of this paper, representing Hewlett Packard Labs.
- **Who funded the creation of the dataset? If there is an associated grant, please provide the name of the grantor and the grant name and number.** The original data sources are generally public or have specific API access terms. The curation and integration effort for DCcluster-Opt were supported by Hewlett Packard Labs.
- **Any other comments?** None.

**Composition (Section 3.2 from Gebru et al.):**

- **What do the instances that comprise the dataset represent? Are there multiple types of instances? Please provide a description.** Each instance (row) in a given processed CSV file represents an hourly electricity price point for a specific geographical region at a specific UTC timestamp. After interpolation, each instance effectively represents a 15-minute price point. The dataset comprises multiple such time series, one for each supported region and year.
- **How many instances are there in total (of each type, if appropriate)?** For each region, there are approximately 8,760 hourly instances per year (more for leap years). After interpolation to 15-minute intervals, this becomes approximately 35,040 instances per region per year. Data is provided for over 20 regions for the years 2020-2024 (where available from sources).
- **Does the dataset contain all possible instances or is it a sample? If the dataset is a sample, then what is the larger set? Is the sample representative? [...]** The dataset aims to be a comprehensive collection of historical spot/wholesale electricity prices for the covered regions and timeframes, as made available by the original sources. It is not a sample in the statistical sense but rather a collection of available historical records. Representativeness is dependent on the coverage and accuracy of the original data providers for their respective markets.
- **What data does each instance consist of? "Raw" data or features? In either case, please provide a description.** Each instance in the standardized CSV files (e.g., `data/electricity_prices/standardized/REGION/YEAR/REGION_electricity_prices_YEAR.csv`) consists of processed features:
  - `Datetime (UTC)`: Timestamp in UTC (datetime object or string).
  - `Price (USD/MWh)`: Electricity price in US dollars per Megawatt-hour (float).
  - (Original raw files might contain additional columns like local time, currency, which are standardized during preprocessing).
- **Is there a label or target associated with each instance? If so, please provide a description.** No. This is time-series input data.
- **Is any information missing from individual instances? If so, please provide a description, explaining why this information is missing [...]** Gaps in the original data sources (e.g., due to API outages or reporting issues from the market operator) may result in missing hourly values. Our preprocessing scripts (`data/electricity_prices/scripts/`) attempt to handle short gaps using forward-fill or interpolation where appropriate. Longer gaps might persist if no reliable data was available. The interpolation to 15-minute intervals fills sub-hourly data points.
- **Are relationships between individual instances made explicit?** Instances are temporally related (time-series). Relationships between prices in different regions are not explicitly encoded but can be inferred by comparing their respective time series.

- **Are there recommended data splits (e.g., training, development/validation, testing)? If so, please provide a description of these splits, explaining the rationale behind them.** Not explicitly defined within the dataset. Users training forecasting models would typically split chronologically (e.g., train on 2020-2022, validate on 2023, test on 2024). For DCcluster-Opt simulations, different years or periods can be chosen for training vs. evaluation of scheduling agents.
- **Are there any errors, sources of noise, or redundancies in the dataset? If so, please provide a description.** The original data from public sources may contain errors or noise. We apply some outlier detection/clipping during preprocessing (e.g., for extremely anomalous price spikes based on IQR). Redundancies are unlikely in the processed hourly data.
- **Is the dataset self-contained, or does it link to or otherwise rely on external resources? [...]** The standardized CSV files provided within DCcluster-Opt are self-contained. The scripts used to generate them rely on external APIs (Electricity Maps, GridStatus, etc.) or access to raw data files downloaded from energy market operators (see `data/electricity_prices/README.md` and `data/electricity_prices/raw/`). a) Guarantees for external APIs: Subject to the terms and availability of those APIs. b) Archival versions: We archive the version of the raw data used for generation where possible. The standardized CSVs are versioned with DCcluster-Opt releases. c) Restrictions: Access to original data APIs may require API keys or be subject to rate limits or terms of use from the providers.
- **Does the dataset contain data that might be considered confidential? [...]** No. It is derived from publicly reported electricity market data.
- **Does the dataset contain data that, if viewed directly, might be offensive, insulting, threatening, or might otherwise cause anxiety? If so, please describe why.** No.
- **Does the dataset identify any subpopulations? [...]** N/A.
- **Is it possible to identify individuals [...] from the dataset? If so, please describe how.** N/A.
- **Does the dataset contain data that might be considered sensitive in any way [...]** N/A.
- **Any other comments?** The quality and granularity of available price data vary by region and original source.

**Collection Process (Section 3.3 from Gebru et al.):**

- **How was the data associated with each instance acquired? Was the data directly observable [...], reported by subjects [...], or indirectly inferred/derived [...]** The data is directly observable, reported by energy market operators, exchanges, or third-party aggregators like Electricity Maps and GridStatus.io.
- **What mechanisms or procedures were used to collect the data (e.g., hardware apparatuses or sensors, manual human curation, software programs, software APIs)? How were these mechanisms or procedures validated?** Data is collected via public APIs (e.g., Electricity Maps, GridStatus.io) using Python scripts, or by downloading historical data files directly from ISO/market operator websites. These sources are generally considered authoritative for their respective markets. Validation involves checking for completeness, consistency with reported market trends, and unit conversions.
- **If the dataset is a sample from a larger set, what was the sampling strategy?** N/A, aims to be a collection of available historical records.
- **Who was involved in the data collection process (e.g., students, crowdworkers, contractors) and how were they compensated?** Data collection and processing scripts were developed by the authors of this paper.
- **Over what timeframe was the data collected? Does this timeframe match the creation timeframe of the data associated with the instances? [...]** Data for years 2020-2024 was collected retrospectively (e.g., in 2023-2024). This timeframe matches the creation timeframe of the data itself (i.e., they are historical prices).
- **Were any ethical review processes conducted? [...]** N/A, as it uses publicly available market data.
- **Did you collect the data from the individuals in question directly, or obtain it via third parties or other sources?** N/A.
- **Were the individuals in question notified about the data collection? [...]** N/A.
- **Did the individuals in question consent to the collection and use of their data? [...]** N/A.

- **If consent was obtained, were the consenting individuals provided with a mechanism to revoke their consent [...]** N/A.
- **Has an analysis of the potential impact of the dataset and its use on data subjects [...] been conducted?** N/A.
- **Any other comments?** None.

**Preprocessing/Cleaning/Labeling (Section 3.4 from Gebru et al.):**

- **Was any preprocessing/cleaning/labeling of the data done? [...]** Yes. Preprocessing steps (implemented in scripts in `data/electricity_prices/scripts/`) include:
    - Parsing various raw data formats (CSV, Excel, API JSON).
    - Converting all timestamps to UTC.
    - Converting all prices to a standard unit (USD/MWh). This involves fetching historical exchange rates for non-USD currencies.
    - Handling missing data points through methods like forward-filling or linear interpolation for short gaps.
    - Applying outlier detection/capping (e.g., based on IQR) to handle extreme, potentially erroneous price spikes.
    - Interpolating hourly data to 15-minute intervals for simulation use.

    No labeling was performed.
- **Was the "raw" data saved in addition to the preprocessed/cleaned/labeled data? [...]** Yes, where feasible, raw downloaded files are stored in `data/electricity_prices/raw/REGION/` to allow for reprocessing or verification.
- **Is the software that was used to preprocess/clean/label the data available? [...]** Yes, Python scripts for data fetching and standardization are provided in `data/electricity_prices/scripts/`.
- **Any other comments?** The goal of preprocessing is to create a consistent, clean, and usable dataset for the benchmark across multiple heterogeneous sources.

**Uses (Section 3.5 from Gebru et al.):**

- **Has the dataset been used for any tasks already? If so, please provide a description.** This curated collection is used within the DCcluster-Opt benchmark to provide the dynamic electricity price signal for each simulated data center, influencing energy cost calculations and providing input for cost-aware scheduling agents.
- **Is there a repository that links to any or all papers or systems that use the dataset? If so, please provide a link or other access point.** Papers using DCcluster-Opt will cite this benchmark paper. The DCcluster-Opt GitHub repository is the central point.
- **What (other) tasks could the dataset be used for?** Beyond DCcluster-Opt, this standardized multi-region electricity price dataset could be valuable for research in energy market analysis, electricity price forecasting, or other simulations requiring realistic energy cost inputs.
- **Is there anything about the composition of the dataset or the way it was collected and preprocessed/cleaned/labeled that might impact future uses? [...]** The interpolation to 15-minute intervals from hourly data is an approximation. Users requiring true sub-hourly price data would need to consult original sources if available at finer granularity. Outlier capping might remove legitimate extreme price events, though it's intended to improve stability for general RL training. The accuracy is dependent on the original data providers.
- **Are there tasks for which the dataset should not be used?** It should not be used for high-frequency trading or applications requiring legally binding price data, as it's processed historical data intended for research simulation. Always refer to original sources for official market data.
- **Any other comments?** None.

**Distribution (Section 3.6 from Gebru et al.):**

- **Will the dataset be distributed to third parties outside of the entity [...]** Yes, as part of the open-source DCcluster-Opt benchmark via GitHub.

- **How will the dataset will be distributed? Does the dataset have a digital object identifier (DOI)?** Distributed as CSV files within the DCcluster-Opt GitHub repository (`https://github.com/HewlettPackard/sustain-cluster`) under `data/electricity_prices/standardized/`. We will pursue a DOI for the overall benchmark and its core datasets.
- **When will the dataset be distributed?** Currently available with the DCcluster-Opt codebase.
- **Will the dataset be distributed under a copyright or other intellectual property (IP) license, and/or under applicable terms of use (ToU)? [...]** The DCcluster-Opt codebase and our processing scripts are under the MIT License. The underlying price data is sourced from public entities and APIs, subject to their respective terms (generally allowing public access and research use). Users are responsible for adhering to the terms of the original data sources.
- **Have any third parties imposed IP-based or other restrictions on the data associated with the instances? [...]** The original data providers (ISOs, Electricity Maps, GridStatus) have their own terms of service for API usage or data access. Our processed data is for research benchmark purposes.
- **Do any export controls or other regulatory restrictions apply to the dataset or to individual instances? [...]** Not to our knowledge for this publicly derived market data.
- **Any other comments?** None.

**Maintenance (Section 3.7 from Gebru et al.):**

- **Who will be supporting/hosting/maintaining the dataset?** The authors and maintainers of the DCcluster-Opt project (Hewlett Packard Labs).
- **How can the owner/curator/manager of the dataset be contacted?** Via GitHub issues on the DCcluster-Opt repository or corresponding author.
- **Is there an erratum? If so, please provide a link or other access point.** Errata will be documented in GitHub issues or release notes.
- **Will the dataset be updated? [...]** We aim to update the price data periodically (annually) by fetching new historical data from the sources and running the standardization scripts. Updates will be communicated via GitHub releases.
- **If the dataset relates to people, are there applicable limits on the retention of the data [...]** N/A.
- **Will older versions of the dataset continue to be supported/hosted/maintained? [...]** Major versions of the standardized price datasets will be versioned with DCcluster-Opt releases. Older versions will remain accessible via Git history.
- **If others want to extend/augment/build on/contribute to the dataset, is there a mechanism for them to do so? [...]** Contributions to data processing scripts or suggestions for new regions/data sources can be made via GitHub pull requests and issues. Validation would involve code review and data consistency checks by core maintainers.
- **Any other comments?** See overall benchmark maintenance plan in Appendix K.

### D.10.3 Datasheet: Grid Carbon Intensity

**Motivation (Section 3.1 from Gebru et al.):**

- **For what purpose was the dataset created? Was there a specific task in mind? Was there a specific gap that needed to be filled? Please provide a description.** This curated collection of grid carbon intensity data was assembled for the DCcluster-Opt benchmark. Its purpose is to provide realistic, time-varying carbon intensity values (gCO$_2$eq/kWh) for over 20 global regions, enabling the simulation and evaluation of carbon-aware and multi-objective scheduling algorithms. The specific task is to allow the simulated data centers to calculate the carbon footprint of their energy consumption (for IT operations, cooling, and data transmission), creating a dynamic environmental signal for scheduling agents. The gap filled is the need for standardized, integrated, multi-year carbon intensity datasets across diverse global grids for use in DC sustainability research.
- **Who created the dataset (e.g., which team, research group) and on behalf of which entity (e.g., company, institution, organization)?** The raw data is primarily sourced from Electricity Maps [7]. The collection, any necessary cleaning/standardization (to UTC,

$gCO_2eq/kWh$), interpolation to 15-minute intervals, and organization for DCcluster-Opt were performed by the authors of this paper, representing Hewlett Packard Labs.

- **Who funded the creation of the dataset? If there is an associated grant, please provide the name of the grantor and the grant name and number.** Electricity Maps data is available via a public API, often with specific terms for research use. The curation and integration effort for DCcluster-Opt were supported by Hewlett Packard Labs.
- **Any other comments?** The accuracy and granularity of carbon intensity data can vary by region based on the methodologies used by the original provider (Electricity Maps).

**Composition (Section 3.2 from Gebru et al.):**

- **What do the instances that comprise the dataset represent? Are there multiple types of instances? Please provide a description.** Each instance (row) in a given processed CSV file represents an hourly grid carbon intensity data point for a specific geographical region at a specific UTC timestamp. After interpolation, each instance effectively represents a 15-minute carbon intensity value. The dataset comprises multiple such time series, one for each supported region and year.
- **How many instances are there in total (of each type, if appropriate)?** For each region, there are approximately 8,760 hourly instances per year (more for leap years). After interpolation to 15-minute intervals, this becomes approximately 35,040 instances per region per year. Data is provided for over 20 regions for the years 2021-2024 (where available from the source).
- **Does the dataset contain all possible instances or is it a sample? If the dataset is a sample, then what is the larger set? Is the sample representative? [...]** The dataset aims to be a comprehensive collection of historical grid carbon intensities for the covered regions and timeframes, as provided by Electricity Maps. It is not a sample in a statistical sense but a collection of their reported historical data. Representativeness is dependent on Electricity Maps' data collection and modeling methodology for each grid.
- **What data does each instance consist of? "Raw" data or features? In either case, please provide a description.** Each instance in the standardized CSV files (e.g., `data/carbon_intensity/REGION/YEAR/REGION_YEAR_hourly.csv`) consists of processed features:

  - `Datetime (UTC)`: Timestamp in UTC (datetime object or string).
  - `Carbon Intensity gCOeq/kWh (direct)`: Grid carbon intensity in grams of $CO_2$ equivalent per kilowatt-hour (float). (The column name might vary slightly, ensure it matches your files).

  Original API responses from Electricity Maps might contain additional metadata.
- **Is there a label or target associated with each instance? If so, please provide a description.** No. This is time-series input data.
- **Is any information missing from individual instances? If so, please provide a description, explaining why this information is missing [...]** Gaps in the original data source (e.g., due to API availability or reporting issues from Electricity Maps for certain periods/regions) may result in missing hourly values. Preprocessing scripts (e.g., `data/carbon_intensity/analyze_carbon_intensity_data.py` or similar) may attempt to handle short gaps using forward-fill or interpolation. Longer gaps might persist. The interpolation to 15-minute intervals fills sub-hourly data points.
- **Are relationships between individual instances made explicit?** Instances are temporally related (time-series).
- **Are there recommended data splits (e.g., training, development/validation, testing)? [...]** Not explicitly defined. Users training forecasting models would typically split chronologically. For DCcluster-Opt simulations, different years/periods can be used for training vs. evaluation of scheduling agents.
- **Are there any errors, sources of noise, or redundancies in the dataset? [...]** The original data from Electricity Maps is subject to their modeling accuracy and data availability. Small amounts of noise or estimation errors may be present. Redundancies are unlikely in the processed hourly data.
- **Is the dataset self-contained, or does it link to or otherwise rely on external resources? [...]** The standardized CSV files provided within DCcluster-Opt are self-contained. The scripts used to generate them rely on the Electricity Maps API. a) Guarantees for external

APIs: Subject to Electricity Maps' API terms and availability. b) Archival versions: We archive the version of the raw data fetched where feasible. Standardized CSVs are versioned with DCcluster-Opt releases. c) Restrictions: Access to the Electricity Maps API may require an API key and is subject to their terms of use.

- **Does the dataset contain data that might be considered confidential? [...]** No.
- **Does the dataset contain data that, if viewed directly, might be offensive, insulting, threatening, or might otherwise cause anxiety? [...]** No.
- **Does the dataset identify any subpopulations? [...]** N/A.
- **Is it possible to identify individuals [...] from the dataset? If so, please describe how.** N/A.
- **Does the dataset contain data that might be considered sensitive in any way [...]** N/A.
- **Any other comments?** The carbon intensity data reflects the average mix of generation sources on a given grid at a given time and is a crucial factor for "follow the green" scheduling.

**Collection Process (Section 3.3 from Gebru et al.):**

- **How was the data associated with each instance acquired? [...]** Directly observable data reported/modeled by Electricity Maps based on real-time grid generation data.
- **What mechanisms or procedures were used to collect the data [...]** Data collected via the public API provided by Electricity Maps using Python scripts. Electricity Maps employs its own methodologies for data collection and validation from various grid operators and sources.
- **If the dataset is a sample from a larger set, what was the sampling strategy?** N/A.
- **Who was involved in the data collection process [...]** Data fetching and processing scripts developed by the authors of this paper.
- **Over what timeframe was the data collected? [...]** Data for years 2021-2024 collected retrospectively. This matches the creation timeframe of the data (historical records).
- **Were any ethical review processes conducted? [...]** N/A, uses publicly available environmental data.
- **Remaining questions regarding individuals:** N/A.
- **Any other comments?** None.

**Preprocessing/Cleaning/Labeling (Section 3.4 from Gebru et al.):**

- **Was any preprocessing/cleaning/labeling of the data done? [...]** Yes. Preprocessing steps include:
    - Fetching data via the Electricity Maps API for specified regions and timeframes.
    - Parsing JSON responses.
    - Aligning timestamps to UTC.
    - Handling missing data points (e.g., via forward-fill or interpolation for short gaps).
    - Saving to standardized hourly CSV format per region/year.
    - Interpolating hourly data to 15-minute intervals for simulation use by the `CI_Manager`.

  No labeling was performed.
- **Was the "raw" data saved in addition to the preprocessed/cleaned/labeled data? [...]** The hourly standardized CSVs serve as our "cleaned raw" data. The very original JSON responses from the API are not typically archived long-term, but the fetching scripts can be re-run.
- **Is the software that was used to preprocess/clean/label the data available? [...]** Yes, Python scripts for data fetching and standardization are part of the DCcluster-Opt codebase.
- **Any other comments?** The primary goal is to provide a consistent time-series of carbon intensity values for each simulated location.

**Uses (Section 3.5 from Gebru et al.):**

- **Has the dataset been used for any tasks already? [...]** Used within DCcluster-Opt to calculate the carbon emissions associated with energy consumption (IT, cooling, transmission), providing a dynamic signal for carbon-aware scheduling agents.

- **Is there a repository that links to any or all papers or systems that use the dataset?** [...] Papers using DCcluster-Opt will cite this benchmark paper. The DCcluster-Opt GitHub repository is the central point.
- **What (other) tasks could the dataset be used for?** This standardized multi-region carbon intensity dataset could be useful for research in carbon footprint analysis of distributed systems, carbon intensity forecasting, or other simulations requiring realistic grid emissions data.
- **Is there anything about the composition of the dataset or the way it was collected and preprocessed/cleaned/labeled that might impact future uses?** [...] The accuracy is dependent on the original Electricity Maps data and their models. Interpolation to 15-minute intervals is an approximation of sub-hourly variations.
- **Are there tasks for which the dataset should not be used?** Should not be used for applications requiring certified or legally binding emissions reporting where official national/regional inventories are mandated. It's intended for research simulation.
- **Any other comments?** None.

**Distribution (Section 3.6 from Gebru et al.):**

- **Will the dataset be distributed to third parties [...]** Yes, publicly via GitHub as part of DCcluster-Opt.
- **How will the dataset will be distributed? [...]** As CSV files within the DCcluster-Opt GitHub repository (`https://github.com/HewlettPackard/sustain-cluster`) under `data/carbon_intensity/`. A DOI for the benchmark will be pursued.
- **When will the dataset be distributed?** Currently available.
- **Will the dataset be distributed under a copyright or other IP license, and/or under applicable terms of use (ToU)? [...]** DCcluster-Opt codebase (including processing scripts) is MIT Licensed. The underlying data from Electricity Maps is subject to their terms (often allowing research use with attribution). Users must respect these original terms.
- **Have any third parties imposed IP-based or other restrictions [...]** See Electricity Maps API terms.
- **Do any export controls or other regulatory restrictions apply [...]** Not to our knowledge.
- **Any other comments?** None.

**Maintenance (Section 3.7 from Gebru et al.):**

- **Who will be supporting/hosting/maintaining the dataset?** The DCcluster-Opt authors (Hewlett Packard Labs).
- **How can the owner/curator/manager [...] be contacted?** Via GitHub issues or corresponding author.
- **Is there an erratum? [...]** Documented in GitHub issues/releases.
- **Will the dataset be updated? [...]** We aim to update periodically (annually) by fetching new historical data from Electricity Maps and re-running standardization. Updates via GitHub releases.
- **If the dataset relates to people [...]** N/A.
- **Will older versions [...] continue to be supported/hosted/maintained? [...]** Via Git history.
- **If others want to extend/augment/build on/contribute [...]** Via GitHub pull requests/issues. Contributions reviewed by core maintainers.
- **Any other comments?** See overall benchmark maintenance plan in Appendix K.

### D.10.4 Datasheet: Weather Data

**Motivation (Section 3.1 from Gebru et al.):**

- **For what purpose was the dataset created? Was there a specific task in mind? Was there a specific gap that needed to be filled? Please provide a description.** This collection of historical weather data was assembled for the DCcluster-Opt benchmark. Its primary purpose is to provide realistic, time-varying ambient temperature data for the selected global data center locations. This data is crucial for the physics-informed HVAC (Heating, Ventilation, and Air Conditioning) model within DCcluster-Opt, as outdoor temperature

significantly impacts cooling system efficiency and energy consumption. The gap filled is the need for easily accessible, standardized weather data integrated with a DC sustainability benchmark.

- **Who created the dataset (e.g., which team, research group) and on behalf of which entity (e.g., company, institution, organization)?** The raw weather data is sourced from the Open-Meteo API [9]. The collection (via API calls), any necessary processing (e.g., interpolation to 15-minute intervals), and organization for DCcluster-Opt were performed by the authors of this paper, representing Hewlett Packard Labs.
- **Who funded the creation of the dataset? If there is an associated grant, please provide the name of the grantor and the grant name and number.** Open-Meteo provides free access for non-commercial use. The curation and integration effort for DCcluster-Opt were supported by Hewlett Packard Labs.
- **Any other comments?** The accuracy of the weather data is dependent on the meteorological models and sources used by Open-Meteo.

**Composition (Section 3.2 from Gebru et al.):**

- **What do the instances that comprise the dataset represent? Are there multiple types of instances? Please provide a description.** Each instance in the processed JSON files (one file per region per year) represents a set of hourly meteorological readings for a specific geographical location corresponding to a simulated data center. After interpolation, these effectively provide 15-minute data points for key variables like temperature.
- **How many instances are there in total (of each type, if appropriate)?** For each region, there are approximately 8,760 hourly instances (sets of readings) per year. After interpolation to 15-minute intervals, this becomes approximately 35,040 instances per region per year. Data is provided for over 20 regions for the years 2021-2024 (where available from Open-Meteo).
- **Does the dataset contain all possible instances or is it a sample? [...]** The dataset aims to be a comprehensive collection of historical weather parameters for the covered locations and timeframes, as provided by Open-Meteo, which aggregates data from various national weather services and numerical weather prediction models.
- **What data does each instance consist of? "Raw" data or features? In either case, please provide a description.** Each processed JSON file (e.g., `data/weather/REGION/YEAR.json`) contains arrays under an "hourly" key. The primary features used by DCcluster-Opt are:
  - `time`: List of UTC timestamps (datetime strings).
  - `temperature_2m`: List of ambient air temperatures (°C) at 2 meters above ground (floats).
  - (Potentially other variables like `relative_humidity_2m`, `wetbulb_temperature` if your model explicitly uses them for more advanced cooling/water usage calculations, though your current model might derive wet-bulb).
- **Is there a label or target associated with each instance? [...]** No. This is time-series input data.
- **Is any information missing from individual instances? [...]** Gaps are generally unlikely from Open-Meteo as it uses model reanalysis data, but any missing values would be handled by interpolation during preprocessing or by the `Weather_Manager`.
- **Are relationships between individual instances made explicit?** Instances are temporally related.
- **Are there recommended data splits [...]** Not explicitly. Users can select different years/periods for training vs. evaluation.
- **Are there any errors, sources of noise, or redundancies in the dataset? [...]** Data is subject to the accuracy of the underlying weather models and observations used by Open-Meteo.
- **Is the dataset self-contained, or does it link to or otherwise rely on external resources? [...]** The processed JSON files within DCcluster-Opt are self-contained. The script used to generate them (`data/weather/extract_weather_data.py`) relies on the external Open-Meteo API. a) Guarantees for Open-Meteo API: Subject to Open-Meteo's terms and availability. b) Archival versions: The downloaded JSONs are archived with DCcluster-Opt

releases. c) Restrictions: Open-Meteo is generally free for non-commercial use; refer to their terms.

- **Does the dataset contain data that might be considered confidential? [...]** No.
- **Does the dataset contain data that, if viewed directly, might be offensive, insulting, threatening, or might otherwise cause anxiety? [...]** No.
- **Questions regarding individuals/subpopulations/sensitive data:** N/A.
- **Any other comments?** Wet-bulb temperature, crucial for some cooling tower models and water usage calculations, is either fetched directly if available from Open-Meteo or estimated using temperature and relative humidity via psychrometric libraries (e.g., PsychroLib, as used in your `Weather_Manager`).

**Collection Process (Section 3.3 from Gebru et al.):**

- **How was the data associated with each instance acquired? [...]** Directly observable data provided by the Open-Meteo API, which aggregates from numerical weather models and weather station data.
- **What mechanisms or procedures were used to collect the data [...]** Data collected via the Open-Meteo HTTP API using Python scripts (e.g., `data/weather/extract_weather_data.py`). Open-Meteo validates its data sources.
- **If the dataset is a sample [...]** N/A.
- **Who was involved in the data collection process [...]** Data fetching and processing scripts developed by the authors of this paper.
- **Over what timeframe was the data collected? [...]** Data for years 2021-2024 typically collected retrospectively. Matches creation timeframe.
- **Were any ethical review processes conducted? [...]** N/A, uses publicly available environmental data.
- **Remaining questions regarding individuals:** N/A.
- **Any other comments?** None.

**Preprocessing/Cleaning/Labeling (Section 3.4 from Gebru et al.):**

- **Was any preprocessing/cleaning/labeling of the data done? [...]** Yes. Preprocessing steps include:
    - Fetching hourly data (temperature, relative humidity, etc.) for specified latitude/longitude (corresponding to DC locations) and timeframes via the Open-Meteo API.
    - Saving raw API responses as yearly JSON files per location.
    - The `Weather_Manager` at runtime reads these JSONs, applies timezone shifts if necessary, and interpolates hourly data to 15-minute intervals.
    - It may also calculate derived values like wet-bulb temperature using PsychroLib if not directly available.
    - Optional coherent noise can be added for training variability.

    No labeling was performed.
- **Was the "raw" data saved in addition to the preprocessed/cleaned/labeled data? [...]** Yes, the downloaded hourly JSON files serve as the "raw" data for the `Weather_Manager`.
- **Is the software that was used to preprocess/clean/label the data available? [...]** Yes, the API fetching script (`extract_weather_data.py`) and the runtime processing logic within `utils/managers.py::Weather_Manager` are part of the DCcluster-Opt codebase.
- **Any other comments?** The primary goal is to provide consistent time-series of ambient temperature (and potentially wet-bulb) as input to the DC's HVAC model.

**Uses (Section 3.5 from Gebru et al.):**

- **Has the dataset been used for any tasks already? [...]** Used within DCcluster-Opt to provide dynamic ambient temperature conditions for each simulated data center, which directly impacts the performance and energy consumption of the HVAC model.
- **Is there a repository that links to any or all papers or systems that use the dataset? [...]** Papers using DCcluster-Opt will cite this benchmark paper. The DCcluster-Opt GitHub repository is the central point.

- **What (other) tasks could the dataset be used for?** Could be used for other types of simulations requiring historical weather data for specific global locations, or for research into weather forecasting effects on DC operations.
- **Is there anything about the composition of the dataset or the way it was collected and preprocessed/cleaned/labeled that might impact future uses? [...]** The accuracy is dependent on Open-Meteo's underlying data sources and models. Interpolation to 15-minute intervals is an approximation. The optional addition of coherent noise during training is a synthetic augmentation.
- **Are there tasks for which the dataset should not be used?** Should not be used for applications requiring certified meteorological data for legal or safety-critical purposes where official national weather service data is mandated.
- **Any other comments?** None.

**Distribution (Section 3.6 from Gebru et al.):**

- **Will the dataset be distributed to third parties [...]** Yes, publicly via GitHub as part of DCcluster-Opt.
- **How will the dataset will be distributed? [...]** As JSON files within the DCcluster-Opt GitHub repository (`https://github.com/HewlettPackard/sustain-cluster`) under `data/weather/`. A DOI for the benchmark will be pursued.
- **When will the dataset be distributed?** Currently available.
- **Will the dataset be distributed under a copyright or other IP license, and/or under applicable terms of use (ToU)? [...]** DCcluster-Opt codebase (including processing scripts) is MIT Licensed. Open-Meteo data is typically available under permissive licenses (e.g., CC BY 4.0) for non-commercial use, requiring attribution. Users must respect these original terms.
- **Have any third parties imposed IP-based or other restrictions [...]** See Open-Meteo terms.
- **Do any export controls or other regulatory restrictions apply [...]** Not to our knowledge.
- **Any other comments?** None.

**Maintenance (Section 3.7 from Gebru et al.):**

- **Who will be supporting/hosting/maintaining the dataset?** The DCcluster-Opt authors (Hewlett Packard Labs).
- **How can the owner/curator/manager [...] be contacted?** Via GitHub issues or corresponding author.
- **Is there an erratum? [...]** Documented in GitHub issues/releases.
- **Will the dataset be updated? [...]** We aim to update periodically (annually) by fetching new historical data from Open-Meteo. Updates via GitHub releases.
- **If the dataset relates to people [...]** N/A.
- **Will older versions [...] continue to be supported/hosted/maintained? [...]** Via Git history.
- **If others want to extend/augment/build on/contribute [...]** Via GitHub pull requests/issues.
- **Any other comments?** See overall benchmark maintenance plan in Appendix K.

### D.10.5 Datasheet: Cloud Provider Transmission Cost Matrices

**Motivation (Section 3.1 from Gebru et al.):**

- **For what purpose was the dataset created? Was there a specific task in mind? Was there a specific gap that needed to be filled? Please provide a description.** These datasets were created to provide realistic monetary costs for inter-data center data transmission within the DCcluster-Opt benchmark. The specific task is to allow the simulation to calculate the economic impact of routing tasks to remote data centers. The gap filled is the need for easily usable, structured matrices of cloud provider transmission costs for integrated sustainability and economic analysis in scheduling research.
- **Who created the dataset (e.g., which team, research group) and on behalf of which entity (e.g., company, institution, organization)?** The raw pricing data is published by

respective cloud providers (AWS, GCP, Azure). The compilation into structured CSV matrices for DCcluster-Opt was performed by the authors of this paper, representing Hewlett Packard Labs.

- **Who funded the creation of the dataset? [...]** The original pricing data is publicly available from cloud providers. The curation and compilation effort for DCcluster-Opt were supported by Hewlett Packard Labs.
- **Any other comments?** None.

**Composition (Section 3.2 from Gebru et al.):**

- **What do the instances that comprise the dataset represent? [...]** Each dataset is a CSV file representing a cost matrix for a specific cloud provider (e.g., `aws_transmission_cost_matrix.csv`). Each cell $(i, j)$ in the matrix represents the cost in USD to transfer 1 GB of data from an origin cloud region $i$ (row header) to a destination cloud region $j$ (column header).
- **How many instances are there in total (of each type, if appropriate)?** There are typically 3 main matrices (AWS, GCP, Azure) plus an optional `custom_transmission_cost_matrix.csv`. The number of rows/columns in each matrix depends on the number of distinct cloud regions for which pricing was compiled for that provider (e.g., AWS might have 20-30 regions, so a 20x20 matrix).
- **Does the dataset contain all possible instances or is it a sample? [...]** They aim to cover the major, publicly listed inter-region data transfer costs for the selected cloud providers at the time of compilation. Pricing tiers (e.g., costs decreasing with volume) are generally simplified to a representative per-GB rate, often the initial tier or an estimated average. It is a snapshot and simplification of complex, potentially tiered pricing.
- **What data does each instance consist of? "Raw" data or features? [...]** Each cell contains a single floating-point number representing the cost in USD per GB. Row and column headers are strings representing cloud provider region names (e.g., "us-east-1", "europe-west3").
- **Is there a label or target associated with each instance? [...]** No.
- **Is any information missing from individual instances? [...]** Some region-pairs might have no direct pricing (e.g., if data must transit through another region); these might be represented as very high costs or require inference. Costs for new regions might not be immediately reflected.
- **Are relationships between individual instances made explicit?** The matrix structure defines the relationship: cost from origin region (row) to destination region (column).
- **Are there recommended data splits [...]** N/A.
- **Are there any errors, sources of noise, or redundancies in the dataset? [...]** Pricing data is subject to change by cloud providers. These matrices represent a snapshot at the time of compilation. Errors could arise from manual transcription or interpretation of complex pricing pages. We strive for accuracy based on published rates.
- **Is the dataset self-contained, or does it link to or otherwise rely on external resources? [...]** The CSV files provided in `data/network_cost/` are self-contained. Their creation relied on information from the cloud provider websites (URLs provided in Appendix E). a) Guarantees: Cloud provider pricing can change without notice. b) Archival versions: The CSVs are versioned with DCcluster-Opt. c) Restrictions: None for using the compiled data within the benchmark, but direct use of provider services is subject to their terms.
- **Confidential/Offensive/Sensitive Data/Individuals/Subpopulations:** N/A.
- **Any other comments?** These matrices are crucial for modeling the economic aspect of geo-distributed scheduling.

**Collection Process (Section 3.3 from Gebru et al.):**

- **How was the data associated with each instance acquired? [...]** Directly observable pricing information published on the official websites of AWS, GCP, and Azure.
- **What mechanisms or procedures were used to collect the data [...]** Manual collection and transcription of inter-region data transfer pricing into a matrix format (CSV). Validation involved cross-checking rates where possible.
- **If the dataset is a sample [...]** N/A, it's a compilation of published rates.
- **Who was involved in the data collection process [...]** Authors of this paper.

- **Over what timeframe was the data collected? [...]** Compiled based on pricing information available as of early 2025.
- **Were any ethical review processes conducted? [...]** N/A, uses public commercial pricing data.
- **Remaining questions regarding individuals:** N/A.
- **Any other comments?** None.

**Preprocessing/Cleaning/Labeling (Section 3.4 from Gebru et al.):**

- **Was any preprocessing/cleaning/labeling of the data done? [...]** Yes. Preprocessing involved:
  - Identifying relevant inter-region data transfer/egress costs from complex pricing pages.
  - Standardizing to USD per GB.
  - Structuring the data into origin-destination matrices.
  - Handling cases where direct pricing between two regions isn't listed (e.g., may require estimating based on multi-hop or using a high default).

  No labeling was performed.
- **Was the "raw" data saved [...]** The "raw" data is the public information on the cloud provider websites.
- **Is the software that was used to preprocess/clean/label the data available? [...]** N/A (manual compilation into CSVs).
- **Any other comments?** None.

**Uses (Section 3.5 from Gebru et al.):**

- **Has the dataset been used for any tasks already? [...]** Used within DCcluster-Opt by the `DatacenterClusterManager` (via `transmission_cost_loader.py`) to calculate the monetary cost of data transfers, which can be part of the agent's reward function and is reported as an evaluation metric.
- **Is there a repository that links to any or all papers or systems that use the dataset? [...]** The DCcluster-Opt GitHub repository and papers citing this benchmark.
- **What (other) tasks could the dataset be used for?** Could be used in other network-aware cloud simulators or for research on cloud networking costs, keeping in mind it's a snapshot.
- **Is there anything about the composition [...] that might impact future uses? [...]** Cloud provider pricing is dynamic and changes over time. This dataset represents pricing at the time of its compilation. Users should be aware that current real-world prices may differ. The simplification to a single per-GB rate might not capture all volume-based tiering.
- **Are there tasks for which the dataset should not be used?** Should not be used for precise, up-to-the-minute financial planning for actual cloud usage without first verifying against current official cloud provider pricing.
- **Any other comments?** None.

**Distribution (Section 3.6 from Gebru et al.):**

- **Will the dataset be distributed to third parties [...]** Yes, via the DCcluster-Opt GitHub repository.
- **How will the dataset will be distributed? [...]** As CSV files in `data/network_cost/`. A DOI for DCcluster-Opt will cover its datasets.
- **When will the dataset be distributed?** Currently available.
- **Will the dataset be distributed under a copyright or other IP license [...]** The compiled matrices are part of DCcluster-Opt (MIT License). The underlying pricing data is publicly available from the respective cloud providers.
- **Have any third parties imposed IP-based or other restrictions [...]** The cloud providers own their pricing information. This compilation is for research benchmark use.
- **Do any export controls or other regulatory restrictions apply [...]** Not to our knowledge.
- **Any other comments?** None.

**Maintenance (Section 3.7 from Gebru et al.):**

- **Who will be supporting/hosting/maintaining the dataset?** The DCcluster-Opt authors (Hewlett Packard Labs).
- **How can the owner/curator/manager [...] be contacted?** GitHub issues.
- **Is there an erratum? [...]** Via GitHub issues/releases.
- **Will the dataset be updated? [...]** Periodically (e.g., annually or biennially), we may review and update these matrices based on significant changes in public cloud provider pricing. Updates via GitHub releases.
- **If the dataset relates to people [...]** N/A.
- **Will older versions [...] continue to be supported/hosted/maintained? [...]** Via Git history.
- **If others want to extend/augment/build on/contribute [...]** Via GitHub pull requests/issues for the matrices or new provider data.
- **Any other comments?** See overall benchmark maintenance plan in Appendix K.

### D.10.6 Datasheet: Transmission Delay Parameters (Empirical)

**Motivation (Section 3.1 from Gebru et al.):**

- **For what purpose was the dataset created? Was there a specific task in mind? Was there a specific gap that needed to be filled? Please provide a description.** This set of parameters (mean TCP throughput and Round-Trip Time) was curated for the DCcluster-Opt benchmark to model realistic network latency for inter-data center task transfers. The specific task is to calculate the transmission delay component in the simulation, affecting task arrival times at remote DCs. The gap filled is the need for easily usable, empirically grounded network performance parameters for geo-distributed DC simulations, beyond simple fixed latency assumptions.
- **Who created the dataset (e.g., which team, research group) and on behalf of which entity (e.g., company, institution, organization)?** The original empirical measurements were conducted and published by Persico et al. [16]. The extraction of specific values and their integration into the DCcluster-Opt codebase (`data/network_cost/network_delay.py`) were performed by the authors of this paper, representing Hewlett Packard Labs.
- **Who funded the creation of the dataset? [...]** The original research by Persico et al. had its own funding. The integration into DCcluster-Opt was supported by Hewlett Packard Labs.
- **Any other comments?** These parameters represent average performance between broad geographical macro-clusters.

**Composition (Section 3.2 from Gebru et al.):**

- **What do the instances that comprise the dataset represent? [...]** The "dataset" consists of a small set of numerical parameters: mean TCP throughput (in Mbps) and mean RTT (in ms) between pairs of four geographical macro-clusters (US, EU, AP, SA), separately for AWS and Azure networks as reported in the source paper. These are hardcoded as dictionaries in `network_delay.py`.
- **How many instances are there in total (of each type, if appropriate)?** For each provider (AWS, Azure), there are $4 \times 3 = 12$ inter-macro-cluster throughput values and 12 RTT values (e.g., EU-US, EU-AP, EU-SA, etc.).
- **Does the dataset contain all possible instances or is it a sample? [...]** It's an extraction of specific reported mean values from the study by Persico et al. [16], which itself was based on measurements over a specific period.
- **What data does each instance consist of? "Raw" data or features? [...]** Each instance is a pair of floating-point numbers: (Throughput_Mbps, RTT_ms) for a given origin macro-cluster, destination macro-cluster, and cloud provider.
- **Is there a label or target associated with each instance? [...]** No.
- **Is any information missing from individual instances? [...]** The source paper provides mean values; variances or full distributions of these measurements are not used in our current model. Data for other cloud providers (besides AWS/Azure) or other inter-cluster links (e.g., to/from Africa if modeled as a separate macro-cluster) are not present in the original source and would require new data or estimations.

- **Are relationships between individual instances made explicit?** The relationship is defined by the origin-destination macro-cluster pair and the provider.
- **Are there recommended data splits [...]** N/A.
- **Are there any errors, sources of noise, or redundancies in the dataset? [...]** These are reported mean values from an empirical study, so they represent an average and actual network performance can vary significantly. Potential for transcription errors when extracting from the paper.
- **Is the dataset self-contained, or does it link to or otherwise rely on external resources?** **[...]** The parameters, once extracted and coded into `network_delay.py`, are self-contained within the DCcluster-Opt codebase. The ultimate source is the academic publication by Persico et al. [16].
- **Confidential/Offensive/Sensitive Data/Individuals/Subpopulations:** N/A.
- **Any other comments?** These parameters are used in the transmission delay formula detailed in Appendix E.

**Collection Process (Section 3.3 from Gebru et al.):**

- **How was the data associated with each instance acquired? [...]** Extracted directly from figures/tables in the published paper by Persico et al. [16].
- **What mechanisms or procedures were used to collect the data [...]** Manual extraction from the academic publication. The original paper details their measurement methodology.
- **If the dataset is a sample [...]** N/A (it's a specific set of reported values).
- **Who was involved in the data collection process [...]** Authors of this paper for extraction.
- **Over what timeframe was the data collected? [...]** The Persico et al. study was published in 2016; their measurements would predate that. Our extraction occurred during DCcluster-Opt development.
- **Were any ethical review processes conducted? [...]** N/A for data extraction.
- **Remaining questions regarding individuals:** N/A.
- **Any other comments?** None.

**Preprocessing/Cleaning/Labeling (Section 3.4 from Gebru et al.):**

- **Was any preprocessing/cleaning/labeling of the data done? [...]** Primarily data entry of the reported mean values into Python dictionaries within `network_delay.py`. Units (Mbps, ms) were maintained as per the source. No other significant preprocessing by us.
- **Was the "raw" data saved [...]** The "raw" data is the Persico et al. paper itself.
- **Is the software that was used to preprocess/clean/label the data available? [...]** The relevant Python dictionaries are in `data/network_cost/network_delay.py`.
- **Any other comments?** None.

**Uses (Section 3.5 from Gebru et al.):**

- **Has the dataset been used for any tasks already? [...]** Used within DCcluster-Opt's `network_delay.py` module to calculate the serialization and propagation delay components for inter-DC task transfers.
- **Is there a repository that links to any or all papers or systems that use the dataset? [...]** The DCcluster-Opt GitHub repository. Papers citing Persico et al. [16] use the original data.
- **What (other) tasks could the dataset be used for?** Could inform other network simulation models requiring approximate inter-continental cloud network performance parameters, with the caveat of its age.
- **Is there anything about the composition [...] that might impact future uses? [...]** The data is from circa 2016. Actual inter-cloud network performance has likely evolved. These values provide a consistent, empirically grounded (for their time) baseline. Using more current empirical data would be a future enhancement.
- **Are there tasks for which the dataset should not be used?** Should not be used for precise prediction of current-day network performance for specific cloud links without acknowledging its age.
- **Any other comments?** None.

**Distribution (Section 3.6 from Gebru et al.):**

- **Will the dataset be distributed to third parties [...]** Yes, as embedded parameters within the DCcluster-Opt codebase.
- **How will the dataset will be distributed?** **[...]** As Python dictionaries in `data/network_cost/network_delay.py`.
- **When will the dataset be distributed?** Currently available with the codebase.
- **Will the dataset be distributed under a copyright or other IP license [...]** The parameters themselves are facts extracted from a public academic paper. Standard academic citation practices apply to the source.
- **Have any third parties imposed IP-based or other restrictions [...]** N/A.
- **Do any export controls or other regulatory restrictions apply [...]** Not to our knowledge.
- **Any other comments?** None.

**Maintenance (Section 3.7 from Gebru et al.):**

- **Who will be supporting/hosting/maintaining the dataset?** The DCcluster-Opt authors (Hewlett Packard Labs).
- **How can the owner/curator/manager [...] be contacted?** GitHub issues.
- **Is there an erratum? [...]** Via GitHub issues/releases.
- **Will the dataset be updated? [...]** These parameters would only be updated if newer, comparable, and publicly available empirical studies on inter-macro-cluster cloud network performance are identified and deemed suitable for integration.
- **If the dataset relates to people [...]** N/A.
- **Will older versions [...] continue to be supported/hosted/maintained? [...]** Via Git history of the file.
- **If others want to extend/augment/build on/contribute [...]** By suggesting new data sources or providing updated parameters via GitHub issues/PRs.
- **Any other comments?** See overall benchmark maintenance plan in Appendix K.

### D.11 Data Management, Licensing, and Accessibility

Ensuring high-quality dataset practices is crucial for a benchmark resource.

- **Accessibility:** All necessary code and configuration files are available in the open-source repository (`https://github.com/HewlettPackard/sustain-cluster`). Processed datasets (prices, carbon intensity, weather) are included directly. The large workload trace is provided as a `.zip` file within the repository, which is automatically extracted on first use. Raw data or links to original sources are provided where applicable (e.g., in `data/electricity_prices/README.md`).

- **Licensing:** The DCcluster-Opt codebase is released under the MIT License. The integrated datasets are subject to the licenses and terms of use of their original sources. The Alibaba trace is typically available under a license requiring non-commercial use and attribution [6]. Data from Electricity Maps and Open-Meteo are subject to their respective API terms (often allowing research use with attribution). Cloud provider pricing is publicly available information. Users are responsible for adhering to the terms of the original data sources.

- **Metadata (Croissant):** In line with best practices promoted by the NeurIPS Datasets & Benchmarks track [56], we are committed to providing machine-readable metadata using the Croissant standard (`https://mlcommons.org/croissant/`) for the core processed datasets (workload, standardized price, carbon intensity, weather). This will enhance discoverability, validation, and integration with standard ML tools. Initial Croissant descriptions will be made available alongside the data files or in a dedicated metadata directory within the repository.

- **Maintenance:** A dataset and benchmark maintenance plan outlining procedures for updates, bug fixes, versioning, and potential additions of new data or locations is provided in Appendix K.

**Licensing Considerations** The DCcluster-Opt codebase itself is released under the MIT License (see Appendix L). The various real-world datasets integrated into DCcluster-Opt are subject to the licenses and terms of use of their original respective sources. These typically include:

- **Alibaba Cluster Trace 2020 [6]:** Generally requires attribution and is often for non-commercial research use.
- **Electricity Maps Data [7]:** Subject to their API terms and potentially the Open Database License (ODbL).
- **GridStatus.io Data [8]:** Subject to their terms; open-source components under licenses like BSD-3-Clause.
- **Open-Meteo Data [9]:** under Creative Commons Attribution 4.0 International (CC BY 4.0).
- **Cloud Provider Transmission Costs:** Based on publicly available pricing information.
- **Persico et al. Network Delay Data [16]:** Academic publication, standard citation practices apply.

Detailed licensing information for each dataset component is provided within its respective datasheet in Appendix D.10. Users of DCcluster-Opt are responsible for understanding and complying with all applicable original data source licenses and terms.

# E  Network Model Details

This section provides detailed information on how DCcluster-Opt models the penalties associated with transmitting tasks between geographically distributed data centers, as introduced in Section 3.4 of the main paper. These penalties include monetary cost, energy consumption, associated carbon emissions, and network latency (delay).

## E.1  Transmission Cost (Monetary)

The monetary cost of transferring a task's data to a remote data center is calculated based on per-Gigabyte (GB) egress and inter-region data transfer rates published by major cloud providers.

- **Data Sources:** We have compiled cost matrices based on public pricing from:
    - Amazon Web Services (AWS): `https://aws.amazon.com/ec2/pricing/on-demand/` (see "Data Transfer OUT from Amazon EC2" to Internet/other AWS regions).
    - Google Cloud Platform (GCP): `https://cloud.google.com/vpc/pricing` (see "Inter-region data transfer").
    - Microsoft Azure: `https://azure.microsoft.com/en-us/pricing/details/bandwidth/` (see "Inter Region" section).
- **Storage:** These rates are compiled into provider-specific CSV files (e.g., `data/network_cost/aws_transmission_cost_matrix.csv`, `gcp_transmission_cost_matrix.csv`, `azure_transmission_cost_matrix.csv`). Each matrix contains the cost in USD to transfer 1 GB of data from an origin cloud region to a destination cloud region. A `custom_transmission_cost_matrix.csv` can also be used if `cloud_provider: "custom"` is specified in `sim_config.yaml`.
- **Calculation:** When a task with data size $S_{bw}$ (in GB, from `task.bandwidth_gb`) is routed from an origin data center (mapped to origin cloud region $R_{orig}$) to a destination data center (mapped to destination cloud region $R_{dest}$), the cost is:

$$\text{Cost}_{Tx}(\$) = S_{bw} \times \text{CostPerGB}(R_{orig}, R_{dest})$$

  The mapping from DCcluster-Opt `location` codes to cloud provider regions is handled by the `utils/transmission_region_mapper.py` script, based on the selected `cloud_provider`. If $R_{orig} = R_{dest}$ (i.e., an intra-region transfer, or local execution), the transmission cost is typically $0.00, though some providers might have intra-region costs between availability zones, which are simplified to zero for inter-DC focus here.

### E.2 Transmission Energy Consumption

Energy is consumed by network equipment (routers, switches, optical gear) to transfer data.

- **Methodology:** We estimate this energy using an average electricity intensity factor for data transmission.
- **Formula:**
$$\text{Energy}_{Tx}(\text{kWh}) = S_{bw} \text{ (GB)} \times F_{kWh/GB}$$
- **Factor Used ($F_{kWh/GB}$):** By default, DCcluster-Opt uses $F_{kWh/GB} = 0.06$ kWh/GB. This value is a commonly cited approximate average based on studies analyzing the energy intensity of data transmission over the internet, such as Aslan et al. [38].
- **Scope:** This represents the end-to-end energy consumed by the network infrastructure for the transfer.

### E.3 Transmission Carbon Emissions

The energy consumed for data transmission results in carbon emissions, dependent on the carbon intensity of the electricity grids powering the network infrastructure.

- **Methodology:** We attribute the emissions for a data transfer primarily to the electricity grid of the *origin* data center, as this is often where significant data processing and uplink activities occur, and it provides a tractable and consistent assignment point.
- **Formula:**
$$\text{Emissions}_{Tx}(\text{kgCO}_2\text{eq}) = \text{Energy}_{Tx}(\text{kWh}) \times CI_{orig}(\text{kgCO}_2\text{eq/kWh})$$
- $CI_{orig}$: The current grid carbon intensity of the data center from which the task's data is being transmitted (the origin DC of the transfer), obtained from the `CI_Manager` for that location at the time of transmission.

### E.4 Transmission Delay (Latency)

Routing tasks to remote data centers introduces latency, which impacts when a task can begin execution. This delay is modeled as the sum of serialization delay and propagation delay.

- **Formula:** The total delay $D_{Tx}$ in seconds is calculated as:
$$D_{Tx}(\text{s}) = \left( \frac{S_{bw} \text{ (GB)} \times 8000 \text{ (Mb/GB)}}{\text{Throughput (Mbps)}} \right) + \left( \frac{\text{RTT (ms)}}{1000 \text{ (ms/s)}} \right)$$
- **Throughput (Mbps) and RTT (ms) Source:** These values are based on empirical inter-datacenter network performance measurements between major geographical macro-clusters (Europe (EU), United States (US), Asia-Pacific (AP), South America (SA)) as published by Persico et al. [16]. The specific throughput and RTT values are hardcoded as dictionaries within `data/network_cost/network_delay.py`.
- **Mapping Process:**
    1. The DCcluster-Opt `location` codes for the origin and destination data centers are first mapped to their corresponding cloud provider region strings (e.g., 'us-east-1', 'eu-west-3') using `utils/transmission_region_mapper.py`.
    2. These cloud provider region strings are then mapped to one of the four macro-clusters (EU, US, AP, SA) using dictionaries within `data/network_cost/network_delay.py` (e.g., 'aws_region_to_cluster').
    3. The throughput and RTT values between the determined origin and destination macro-clusters are then looked up from the stored tables.
- **Intra-Macro-Cluster Transfers:** If the origin and destination DCs map to the *same* macro-cluster (e.g., US to US), a default high throughput (e.g., 1000 Mbps, representing 1 Gbps) and a minimal RTT (e.g., 5-10 ms, depending on implementation details in `network_delay.py` when 'source == dest') are typically assumed. In such cases, the delay is primarily dominated

by the serialization time for very large data transfers. For transfers within the **exact same specific cloud region** (if this distinction is made before macro-cluster mapping), the delay is often considered negligible or a very small fixed value in many models, but DCcluster-Opt applies the formula using these intra-cluster parameters.

- **Simulation Impact:** This calculated $D_{Tx}$ is the duration for which a remotely assigned task is held "in-transit" before being added to the destination data center's pending queue.

# F    Environment API & Configuration Details

This section expands on the environment API and configuration details mentioned in Section 5 of the main paper, providing specific implementation details relevant for users and developers interacting with the DCcluster-Opt benchmark.

## F.1    MDP Formulation Details

This section provides a detailed breakdown of the Markov Decision Process (MDP) components for the DCcluster-Opt scheduling problem, as summarized in Table 1 of the main paper. The centralized agent interacts with the environment at each 15-minute timestep $t$.

- **State ($s_t$):** The state observed by the agent at timestep $t$. As detailed in Section 5.1 of the main paper, this is typically a list of feature vectors, one for each of the $k_t$ pending tasks. Each vector includes:
    - Global context: Time of day/year (e.g., via sine/cosine encoding), potentially forecasts for environmental factors if available and configured.
    - Task attributes: For each pending task, its origin DC, resource requirements (CPU, GPU, Memory, Bandwidth), estimated duration, time remaining until its SLA deadline, and potentially how long it has already been waiting or deferred.
    - Current DC status: For all $N$ data centers, information such as current resource availability (CPU, GPU, Memory as percentages or absolute values), estimated queue lengths or current load, real-time electricity price, and real-time grid carbon intensity.

    The state space $\mathcal{S}$ is high-dimensional. A key characteristic is its variable structure: since the number of pending tasks $k_t$ can change at each timestep, the overall size or length of the state representation $s_t$ (e.g., the length of the list of task vectors) is not fixed. This necessitates specific handling by RL algorithms (see Appendix F.2.3 for details on handling).

- **Action ($a_t$):** The action taken by the agent at timestep $t$. This consists of a sequence of $k_t$ individual decisions $\{a_{t,1}, a_{t,2}, ..., a_{t,k_t}\}$, one for each of the $k_t$ pending tasks identified in state $s_t$. Each individual action $a_{t,i}$ for task $i$ is chosen from the discrete action set $\mathcal{A}_i = \{0, 1, ..., N\}$, where $N$ is the number of data centers:
    - $a_{t,i} = 0$ (Temporal Deferral): The $i$-th task is held over and will be reconsidered by the agent at the next timestep, $t + 1$.
    - $a_{t,i} = j$, where $j \in \{1, ..., N\}$ (Geographical Placement): The $i$-th task is assigned to data center $j$. If data center $j$ is remote from the task's origin, this will typically incur transmission costs and a transmission delay before the task is added to data center $j$'s local processing queue.

    The full action taken by the agent at timestep $t$, $a_t$, is the collection of these $k_t$ individual decisions. The size of $a_t$ thus also varies with $k_t$.

- **Transition ($P(s_{t+1}|s_t, a_t)$):** The environment dynamics are complex and determined by the DCcluster-Opt simulator. Given the current state $s_t$ and the agent's actions $a_t$:
    - New tasks may arrive into the system based on the pre-defined workload trace (e.g., Alibaba trace) and the task origin generation model (Section F.5).
    - Tasks that were assigned to a data center $j$ (i.e., $a_{t,i} = j > 0$) are routed. If remote, transmission costs are logged, and a transmission delay is calculated; the task enters an "in-transit" state and will only become available at DC $j$'s queue after this delay elapses (potentially in a future timestep).

– Each data center $j$ attempts to schedule tasks from its local pending queue based on its current resource availability (CPU, GPU, Memory). Successfully scheduled tasks transition to a "running" state, consuming resources.

– The internal physical state of each data center (IT power consumption, heat generated, cooling power, internal temperatures) is updated based on its current load and environmental conditions using the physics-informed models.

– External environmental factors (electricity price, carbon intensity, weather) for each location advance to their values for the next 15-minute interval based on the integrated real-world data streams.

– The next state $s_{t+1}$ is then constructed, comprising information about all tasks now pending (newly arrived + tasks deferred from $s_t$ + tasks that completed transmission delay) and the updated global and per-DC context.

The transition function $P$ is thus implicitly defined by the comprehensive simulation logic.

- **Reward** ($R(s_t, a_t, s_{t+1})$)**:** A scalar reward $r_t$ is computed after the transition to $s_{t+1}$ has occurred and all outcomes for timestep $t$ (from taking actions $a_t$ in state $s_t$) are known. As detailed in Section 5.3 of the main paper and further in Appendix F.4, this reward is typically a weighted sum of multiple objectives, configured via the modular reward system. Common components include penalties for energy cost, carbon emissions (from both operations and transmission), SLA violations, and transmission costs. The goal is usually to maximize this reward (implying minimization of penalties).

- **Goal:** The agent's objective is to learn an optimal policy $\pi(a_t|s_t)$ that maximizes the expected cumulative discounted reward over an episode (e.g., a fixed duration like 30 days): $E_\pi[\sum_{k=0}^{T} \gamma^k r_{t+k}]$, where $\gamma \in [0, 1]$ is the discount factor and $T$ is the episode horizon.

### F.2 Observation Space Specification

The observation provided to the agent at each timestep $t$ is designed to facilitate per-task decision making within a global context.

#### F.2.1 Structure and Format

The `env.reset()` and `env.step()` methods return the observation $s_t$ as a standard Python **list**.

- If $k_t > 0$ tasks are pending, the list contains $k_t$ elements.
- Each element is a NumPy array of `dtype=np.float32`, representing the feature vector for a single task $i$.
- If $k_t = 0$ (no tasks pending), an empty list `[]` is returned.

This variable-length list structure necessitates handling mechanisms within the RL algorithm, as discussed in Section 5.1 of the main paper and detailed below for the provided examples.

#### F.2.2 Default Per-Task Feature Vector

The default feature vector for each task $i$, constructed by the `_get_obs` method in `TaskSchedulingEnv`, concatenates the following features in order:

**Global Time Features (4 dimensions):** Captures the cyclical nature of time.

- Sine of Day of Year: $\sin(2\pi \times \text{day\_of\_year}/365.0)$
- Cosine of Day of Year: $\cos(2\pi \times \text{day\_of\_year}/365.0)$
- Sine of Hour of Day: $\sin(2\pi \times (\text{hour} + \text{minute}/60.0)/24.0)$
- Cosine of Hour of Day: $\cos(2\pi \times (\text{hour} + \text{minute}/60.0)/24.0)$

**Task-Specific Features (5 dimensions):** Describes the task requiring a decision.

- Origin DC ID: Integer ID of the datacenter where the task originated (`task.origin_dc_id`).
- CPU Cores Required: Normalized core requirement (`task.cores_req`).

- GPU Units Required: Normalized GPU requirement (`task.gpu_req`, fractional value).
- Estimated Duration: Task execution time in minutes (`task.duration`).
- Time Until Deadline: Remaining time in minutes until the task's SLA deadline expires, clipped at 0 $(\max(0, (\texttt{task.sla\_deadline} - \texttt{current\_time}).total\_seconds()/60.0))$.

**Per-Datacenter State Features ($5 \times N$ dimensions):** Provides the current status of all $N$ data centers, concatenated sequentially (DC1 features, DC2 features, ..., DCN features). For each DC $j$:

- Available CPU Cores (%): Fraction of total cores currently available ($dc_j.\texttt{available\_cores}/dc_j.\texttt{total\_cores}$).
- Available GPUs (%): Fraction of total GPUs currently available ($dc_j.\texttt{available\_gpus}/dc_j.\texttt{total\_gpus}$).
- Available Memory (%): Fraction of total memory currently available ($dc_j.\texttt{available\_mem}/dc_j.\texttt{total\_mem\_GB}$).
- Current Carbon Intensity (kgCO$_2$/kWh): Normalized grid carbon intensity ($dc_j.\texttt{ci\_manager.get\_current\_ci(norm=False)}/1000$).
- Current Electricity Price ($/kWh): Normalized electricity price ($dc_j.\texttt{price\_manager.get\_current\_price()}/100$).

Therefore, the default dimension of each per-task observation vector is $4 + 5 + (5 \times N)$.

### F.2.3 Handling Variable-Length Observations and Actions in RL Agents

A key characteristic of the DCcluster-Opt benchmark is that the number of pending tasks ($k_t$) at each timestep $t$ is dynamic. This results in:

- **Variable-Length Observation List:** The state $s_t$ provided by `env.step()` is a list of $k_t$ feature vectors (one per task).
- **Variable-Length Action Requirement:** The agent must output a corresponding list of $k_t$ actions.

Standard deep RL neural network architectures (e.g., MLPs) typically expect fixed-size tensor inputs. Therefore, specific strategies are needed to bridge this gap when implementing learning agents. Below, we discuss common approaches, including those used in our provided examples (SAC with MLPs, and considerations for attention models or other algorithms like PPO).

**1. Padding and Masking (Common for Off-Policy MLP-based Agents like SAC):** This is the strategy employed in our baseline SAC implementation (`train_rl_agent.py`) using MLP-based actor and critic networks (`ActorNet`, `CriticNet`).

- **Replay Buffer** (`FastReplayBuffer`)**:**
  - When a transition $(s_t, a_t, r_t, s_{t+1}, d_t)$ is added to the buffer, the list of $k_t$ observation vectors for $s_t$ (and $s_{t+1}$) is padded with dummy values (e.g., zeros) up to a pre-defined maximum length, `max_tasks`. This `max_tasks` is a crucial hyperparameter defined in `algorithm_config.yaml`.
  - Similarly, the list of $k_t$ actions for $a_t$ is padded (e.g., with -1).
  - Crucially, the buffer also stores boolean masks (`mask_obs_b` for $s_t$ and `mask_next_obs_b` for $s_{t+1}$) of shape [Batch, `max_tasks`]. These masks indicate which entries in the padded tensors correspond to actual tasks versus padding.

- **Training Update:**
  - When a batch is sampled, it consists of fixed-size padded tensors (e.g., observations of shape [Batch, `max_tasks`, ObsDimPerTask]).
  - These tensors are typically flattened or reshaped (e.g., to [(Batch $\times$ `max_tasks`), ObsDimPerTask]) before being fed into the MLP networks.
  - The masks are flattened correspondingly.

- **Masked Computation:** All subsequent computations (e.g., Q-value estimation, policy log-probabilities, loss calculations, target value calculations) are performed element-wise on the outputs corresponding to all `max_tasks` slots. However, before any aggregation (summing or averaging losses, calculating expected values for SAC's soft Bellman update), the computed values for padded/invalid entries are effectively zeroed out or excluded by multiplying with the mask or selecting only valid entries. This ensures that only real task transitions contribute to the learning signal and gradients.

- **Parameter Sharing:** The same MLP network weights (actor or critic) process each valid (unmasked) task's observation vector from the flattened batch. This allows the agent to learn a general per-task decision function.

- **Pros:** Allows use of standard MLP architectures.

- **Cons:** Can be memory-inefficient if `max_tasks` is much larger than typical $k_t$. Introduces computation on padded elements (though their contribution to gradients is nullified).

**2. Attention Mechanisms (e.g., Transformers):** Our codebase includes experimental attention-based networks (`AttentionActorNet`, `AttentionCriticNet`). These are designed to directly process a set of task observations without requiring explicit iteration or fixed-order assumptions like RNNs.

- **Input Handling:**
  - The list of $k_t$ task observation vectors for state $s_t$ can be fed as a sequence to the attention network.
  - Padding is still typically required to form batches of sequences if $k_t$ varies across different transitions in a batch. An attention mask is then used within the transformer layers to ensure that padded tokens do not influence the attention scores of real tokens.

- **Processing:** Self-attention layers allow the model to weigh the importance of different tasks relative to each other when forming a representation for each task, or a global context representation.

- **Output:** The network can be designed to output $k_t$ action logits (one for each input task observation) or a global value estimate.

- **Pros:** Can model inter-task dependencies within a timestep. More naturally handles sets of inputs.

- **Cons:** More complex architecture, potentially higher computational cost per forward pass. Still often requires padding for batching in off-policy settings.

**3. On-Policy Algorithms (e.g., A2C, PPO):** On-policy algorithms do not typically use a large replay buffer of diverse past experiences, which simplifies handling variable lengths during data storage.

- **Rollout Storage** (`RolloutStorage`): Transitions are collected sequentially during a rollout.
  - The list of $k_t$ observation vectors for $s_t$ and the corresponding list of $k_t$ actions/log-probabilities can be stored directly as lists for each step in the rollout. No padding is strictly necessary for storage.

- **Actor Update:**
  - When processing the rollout data for actor updates, the agent iterates through each step $t$.
  - For each of the $k_t$ tasks in that step, its specific observation vector is fed through the actor network (parameter sharing) to compute new log-probabilities (for PPO) or for gradient calculation.
  - The advantage $A_t$ (which is a single scalar value for the entire timestep $t$, derived from the critic) is applied to each of the $k_t$ (log_prob $\times$ advantage) terms.
  - Losses are typically averaged over all valid task-actions in the rollout.

- **Critic Update (for Actor-Critic methods like A2C/PPO):**

- To avoid the critic network also needing to process variable-length inputs, a common strategy is to use an **aggregation function** (`aggregate_state_for_critic`). This function takes the full state $s_t$ (including the list of $k_t$ tasks and global context) and computes a *fixed-size* feature vector summarizing the overall situation (e.g., number of tasks, total resource demand, average SLA urgency, global DC states).
  - This fixed-size aggregated vector is then fed into a standard MLP-based value network (`ValueNet`) to estimate $V(s_t)$.

- **Pros:** No need for large padded replay buffers. Can be more direct in handling variable lists during updates for the actor.

- **Cons:** Generally less sample efficient. The aggregated state for the critic might lose some fine-grained information.

**4. Considerations for RLlib Integration:** When using libraries like Ray RLlib (Appendix G.4), users typically need to implement custom models (`TorchModelV2` or `TFModelV2`) that incorporate one of the above strategies (most commonly padding and masking, or leveraging RLlib's support for sequence inputs if using appropriate recurrent/attention base models). The environment's observation and action spaces need to be defined in a way that RLlib can understand, often using `gym.spaces.Dict` or custom preprocessors if complex list structures are returned directly.

**General Guideline:** The choice of handling mechanism depends on the RL algorithm (on-policy vs. off-policy) and the desired network architecture (MLP vs. attention/recurrent). For DCcluster-Opt, the `max_tasks` hyperparameter is key when using padding, and careful design of aggregation functions is important if that strategy is chosen for value estimation.

### F.2.4 Customizing the Observation Space

Users can modify the `_get_obs` method in `TaskSchedulingEnv` to tailor the observation space. The following attributes provide access to relevant simulation state:

- `self.current_time`: A Pandas Timestamp object for the current simulation time (UTC). Useful for extracting more complex time features.

- `self.current_tasks`: A Python list of Task objects (defined in `rl_components/task.py`) currently pending decision. Each Task object has attributes like:
  - `job_name` (str)
  - `arrival_time` (datetime)
  - `duration` (float, minutes)
  - `cores_req` (float)
  - `gpu_req` (float)
  - `mem_req` (float, GB)
  - `bandwidth_gb` (float)
  - `origin_dc_id` (int)
  - `sla_deadline` (datetime)
  - `sla_multiplier` (float)
  - `dest_dc_id` (int, assigned after action)

- `self.cluster_manager.datacenters`: A Python dictionary mapping data center names (e.g., "DC1") to their respective SustainDC environment instances. From each `dc` instance, one can access:
  - Resource Status: `dc.total_cores`, `dc.available_cores`, `dc.total_gpus`, `dc.available_gpus`, `dc.total_mem_GB`, `dc.available_mem`.
  - Queues: `dc.pending_tasks` (list of Task objects waiting), `dc.running_tasks` (list of Task objects executing).
  - Environmental Data Managers:
    * `dc.price_manager.get_current_price()`: Gets current electricity price.
    * `dc.ci_manager.get_current_ci()`: Gets current carbon intensity.

* `dc.weather_manager.get_current_weather()`: Gets current weather data (e.g., temperature).
        * Forecasts (if implemented): Methods like `get_forecasted_ci(steps=N)` could potentially be added to managers.
    – Internal State (Advanced): Access to the underlying `DatacenterModel` instance (`dc.dc_env.model`) provides detailed thermal states (e.g., average rack temperatures), though using these may require deeper understanding of the physical simulation.

Example custom features could include: length of each DC's pending queue, estimated transmission delay/cost from the task's origin to each DC, forecasted CI/price values, or the task's bandwidth requirement.

Remember to adjust the observation dimension ('obs_dim') in the RL agent initialization if customizing the feature vector.

## F.3   Action Space Implementation Notes

This section details how the agent's actions interface with the DCcluster-Opt environment and how the variable number of actions is handled internally.

### F.3.1   Action Format and API

As specified in the main paper (Section 5.2), the agent interacts with the environment via the `env.step()` method.

- **Input to** `step()`: The `step()` method expects a single argument, `actions`, which must be a Python list or a 1D NumPy array containing integers.

- **Length Requirement:** The length of the `actions` list/array, let's call it $k'_t$, *must exactly match* the number of tasks $k_t$ for which observation vectors were provided in the list returned by the *previous* call to `env.step()` or `env.reset()`. That is, $k'_t = k_t = $ len(previous_observation_list).

- **Zero Task Case:** If the previous observation was an empty list (`[]`, meaning $k_t = 0$), the agent *must* pass an empty list `[]` as the action to `env.step()`.

- **Action Values:** Each integer $a_i$ within the `actions` list must be in the discrete set $\{0, 1, \ldots, N\}$, where $N$ is the total number of configured data centers (`env.num_dcs`).

Failure to provide an action list of the correct length $k_t$ will typically raise an assertion error within the `TaskSchedulingEnv.step` method.

### F.3.2   Processing Actions within `TaskSchedulingEnv.step`

When the environment receives the `actions` list (of length $k_t$), it processes each action $a_i$ in conjunction with the corresponding $i$-th task from the internally stored `self.current_tasks` list (which matches the order of the previously returned observation list). The core logic iterates through these pairs:

1. **Iteration:** The code loops from $i = 0$ to $k_t - 1$.

2. **Task Retrieval:** Gets the $i$-th task object, `task = self.current_tasks[i]`.

3. **Action Retrieval:** Gets the corresponding action, `action = actions[i]`.

4. **SLA Check (Pre-Action):** Before processing the action, it checks if the task has already exceeded its SLA deadline (`self.current_time > task.sla_deadline`). If so, the chosen action is overridden, and the task is forcibly assigned to its *origin* data center to minimize further delay, logging a warning.

5. **Deferral Processing (Action 0):** If `action == 0` (and the SLA check didn't override), the task is marked as deferred (`task.temporarily_deferred = True`) and appended to a separate internal list, `self.deferred_tasks`. These tasks will be added back to the beginning of the `self.current_tasks` list in the *next* timestep for reconsideration.

6. **Assignment Processing (Action > 0):** If `action = j` where $1 \leq j \leq N$ (and the SLA check didn't override):

- The destination data center object (`dest_dc`) corresponding to index $j$ is identified (based on the order in `self.cluster_manager.datacenters`).
- The task's destination attributes are set: `task.dest_dc_id = dest_dc.dc_id` and `task.dest_dc = dest_dc`.
- The task object is appended to the *pending queue* of the chosen destination data center: `dest_dc.pending_tasks.append(task)`. Note: If the destination is remote, the task is not immediately schedulable; the `DatacenterClusterManager` handles the "in-transit" logic based on the assigned `dest_dc_id` and calculated delay during its own step phase (see Section E).

After iterating through all $k_t$ task-action pairs and assigning them either to the deferred list or a specific data center's pending queue (potentially implicitly triggering the "in-transit" state managed by the cluster manager), the `TaskSchedulingEnv` then calls `self.cluster_manager.step()`. This backend manager simulates the network delays for remote tasks, updates the state of all individual data centers (including attempting to schedule tasks from their respective pending queues), and returns the aggregated results used for reward calculation and the next observation.

### F.3.3   Implications for Agent Design

The agent's policy network needs to be capable of outputting $k_t$ discrete actions based on the $k_t$ observation vectors it receives.

- **Parameter Sharing (Common Approach):** As implemented in the provided `ActorNet`, a single network processes each of the $k_t$ observation vectors independently (often by flattening the batch and task dimensions during training). For each input vector (representing task $i$ in its global context), the network outputs logits or probabilities over the action space $\{0, 1, ..., N\}$. Actions are then sampled independently for each task based on these outputs.
- **Sequence Models (Alternative):** More complex architectures like RNNs or Transformers could potentially process the entire list of $k_t$ observation vectors as a sequence to output the $k_t$ actions, potentially capturing inter-task dependencies within the current batch, although this adds complexity. The provided examples use the simpler parameter-sharing approach.

The agent must be designed to handle the case where the input observation list is empty ($k_t = 0$) and output an empty action list accordingly.

### F.4   Reward Function Details

The reward signal $r_t$ drives the RL agent's learning process. DCcluster-Opt utilizes a modular and configurable reward system, allowing users to tailor the optimization objective. This section details the built-in reward components and how to combine or create custom ones.

### F.4.1   Reward System Integration

As mentioned in Section 5.3, a reward function object, typically inheriting from `rewards.base_reward.BaseReward`, is instantiated based on the `reward_config.yaml` file and passed to the `TaskSchedulingEnv` during initialization. After each simulation step, the environment calls this object's `__call__` method:

```
reward = reward_fn(cluster_info, current_tasks, current_time)
```

where `cluster_info` is a dictionary containing aggregated results and detailed information from all data centers for the just-completed timestep. The `reward_fn` then computes and returns the scalar reward $r_t$.

### F.4.2   Built-in Reward Components

The following reward components are provided in `rewards/predefined/`. They typically return negative values representing penalties to be minimized (or positive values for rewards to be maximized,

like efficiency). The `cluster_info` dictionary passed to `__call__` contains nested information, primarily under the key `"datacenter_infos"` (a dict mapping DC names to their individual info dicts) and top-level keys for global metrics like transmission costs/emissions.

- **Energy Price** (`energy_price_reward.py`): Penalizes total energy cost.
    - Class: `EnergyPriceReward`
    - Init Args: `normalize_factor` (float, default: 1000.0)
    - Calculation:
    $$r = -\frac{\sum_{dc} \texttt{cluster\_info["datacenter\_infos"][dc]["\_\_common\_\_"]["energy\_cost\_USD"]}}{\texttt{normalize\_factor}}$$

- **Carbon Emissions** (`carbon_emissions_reward.py`): Penalizes total carbon emissions (compute + cooling + transmission).
    - Class: `CarbonEmissionsReward`
    - Init Args: `normalize_factor` (float, default: 100.0)
    - Calculation:
    $$\mathrm{TotalEmissions} = \left( \sum_{dc} \texttt{cluster\_info["datacenter\_infos"][dc]["\_\_common\_\_"]["carbon\_emissions\_kg"]} \right)$$
    $$+ \texttt{cluster\_info["transmission\_emissions\_total\_kg"]}$$
    $$r = -\frac{\mathrm{TotalEmissions}}{\texttt{normalize\_factor}}$$

- **Energy Consumption** (`energy_consumption_reward.py`): Penalizes total energy consumed (compute + cooling + transmission).
    - Class: `EnergyConsumptionReward`
    - Init Args: `normalize_factor` (float, default: 1000.0)
    - Calculation:
    $$\mathrm{TotalEnergy} = \left( \sum_{dc} \texttt{cluster\_info["datacenter\_infos"]} \right.$$
    $$\texttt{[dc]["\_\_common\_\_"]["energy\_consumption\_kwh"]} \bigg)$$
    $$+ \texttt{cluster\_info["transmission\_energy\_total\_kwh"]}$$
    $$r = -\frac{\mathrm{TotalEnergy}}{\texttt{normalize\_factor}}$$

- **Transmission Cost** (`transmission_cost_reward.py`): Penalizes monetary cost of data transfers.
    - Class: `TransmissionCostReward`
    - Init Args: `normalize_factor` (float, default: 10.0)
    - Calculation:
    $$r = -\frac{\texttt{cluster\_info["transmission\_cost\_total\_usd"]}}{\texttt{normalize\_factor}}$$

- **Transmission Emissions** (`transmission_emissions_reward.py`): Penalizes carbon emissions specifically from data transfers.
    - Class: `TransmissionEmissionsReward`
    - Init Args: `normalize_factor` (float, default: 10.0)
    - Calculation:
    $$r = -\frac{\texttt{cluster\_info["transmission\_emissions\_total\_kg"]}}{\texttt{normalize\_factor}}$$

- **SLA Penalty** (`sla_penalty_reward.py`): Penalizes tasks finishing after their deadline.
    - Class: `SLAPenaltyReward`
    - Init Args: `penalty_per_violation` (float, default: 1.0)
    - Calculation:
    $$\mathrm{TotalViolations} = \sum_{dc} \texttt{cluster\_info["datacenter\_infos"]}$$
    $$\texttt{[dc]["\_\_common\_\_"]["\_\_sla\_\_"]["violated"]}$$
    $$r = -(\mathrm{TotalViolations} \times \texttt{penalty\_per\_violation})$$

- **Efficiency** (`efficiency_reward.py`): Rewards completing more tasks per unit of energy (example formulation).
    - Class: `EfficiencyReward`
    - Init Args: `epsilon` (float, default: 1e-6, for stability)

– Calculation (Conceptual):

$$\text{TotalCompleted} = \sum_{dc} \texttt{cluster\_info["datacenter\_infos"]}$$
$$\texttt{[dc]["\_\_common\_\_"]["\_\_sla\_\_"]["met"]}$$
$$\text{TotalEnergy} = \dots \text{ (as above) } \dots$$
$$r = \frac{\text{TotalCompleted}}{\text{TotalEnergy} + \texttt{epsilon}}$$

Note: The exact keys within `cluster_info` should be verified against the implementation in `DatacenterClusterManager.step`. Normalization factors should be tuned based on the expected scale of the metrics for a given scenario to ensure balanced reward signals.

### F.4.3 Composite Reward Configuration

The `CompositeReward` class (`rewards/predefined/composite_reward.py`) allows combining multiple reward components with specific weights. It is configured via the `reward` section in `reward_config.yaml`.

**Example** `reward_config.yaml`:

```
reward:
   # Optional:  CompositeReward can perform its own running
normalization
   normalize:  false

   # Dictionary defining components, weights, and args for
sub-rewards
   components:
     energy_price:
       weight:  0.4
       args:  { normalize_factor:  10000 } # Scale raw cost

     carbon_emissions:
       weight:  0.3
       args:  { normalize_factor:  100 } # Scale raw kgCO2

     sla_penalty:
       weight:  0.2
       args:  { penalty_per_violation:  5.0 } # Penalty per
violation

     transmission_cost:
       weight:  0.1
       args:  { normalize_factor:  50 } # Scale raw transmission
cost

     # Example:  Include efficiency if desired
     # efficiency:
       # weight:  0.05
       # args:  { epsilon:  1e-6 }
```

When initialized, `CompositeReward` uses the reward registry (`rewards.reward_registry`) to find and instantiate each named component (e.g., "energy_price" maps to `EnergyPriceReward`) with its specified `args`. During the `__call__`, it calculates each component's value, optionally applies running mean/std normalization if `normalize=True`, multiplies by the `weight`, and sums the results to produce the final scalar reward $r_t$. The raw values of each component are stored and can be retrieved using `get_last_components()` for detailed logging.

### F.4.4 Creating Custom Rewards

Users can implement novel reward functions tailored to specific research questions.

1. **Create File:** Add a new Python file in the rewards/predefined/ directory (e.g., my_custom_reward.py).

2. **Inherit & Implement:** Define a class inheriting from rewards.base_reward.BaseReward. Implement the __call__(self, cluster_info, current_tasks, current_time) method containing your custom logic. Optionally store the result in self.last_reward.

   **Example** my_custom_reward.py**:**

   ```python
   # rewards/predefined/my_custom_reward.py
   from rewards.base_reward import BaseReward
   from rewards.registry_utils import register_reward

   @register_reward("my_custom") # Choose a unique name
   class MyCustomReward(BaseReward):
       def __init__(self, custom_param=1.0):
         super().__init__()
         self.custom_param = custom_param

       def __call__(self, cluster_info, current_tasks,
   current_time):
           # Example:  Penalize variance in CPU utilization across
   DCs
           cpu_utils = [info["__common__"]["cpu_util_percent"]
             for info in cluster_info["datacenter_infos"].values()]
           util_variance = np.var(cpu_utils) if cpu_utils else 0
           reward = -util_variance * self.custom_param
           self.last_reward = reward
           return reward
   ```

3. **Register:** Decorate the class with @register_reward("your_unique_name") from rewards.registry_utils.

4. **Import in Registry:** Crucially, add an import statement for your new class at the top of rewards/reward_registry.py. This ensures the class is loaded and the decorator runs, making it available to CompositeReward and get_reward_function.

   **Example addition to** rewards/reward_registry.py**:**

   ```python
   # rewards/reward_registry.py
   # ...  other imports ...
   from rewards.predefined.my_custom_reward import
   MyCustomReward # Add this line
   ```

Your custom reward can then be used directly or included by name in the components dictionary within reward_config.yaml.

## F.5 Task Origin Generation Logic

To simulate realistic workload arrival patterns across the global cluster, tasks extracted from the base workload trace (e.g., Alibaba trace) are assigned an origin data center before being presented to the scheduling agent. This assignment is not random but follows a hybrid probabilistic model designed to reflect that different geographical regions generate varying amounts of work at different times of day. This logic is primarily implemented in the utils/workload_utils.py::assign_task_origins() function, which is called by extract_tasks_from_row() during the task loading phase of the DatacenterClusterManager. A visual explanation of this logic can be seen in Figure 2 of the main paper.

The assignment process involves the following steps for each batch of newly arriving tasks at a given simulation timestep (UTC):

1. **Calculate Activity-Weighted Scores for Each Datacenter:** For each data center $d$ defined in `datacenters.yaml`, a score is computed:

$$\text{score}_d = \text{population\_weight}_d \times \text{activity\_factor}_d(t_{local})$$

where:

- population_weight$_d$: A static, pre-defined relative weight for data center $d$ (from `datacenters.yaml`), representing its baseline importance or the general activity level of the region it serves.
- $t_{local}$: The current local time at data center $d$, calculated by applying its `timezone_shift` (from `datacenters.yaml`) to the current global UTC simulation time.
- activity_factor$_d(t_{local})$: A dynamic factor that boosts the score if $t_{local}$ falls within typical local business hours. As implemented, this factor is:

$$\text{activity\_factor}_d(t_{local}) = \begin{cases} 1.0 & \text{if } 8 \leq \text{hour}(t_{local}) < 20 \\ 0.3 & \text{otherwise} \end{cases}$$

This simulates higher task generation rates during daytime working hours in each respective region.

2. **Normalize Scores to Probabilities:** The scores for all $N$ data centers are normalized to form a probability distribution:

$$P(\text{origin} = d) = \frac{\text{score}_d}{\sum_{j=1}^{N} \text{score}_j}$$

3. **Assign Origin Probabilistically:** For each individual task in the current batch of new arrivals, an origin data center ID is sampled independently from the set of all data center IDs according to the calculated probability distribution $P(\text{origin} = d)$ (using `np.random.choice`). This `task.origin_dc_id` is then set on the task object.

This probabilistic, time-and-population-aware assignment mechanism ensures that the spatial distribution of incoming workloads is dynamic and reflects plausible real-world generation patterns, rather than tasks originating uniformly or from a fixed location. This, in turn, presents a more realistic challenge to the global scheduling agent.

## F.6 Detailed Configuration File Explanations

The behavior and parameters of the DCcluster-Opt simulation environment and associated training scripts are primarily controlled by a set of YAML and JSON configuration files, typically located in the `configs/env/` and `configs/dcs/` directories. This section provides a detailed explanation of each key configuration file and its parameters.

### F.6.1 `sim_config.yaml`: Global Simulation Settings

This file controls the overall simulation setup, including timing, workload, and global strategy.

`simulation:year:` (Integer) The starting year for pulling environmental data (e.g., electricity prices, carbon intensity, weather). Example: 2023.

`simulation:month:` (Integer) The starting month (1-12). Example: 7 for July.

`simulation:init_day:` (Integer) The starting day of the month (1-31). Example: 1.

`simulation:init_hour:` (Integer) The starting hour of the day (0-23, UTC). Example: 0.

`simulation:duration_days:` (Integer) The total length of the simulation period in days. Example: 30 for a one-month run.

`simulation:timestep_minutes:` (Integer) The duration of each simulation step in minutes. Fixed at 15 for DCcluster-Opt.

`simulation:workload_path:` (String) Path to the processed AI workload trace file (typically a `.pkl` file). Example: `"data/workload/alibaba_2020_dataset/result_df_full_year_2020.pkl"`.

`simulation:cloud_provider:` (String) Specifies the cloud provider whose transmission cost matrix and region mapping conventions are used. Supported values: `"aws"`, `"gcp"`, `"azure"`, `"custom"`. Example: `"aws"`.

`simulation:shuffle_datacenters:` (Boolean) If `true`, the internal order of data centers is shuffled at the start of each episode (relevant for some RL agents or rule-based strategies that might otherwise learn an implicit order). For evaluations, often set to `false` for strict reproducibility. Example: `true`.

`simulation:strategy:` (String) Defines the top-level scheduling strategy.

- `"manual_rl"`: Indicates that an external RL agent (via `TaskSchedulingEnv`) will provide task assignment actions. This is used for training and evaluating RL agents.
- Rule-Based Controller Names (e.g., `"lowest_carbon"`, `"local_only"`): Invokes the specified built-in rule-based controller. See Appendix G.2 for a list.

Example: `"manual_rl"`.

`simulation:use_tensorboard:` (Boolean) Whether to enable TensorBoard logging during training (typically `true` for training, `false` for evaluation). Example: `true`.

### F.6.2 `datacenters.yaml`: Cluster and Data Center Definitions

This file defines the composition and characteristics of the simulated data center cluster. It contains a list under the `datacenters:` key, where each item is a dictionary defining a single DC:

`dc_id:` (Integer) A unique numerical identifier for the data center. Example: 1.

`location:` (String) A location code (e.g., `"US-CAL-CISO"`, `"DE-LU"`) that links this DC to its specific real-world environmental datasets (electricity price, carbon intensity, weather) and network region mappings. See Appendix D.1 for a list of supported codes. Example: `"US-NY-NYIS"`.

`timezone_shift:` (Integer) The timezone offset from UTC in hours for this DC's location. Used for calculating local time (e.g., for task origin logic). Example: -5 for US Eastern Time.

`population_weight:` (Float) A relative weight used in the probabilistic task origin generation model (see Appendix F.5). Higher values increase the likelihood of tasks originating from this DC. Example: 0.25.

`total_cores:` (Integer) The total number of schedulable CPU cores in this data center. Example: 50000.

`total_gpus:` (Integer) The total number of schedulable GPUs in this data center. Example: 1000.

`total_mem:` (Integer) The total schedulable memory capacity in Gigabytes (GB). Example: 80000.

`dc_config_file:` (String) Path to the JSON file containing detailed low-level physical parameters for this specific data center's model (see details for `dc_config.json` below). Example: `"configs/dcs/dc_config.json"`.

`use_rl_hvac:` (Boolean, optional) If `true`, this DC will attempt to use a pre-trained RL policy for HVAC control. Defaults to `false`. Example: `true`.

`hvac_policy_path:` (String, optional) Path to the checkpoint file of the pre-trained HVAC control policy, used if `use_rl_hvac` is true. Example: `"checkpoints/hvac_policy_ppo.pth"`.

`hru_enabled:` (Boolean, optional) If `true`, simulates a Heat Recovery Unit for this DC, potentially reducing cooling energy. Defaults to `false`. Example: `true`.

### F.6.3 `reward_config.yaml`: Reward Function Configuration

This file specifies the components and weights for the multi-objective reward function used by the `TaskSchedulingEnv`. It typically defines a `CompositeReward`. See Appendix F.4 for a detailed example and explanation of built-in reward components.

`reward:components:` (Dictionary) Maps reward component names (e.g., `"energy_price"`, `"carbon_emissions"`) to their configurations.

- `weight:` (Float) The weight assigned to this component in the total reward.

- `args:` (Dictionary, optional) Arguments passed to the constructor of the specific reward component class (e.g., `normalize_factor`, `penalty_per_violation`).

`reward:normalize:` (Boolean, optional) If `true`, the `CompositeReward` class will apply its own running mean/std normalization to each component's output before weighting and summing. Defaults to `false` if not specified.

### F.6.4 `algorithm_config.yaml` (or similar, e.g., `sac_config.yaml`, `ppo_config.yaml`): RL Algorithm Hyperparameters

This file contains hyperparameters specific to the RL algorithm being used for training the global scheduler (or local controllers like the HVAC agent). While primarily used by the training scripts (`train_rl_agent.py`, `train_hvac_agent.py`), some parameters interact with or define aspects of the environment setup for RL.

`algorithm:max_tasks:` (Integer, for SAC with `FastReplayBuffer`) The maximum number of tasks to pad to in each transition stored in the replay buffer. This affects memory usage and the input shape for the neural networks. Example: 400.

`algorithm:hidden_dim:` (Integer) Size of hidden layers in actor/critic networks.

`algorithm:device:` (String) Computation device (`"cpu"`, `"cuda"`, or `"auto"`).

Other parameters include learning rates, batch sizes, discount factors ($\gamma$), exploration parameters (e.g., SAC's $\alpha$), etc., specific to the chosen RL algorithm. Refer to Appendix G.3.4 and Appendix I for tables of hyperparameters used.

### F.6.5 `configs/dcs/dc_config.json`: Low-Level Datacenter Physical Parameters

This JSON file provides detailed physical and operational parameters for the `DatacenterModel` used within each `SustainDC` instance. It allows fine-grained control over the simulated hardware and its behavior. Key sections include:

`data_center_configuration:` Defines layout (e.g., `NUM_RACKS`, `CPUS_PER_RACK`), and thermal properties like rack supply/return approach temperatures.

`hvac_configuration:` Specifies parameters for the HVAC system components: CRAC units (reference fan power, flow rates), chillers (EnergyPlus-derived coefficients, COP parameters), cooling towers (reference fan power, flow rates), and pumps (pressure drops, efficiencies, flow rates).

`server_characteristics:` Defines power consumption curves and operational ranges for IT components:
- CPU power ratio bounds (`CPU_POWER_RATIO_LB/UB`) vs. inlet temperature and utilization.
- IT fan airflow ratios (`IT_FAN_AIRFLOW_RATIO_LB/UB`) and reference power/velocity.
- Server inlet temperature operating range (`INLET_TEMP_RANGE`).
- Default power characteristics (idle/max watts) for server types (e.g., `HP_PROLIANT`) and GPU types (e.g., `NVIDIA_V100`).

Modifying this file allows simulation of different hardware generations, cooling system efficiencies, or thermal management strategies. Detailed explanations of these parameters and their impact are often found within the codebase comments and the underlying physics models (see Appendix B).

This detailed configuration system allows users to flexibly define a wide range of scenarios for benchmarking sustainable scheduling algorithms in DCcluster-Opt.

## F.7 Computational Resources and Training Time

The experiments reported in this paper, including the training of RL agents and the execution of evaluation runs, were conducted on a compute node with the following specifications:

- **CPU:** 2x Intel(R) Xeon(R) Platinum 8470 processors (each with 52 cores, supporting 2 threads per core, for a total of 208 threads). The CPUs operate between 800.00 MHz and 3800.00 MHz.

- **GPU:** 2x NVIDIA H100 PCIe. (Note: While GPUs were available, the DCcluster-Opt simulation and the provided baseline RL agent training scripts are primarily CPU-bound. GPU utilization would depend on specific RL library implementations or custom network architectures if users choose to leverage them.)
- **Memory:** 1 TB of RAM.

**RL Agent Training Time (Example for SAC Baseline):** Each training run for the baseline Soft Actor-Critic (SAC) agent (for the main task scheduling, not the optional HVAC controller) typically involved:

- **CPU Cores Utilized:** 8 CPU cores.
- **Approximate Wall-Clock Time:** Approximately 12 hours to complete the configured 10 million environment steps (as specified in `algorithm_config.yaml`).

This time can vary based on the specific RL algorithm, hyperparameters (e.g., network size, update frequency), the number of simulated data centers, and the observation space complexity. Training the optional, separate HVAC control agents (e.g., with PPO or SAC for discrete HVAC actions) typically requires fewer total steps and can be completed in a proportionally shorter time on similar hardware.

**Evaluation Run Time:** Executing a single evaluation run for one controller over a simulated 30-day period is significantly faster, typically completing in (1-3 minutes) on a single CPU core, as it involves policy rollouts without gradient updates.

## F.8 Environmental Impact of Experiments

We acknowledge that conducting the computational experiments necessary for developing and evaluating the DCcluster-Opt benchmark has an associated environmental impact. While individual simulation runs within DCcluster-Opt are designed to be efficient, the cumulative compute time for training multiple RL agents across different configurations and running numerous evaluation seeds contributes to energy consumption and carbon emissions.

Our experiments were primarily conducted using a private compute infrastructure located in North America with an estimated average grid carbon efficiency of approximately 367 gCO$_2$eq/kWh [57].

The total computational effort for the experiments presented in this paper across multiple seeds, and all RBC evaluation runs is estimated to be approximately 750 CPU-hours.

Using the Machine Learning Impact calculator [58], with an estimated power draw of 125W for the CPU-bound training tasks, the total emissions for our reported experiments are estimated to be approximately 34.41 kgCO$_2$eq.

While we have not purchased carbon offsets for this specific set of computations at the time of submission, we are committed to promoting sustainable research practices. The DCcluster-Opt benchmark itself is a tool designed to help the community develop solutions that *reduce* the environmental impact of AI. We recognize that other environmental impacts, such as water usage associated with power generation and the embodied carbon in hardware manufacturing, are also important considerations beyond the operational carbon emissions from compute.

# G  Baseline Implementations & Hyperparameters

This section provides detailed configuration information for the evaluation environment, descriptions of the baseline controllers, and the training setups for all RL agents whose results are presented in Section 6 of the main paper.

## G.1  Evaluation Environment Setup

The following configuration details apply to all baseline evaluation runs reported in Table 4, Table 5 of the main paper, and in Table 13.

### G.1.1 Simulation Configuration (`sim_config.yaml`)

The global simulation parameters controlling the evaluation period and environment setup were defined as follows:

```
simulation:
  year: 2023                # Year for env data (Price, CI, Weather)
  month: 7                  # Starting month (July)
  init_day: 1               # Starting day of the month
  init_hour: 0              # Starting hour (UTC)
  duration_days: 30         # Simulation length for aggregation
  timestep_minutes: 15      # Fixed simulation step duration
  workload_path: "data/workload/alibaba_2020_dataset/result_df_full_year_2020.pkl"
  cloud_provider: "aws"     # For transmission cost/delay mapping
  shuffle_datacenters: true # Shuffle DC order for evaluation reproducibility
  strategy: "manual_rl"     # Overridden for RBC runs by eval script
  use_tensorboard: true     # Typically false during final evaluation runs
```

For evaluation, `shuffle_datacenters` was typically set to `false` to ensure consistent DC ordering across seeds for a given controller, aiding comparison. The `strategy` is set programmatically by the evaluation script for RBCs.

### G.1.2 Datacenter Cluster Configuration (`datacenters.yaml`)

The evaluation experiments reported in the main paper utilized a simulated cluster of 5 geographically distributed data centers. The specific configuration for these data centers, as defined in the `datacenters.yaml` file used for these runs, is detailed in Table 8. All data centers used the same underlying physical parameters defined in `configs/dcs/dc_config.json` for their IT and HVAC models, allowing the differences in performance to be primarily attributed to their geographical location (affecting environmental data) and their provisioned resource capacities. For the baseline comparisons presented in Table 3 and Table 13, RL-based HVAC control was disabled (`use_rl_hvac: false` for each DC in their respective `datacenters.yaml` entries, implying default fixed HVAC setpoints).

Table 8: Datacenter configuration for the 5-DC cluster used in baseline evaluations.[1]

| DC ID | Location Code | Timezone | Population Weight | Cores | GPUs | Mem (GB) | Physics Cfg |
|---|---|---|---|---|---|---|---|
| 1 | US-CAL-CISO | -7 | 0.18 | 50,000 | 1000 | 80,000 | dc_config.json |
| 2 | DE-LU | +1 | 0.22 | 85,000 | 600 | 80,000 | dc_config.json |
| 3 | CL-SIC | -5 | 0.20 | 110,000 | 300 | 60,000 | dc_config.json |
| 4 | SG | +8 | 0.25 | 75,000 | 700 | 50,000 | dc_config.json |
| 5 | AU-NSW | +11 | 0.15 | 65,000 | 300 | 60,000 | dc_config.json |

[1]The distribution and quantity of Cores, GPUs, and Memory for each simulated data center were chosen to approximate a total IT power capacity roughly equivalent to a 1MW facility, following typical power budgets per component (e.g., 20W/core, 500W/GPU, 2.5W/GB RAM, as detailed in the main paper's README/DC modeling guidelines). The variation in resource counts across different locations (e.g., DC1 being more GPU-heavy, DC3 more CPU-heavy) is intentional, designed to represent infrastructural heterogeneity within the cluster and to demonstrate that the proposed scheduling framework can adapt to data centers with diverse hardware configurations and capacities.

**Rationale for Location Selection:** The chosen locations aim to provide a diverse set of environmental and economic conditions to create a challenging and representative scenario for evaluating geo-distributed scheduling strategies:

- **US-CAL-CISO (California, USA):** Represents a major tech hub with significant renewable energy penetration (especially solar), leading to distinct diurnal patterns in both electricity price and grid carbon intensity. Experiences moderate to warm climate conditions.
- **DE-LU (Germany/Luxembourg, Europe):** Represents a central European location within the interconnected ENTSO-E grid, characterized by a mixed energy portfolio with significant wind and solar, leading to variable carbon intensity and prices. Experiences a temperate climate with distinct seasons.
- **CL-SIC (Norte Grande, Chile):** Chosen for its unique energy profile, often with high solar potential. Its location in the Southern Hemisphere provides contra-seasonal weather patterns

compared to Northern Hemisphere DCs, and its grid characteristics can differ significantly. (Note: Formerly CDEC-SIC, now part of Coordinador Eléctrico Nacional).

- **SG (Singapore):** Represents a tropical, equatorial location with consistently high temperatures and humidity year-round, posing a significant and constant cooling challenge. Its energy market and grid carbon intensity (often reliant on natural gas) provide a distinct economic and environmental profile.
- **AU-NSW (New South Wales, Australia):** Represents another Southern Hemisphere location with a different energy mix (historically coal-reliant but with increasing renewables) and distinct seasonal weather patterns that impact both energy demand and cooling. Its timezone also provides significant temporal offset from North American and European DCs.

This selection ensures that the benchmark includes locations with varying:

- **Timezones:** Covering a wide range to test temporal load shifting strategies.
- **Climate Patterns:** Affecting cooling loads and HVAC efficiency differently across the cluster.
- **Electricity Market Structures and Price Volatility:** Providing diverse economic signals.
- **Grid Carbon Intensity Profiles:** Offering clear opportunities for carbon-aware scheduling.
- **Resource Capacities:** The table also shows heterogeneous compute capacities (Cores, GPUs, Memory), adding another layer to the resource allocation challenge.

This diversity is intended to stress-test scheduling algorithms and highlight their ability to adapt to heterogeneous and dynamic global conditions

### G.1.3 Execution and Reproducibility

Each controller configuration presented in the main paper's results tables (Table 3, and Table 4), and in Table 13, was executed for the full simulated 30-day period, as defined in Appendix G.1.1. To account for stochastic elements (e.g., initial day randomization if enabled by the environment, tie-breaking in workload processing, or inherent stochasticity in RL agent policies if applicable during evaluation), each configuration was run using 10 independent random seeds. The reported results (mean ± standard deviation) are aggregated across these 10 runs.

The primary script for conducting these evaluations is the Jupyter Notebook `evaluate_DCcluster-Opt_agent.ipynb`. To reproduce a specific evaluation run:

1. **Navigate to the Notebook Directory and Activate Environment:** Open a terminal in the root of the cloned DCcluster-Opt repository.

```
cd notebooks/
# Activate your Python virtual environment, e.g.:
# source ../DCcluster-Opt_env/bin/activate
```

Then, launch Jupyter Lab or Jupyter Notebook:

```
jupyter lab
# or
# jupyter notebook
```

And open the `evaluate_DCcluster-Opt_agent.ipynb` notebook.

2. **Configure the Evaluation within the Notebook:** Modify the parameters in *Cell Section 4 ("Evaluation Parameters")* of the notebook:
    - Set `EVALUATION_STRATEGY`:
        - For Rule-Based Controllers (RBCs): Set to the desired RBC name string, e.g., `EVALUATION_STRATEGY = "local_only"`, `"lowest_carbon"`, etc. The `make_eval_env` function will then use this strategy when initializing the `DatacenterClusterManager`.
        - For the custom SAC RL agent: Set `EVALUATION_STRATEGY = "manual_rl"`.
        - For RLlib-trained agents: Set `EVALUATION_STRATEGY = "manual_rl"` and ensure you load the correct RLlib agent checkpoint (see next point).

- Set `AGENT_CHECKPOINT_PATH_COLAB` (or a similar variable name if you adapt the notebook for local use outside Colab):
  - If evaluating an RL agent (SAC, PPO, APPO, IMPALA, or SAC with advanced HVAC), update this variable to point to the specific `.pth` checkpoint file for the agent you wish to evaluate. For example:

    ```
    # For custom SAC (Geo+Time)
    AGENT_CHECKPOINT_PATH_COLAB = "/content/sustain-cluster/checkpoints/
                        train_SAC_GeoTime_EXAMPLE/best_checkpoint.pth"

    # For an RLlib PPO agent (example path)
    # AGENT_CHECKPOINT_PATH_COLAB = "/content/sustain-cluster/
            rllib_checkpoints/PPO_GeoTime_runX/checkpoint_000100/model.pt"
    # (Note: RLlib checkpoint structures can vary)
    ```

    Ensure the path is correct within the Colab environment (typically starting with `/content/sustain-cluster/...`) or adjusted for local execution relative to the project root.
- Set `EVALUATION_DAYS = 30` (or as per the specific experiment).
- Set `EVALUATION_SEED`: Iterate this seed from, for example, 0 to 9 (or any 10 distinct integers) to reproduce the 10 runs.

3. **Modify Configuration Files (If necessary for specific scenarios):**
   - For scenarios involving RL-controlled HVAC or Heat Recovery Units (HRU), ensure the relevant flags (e.g., `use_rl_hvac`, `hvac_policy_path`, `hru_enabled`) are correctly set in the `configs/env/datacenters.yaml` file that the notebook's `make_eval_env` function will load. The evaluation notebook itself might need slight modifications to select different `datacenters.yaml` files or to programmatically alter these settings if testing multiple HVAC configurations.
   - Default config files used by the notebook are typically: `configs/env/sim_config.yaml`, `configs/env/datacenters.yaml`, and `configs/env/reward_config.yaml`. Ensure these reflect the setup used for the reported results.

4. **Execute Notebook Cells:** Run all cells in the notebook sequentially. The notebook will:
   - Set up the environment.
   - Load the RL agent model if `EVALUATION_STRATEGY = "manual_rl"`.
   - Run the simulation loop for the specified number of days and seed.
   - Process the collected metrics into a Pandas DataFrame.
   - Print an aggregated summary table to the output.
   - Generate and save time-series plots of key metrics to the `outputs_colab_eval/` directory (or a similar configured output path).

5. **Collect Results:** The primary output for each run will be the aggregated summary table printed by the notebook and the detailed log file (e.g., in `logs_colab_eval/`). The mean and standard deviation across the 10 seeds can then be calculated from these individual run outputs.

By following these steps and ensuring the configuration files and agent checkpoints match those used for the original experiments, users can reproduce the evaluation results presented in this paper.

## G.2 Rule-Based Controllers (RBCs)

The following RBCs were used as baselines. They are selected by setting the `strategy` field in `sim_config.yaml` and implemented in `utils/task_assignment_strategies.py`. These heuristics typically assign tasks immediately upon arrival without using the deferral action.

**Local Only:** Assigns every task to its pre-assigned origin data center, regardless of load or environmental factors. Represents a non-optimized baseline with no geographical shifting.

**Lowest Carbon:** For each task, queries the current carbon intensity ($gCO_2eq/kWh$) of all available data centers. Assigns the task to the data center with the minimum instantaneous carbon intensity, provided it has capacity (basic fitting check might be included).

**Lowest Price:** Similar to Lowest Carbon, but assigns the task to the data center with the minimum instantaneous electricity price ($/kWh), subject to capacity.

**Most Available:** Assigns the task to the data center that currently has the highest percentage of available CPU cores, aiming to avoid heavily loaded sites.

**Round Robin:** Assigns tasks sequentially to data centers in a fixed cyclical order (DC1, DC2, ..., DCN, DC1, ...).

### G.3 Reinforcement Learning Agent (SAC Baselines)

The RL results presented in Table 3 were generated using agents trained with the Soft Actor-Critic (SAC) algorithm [42].

#### G.3.1 Custom SAC Implementation

The SAC agents (*Geo+Time*, *Geo Only*, *Time Only*) were trained using our custom implementation (`train_rl_agent.py`).

**Network Architecture:** The actor (`ActorNet`) and critics (`CriticNet`) used Multi-Layer Perceptrons (MLPs) with ReLU activations:

- Actor: Input(ObsDim) → Linear(H) → ReLU → Linear(H) → ReLU → Linear(ActionDim) → Logits.
- Critic: Input(ObsDim) → Linear(H) → ReLU → Linear(H) → ReLU → Linear(ActionDim) → Q-values. (Where H is the hidden dimension).

#### G.3.2 Training Reward Configuration

All SAC agents were trained using the following multi-objective reward configuration specified in `configs/env/reward_config.yaml`. The individual components are detailed here:.

```
reward:
  normalize: false
  components:
    energy_price: {weight: 0.5, args: {normalize_factor: 100000}}
    carbon_emissions: {weight: 0.3, args: {normalize_factor: 10}}
    transmission_cost: {weight: 0.2, args: {normalize_factor: 1}}
    sla_penalty: {weight: 0.1, args: {penalty_per_violation: 5.0}}
    transmission_emissions: {weight: 0.1, args: {normalize_factor: 1}}
```

As discussed in the main paper, this configuration's low penalty on SLA violations led the agents to prioritize cost/carbon over timeliness via excessive deferral.

#### G.3.3 Network Architecture

Both the actor (policy) and the critics (Q-functions) used Multi-Layer Perceptrons (MLPs) with ReLU activation functions between hidden layers. The specific architecture implemented in `rl_components/agent_net.py` consists of:

- **Actor (`ActorNet`):** Input(ObsDim) -> Linear(HiddenDim) -> ReLU -> Linear(HiddenDim) -> ReLU -> Linear(ActionDim) -> Output(Logits)
- **Critic (`CriticNetSAC`):** Input(ObsDim) -> Linear(HiddenDim) -> ReLU -> Linear(HiddenDim) -> ReLU -> Linear(ActionDim) -> Output(Q-values per action)

The hidden dimension size is specified in the hyperparameters table below.

#### G.3.4 SAC Hyperparameters (`algorithm_config.yaml`)

The hyperparameters used for training the baseline SAC agents are listed in Table 9.

Table 9: SAC Hyperparameters used for Baseline RL Agent Training.

| Parameter | Value |
|---|---|
| Algorithm Name | SAC |
| Discount Factor ($\gamma$) | 0.99 |
| Temperature ($\alpha$) | 0.01 (Fixed) |
| Actor Learning Rate | 1.0e-4 |
| Critic Learning Rate | 3.0e-4 |
| Optimizer | Adam |
| Batch Size | 512 |
| Target Update ($\tau$) | 0.005 |
| Replay Buffer Size | 100,000 |
| Warmup Steps | 1,000 |
| Total Training Steps | 10,000,000 |
| Update Frequency | 1 |
| Policy Update Frequency | 2 |
| Network Hidden Dimension | 64 |
| Max Tasks per Batch | 400 |
| Device | Auto |

### G.3.5   Evaluation Variants

The different RL results in Table 3 were obtained by evaluating the same trained SAC agent but modifying its action selection during the evaluation run:

- *RL (Geo+Time):* Used the policy directly, sampling from the full action space $\{0, \ldots, N\}$.
- *RL (Geo Only):* If the policy sampled action 0 (Defer), the action was discarded and re-sampled from $\{1, \ldots, N\}$ only.
- *RL (Time Only):* If the policy sampled action $j$ for a task originating at DC $k$, the action was overridden to be $k + 1$ (assign to origin) unless $j = 0$. Only actions 0 and $k + 1$ were permitted.

### G.4   Ray RLlib Integration Details

The agents trained using Ray RLlib [18] (PPO, APPO, and IMPALA), whose performances are reported in Table 13 of the main paper, leverage a specific "single action mode" within the DCcluster-Opt environment. This adaptation simplifies the observation and action spaces, making the environment directly compatible with standard RLlib model architectures (e.g., MLPs) without requiring custom models for handling variable-length sequences of per-task data. This section details this adapted setup. Example training scripts, such as `train_rllib_ppo.py`, configure and utilize this mode.

### G.4.1   Environment Configuration for Single Action Mode

To enable this mode for RLlib training, the `sim_config.yaml` file passed to the environment creator must include the flag `single_action_mode: true`. The `TaskSchedulingEnv` constructor checks this flag and accordingly modifies its observation and action space definitions. An additional flag, `disable_defer_action: true/false` (default false), can further restrict the action space by removing the global deferral option.

### G.4.2   Observation Space: Aggregated Global State

When `single_action_mode` is enabled, the `TaskSchedulingEnv` (via its `_get_obs` method calling `_aggregate_task_observations`) computes and returns a **single, fixed-size observation vector** $s_t$ at each timestep $t$. This vector is an aggregated summary of the entire system state, including:

- Global time features (4: sine/cosine of hour and day of year).
- Aggregated statistics of the $k_t$ currently pending tasks (5 features by default: total number of tasks, average CPU requirement, average GPU requirement, average estimated duration,

and minimum time-to-SLA across all pending tasks). The exact features and their number (`num_aggregated_task_features`) can be customized.

- Current status of all $N$ data centers ($5 \times N$ features: average resource availability for CPU/GPU/Memory, current electricity price, and current grid carbon intensity for each DC).

The dimension of this fixed-size aggregated observation vector is $4 + \text{num\_aggregated\_task\_features} + (5 \times N)$. This allows standard Multi-Layer Perceptron (MLP) models within RLlib to be used directly.

### G.4.3 Action Space: Single Global Discrete Action for All Pending Tasks

In "single action mode," the RLlib agent learns to output a **single discrete action** $a_t$ at each timestep. This single integer action is then uniformly applied to *all* $k_t$ tasks that are currently pending in `self.current_tasks` within that timestep (after an initial SLA check for each task). The action space is `gym.spaces.Discrete(M)`, where:

- If `disable_defer_action: false` (default): $M = N + 1$.

    - $a_t = 0$: **Defer all** $k_t$ pending tasks to the next timestep.
    - $a_t = j \in \{1, \ldots, N\}$: **Assign all** $k_t$ pending tasks to data center $j$.

- If `disable_defer_action: true`: $M = N$.

    - $a_t = j' \in \{0, \ldots, N-1\}$: **Assign all** $k_t$ pending tasks to data center $j' + 1$. (The agent outputs $0..N-1$, which is mapped to DCs $1..N$).

This means the agent learns a global strategy for the current batch of tasks (e.g., "this is a good time to defer everything" or "conditions favor sending everything to DC2 right now"). The environment's `step()` method then iterates through `self.current_tasks` and applies this single decision to each task.

### G.4.4 Network Architecture and Training (RLlib Agents)

With fixed-size observation and action spaces under this mode, standard MLP architectures provided by RLlib (e.g., configured via `model["fcnet_hiddens"]`) are used for the policy and value function networks within PPO, APPO, and IMPALA. The agents are trained using Ray Tune. The global reward signal $r_t$ (calculated based on the outcome of applying the single chosen strategy to all tasks) is used for training.

### G.4.5 Hyperparameters for RLlib Agents (Single Action Mode)

The key hyperparameters for training the PPO, APPO, and IMPALA agents using this aggregated state/single action paradigm are specified in their respective configuration files in `configs/rllib/` (i.e., `ppo_config.yaml`, `appo_config.yaml`, `impala_config.yaml`).

**Hyperparameters:** The key hyperparameters for each RLlib algorithm are detailed in Table 10, Table 11, and Table 12. These were specified in their respective YAML configuration files (e.g., `configs/rllib/***_config.yaml`). All agents were trained for approximately 10 million total environment timesteps.

### G.4.6 Implications and Usage

This "single action mode" significantly simplifies the learning problem for the RL agent by abstracting the per-task micro-management into a single global decision based on an aggregated view of the system. It allows the straightforward application of standard RLlib algorithms. Users wishing to train agents that make independent decisions for each task (as with the custom SAC example) would need to disable `single_action_mode` and implement custom RLlib models capable of handling list inputs and outputs, typically via padding and masking (as generally discussed in Appendix F.2.3). Our provided example scripts like `train_rllib_ppo.py` demonstrate the setup for this "single action mode."

Table 10: Key Hyperparameters for PPO (RLlib) Agent.

| Parameter | Value |
|---|---|
| *Rollout Configuration* | |
| Num. Env Runners | 16 |
| CPUs per Env Runner | 1 |
| Train Batch Size | 1024 |
| *PPO Algorithm Specifics* | |
| Learning Rate (LR) | 1e-4 |
| Discount Factor ($\gamma$) | 0.99 |
| GAE Lambda ($\lambda$) | 0.95 |
| KL Coefficient | 0.2 |
| Value Function Loss Coeff | 0.5 |
| Entropy Coefficient | 0.01 |
| PPO Clip Parameter | 0.2 |
| Num. SGD Iterations | 3 |
| SGD Minibatch Size | 128 |
| *Stopping Condition* | |
| Total Timesteps (approx.) | ~10M |

Table 11: Key Hyperparameters for APPO (RLlib) Agent.

| Parameter | Value |
|---|---|
| *Rollout Configuration* | |
| Num. Env Runners | 16 |
| CPUs per Env Runner | 1 |
| Rollout Fragment Length | 50 |
| *APPO Algorithm Specifics* | |
| Learning Rate (LR) | 1e-4 |
| Discount Factor ($\gamma$) | 0.99 |
| GAE Lambda ($\lambda$) | 1.0 |
| Value Function Loss Coeff | 0.5 |
| Entropy Coefficient | 0.01 |
| V-trace Enabled | True |
| V-trace $\rho$ Clip Thresholds | 1.0 (both) |
| Train Batch Size (per SGD step) | 512 |
| Learner Queue Size | 16 |
| *Stopping Condition* | |
| Total Timesteps (approx.) | ~10M |

Table 12: Key Hyperparameters for IMPALA (RLlib) Agent.

| Parameter | Value |
|---|---|
| *Rollout Configuration* | |
| Num. Workers (Env Runners) | 16 |
| CPUs per Env Runner | 1 |
| Rollout Fragment Length | 128 |
| *IMPALA Algorithm Specifics* | |
| Learning Rate (LR) | 5e-5 |
| Discount Factor ($\gamma$) | 0.99 |
| GAE Lambda ($\lambda$) | 1.0 (for V-trace) |
| Value Function Loss Coeff | 0.5 |
| Entropy Coefficient | 0.01 |
| V-trace Enabled | True |
| V-trace $\rho$ Clip Thresholds | 1.0 (both) |
| Train Batch Size (per SGD step) | 1024 |
| Num. Data Loader Buffers | 1 |
| *Stopping Condition* | |
| Total Timesteps (approx.) | ~10M |

## G.5 Ray RLlib Additional Results

DCcluster-Opt's utility as a testbed for a broader range of state-of-the-art RL algorithms is also evidenced by results from agents trained using the Ray RLlib framework [18]. Table 13 presents the performance of PPO, APPO, and IMPALA agents, all configured with full geo-temporal scheduling capabilities (i.e., ability to assign tasks to any data center and to defer tasks). These results are compared against our custom SAC (Geo+Time) implementation. All agents were trained using the same default multi-objective reward function (detailed in Appendix F.4). Specific hyperparameters for each RLlib agent (PPO, APPO, IMPALA) are provided directly in their respective configuration files within our codebase (e.g., `configs/rllib/ppo_config.yaml`, `configs/rllib/appo_config.yaml`, `configs/rllib/impala_config.yaml`).

The results indicate that, with the current set of general-purpose hyperparameters, the RLlib PPO agent achieves performance comparable to our tuned custom SAC agent across several key metrics. The APPO and IMPALA agents, while demonstrating the benchmark's compatibility, incurred higher overall costs and environmental impact in this configuration, suggesting that these more sample-efficient or distributed algorithms may require more specific tuning or different exploration strategies to fully leverage their potential within the complex, multi-objective landscape of DCcluster-Opt. This highlights the benchmark's role in facilitating such comparative studies and hyperparameter investigations.

Table 13: Performance of RLlib Agents vs. Custom SAC (Geo+Time Capabilities, Mean ± Std Dev across 10 seeds over 30 days).

| Controller | Total Cost ($) | Total CO2 (t) | Total Energy (MWh) | Total Water (m³) | SLA Viol. (%) | Avg CPU Util (%) | Avg GPU Util (%) | Tx Cost ($) | Tasks Deferred |
|---|---|---|---|---|---|---|---|---|---|
| PPO (RLlib) | 92545±4100 | 309.0±7.5 | 1038.0±2.0 | 7260±65 | 25.50±0.30 | 5.0±0.0 | 6.5±0.3 | 4200±35 | 470±160 |
| APPO (RLlib) | 97021±4300 | 324.1±7.9 | 1089.8±2.2 | 7616±69 | 26.74±0.32 | 4.8±0.0 | 6.2±0.3 | 4413±39 | 398±168 |
| IMPALA (RLlib) | 106261±4700 | 355.0±8.7 | 1193.6±2.4 | 8341±76 | 29.29±0.35 | 4.3±0.0 | 5.5±0.3 | 4833±43 | 245±180 |
| SAC (Geo+Time)* | 92401±4134 | 308.7±7.6 | 1037.9±2.1 | 7253±66 | 25.47±0.30 | 5.0±0.0 | 6.5±0.3 | 4203±37 | 474±165 |

*Custom SAC (Geo+Time) from Table 3 included for reference. All agents were trained with the default multi-objective reward. Lower values are better for all metrics except utilization (%). Tx = Transmission

# H    Additional Evaluation Visualizations

This appendix provides supplementary visualizations to further illustrate the dynamic behavior of the `SAC (Geo+Time)` scheduling agent and the environmental conditions within the DCcluster-Opt benchmark during a representative 30-day evaluation period (using one specific random seed). These time-series plots complement the aggregated results presented in Section 6 of the main paper and show data for each of the 5 simulated data centers unless otherwise specified. All x-axes represent simulation timesteps (15-minute intervals).

## H.1    Time-Varying Environmental Conditions per Datacenter

The figures in this subsection (Figures 13 through 15) depict the dynamic nature of the key external environmental signals that drive the optimization problem. Figure 13 shows the fluctuating electricity prices per kilowatt-hour for each data center, highlighting regional differences and temporal opportunities for cost-aware scheduling. Similarly, Figure 14 illustrates the varying grid carbon intensity, offering chances for carbon-aware task placement. Figure 15 displays the ambient outdoor temperature, which directly influences the energy consumption of each data center's HVAC system. These exogenous factors create a complex, non-stationary environment for the scheduling agent.

## H.2    Agent's Dynamic Workload Management Behavior

The following plots (Figures 16 through 19) illustrate how the `SAC (Geo+Time)` agent manages the incoming workload by distributing tasks across data centers and utilizing temporal deferral. Figure 16 shows the number of new tasks assigned by the agent to each data center at each timestep, reflecting its geographical placement decisions. Figure 17 displays the number of tasks actively being processed at each site. The agent's use of temporal flexibility is shown in Figure 18, which plots the total number of tasks deferred across the entire cluster at each timestep. Finally, Figure 19 depicts the total

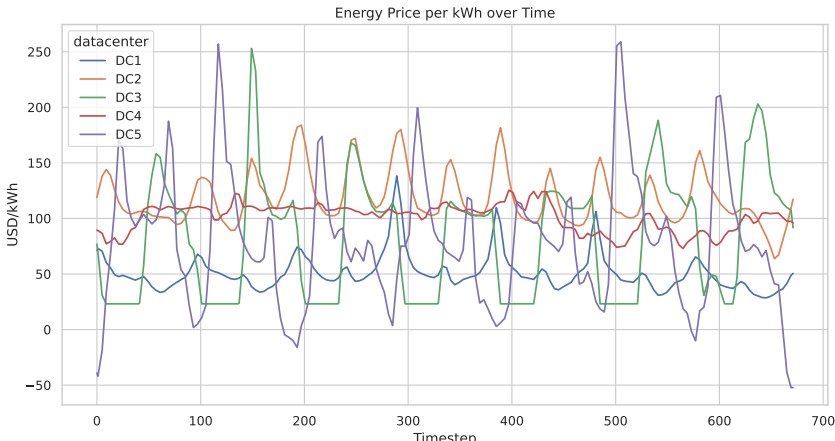

Figure 13: Time-series of Electricity Price ($/kWh) for each data center. Fluctuations highlight opportunities for cost-aware scheduling. DC1: California; DC2: Germany; DC3: Chile; DC4: Singapore; DC5: Australia

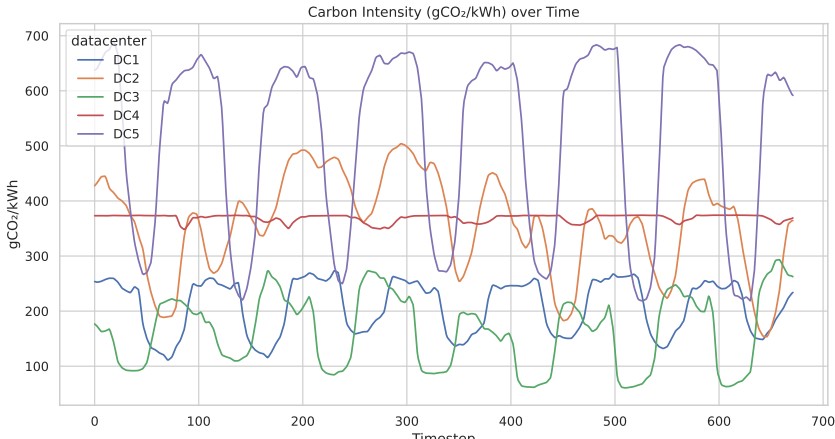

Figure 14: Time-series of Grid Carbon Intensity (gCO$_2$eq/kWh) for each data center. Variations show potential for carbon-aware task placement.

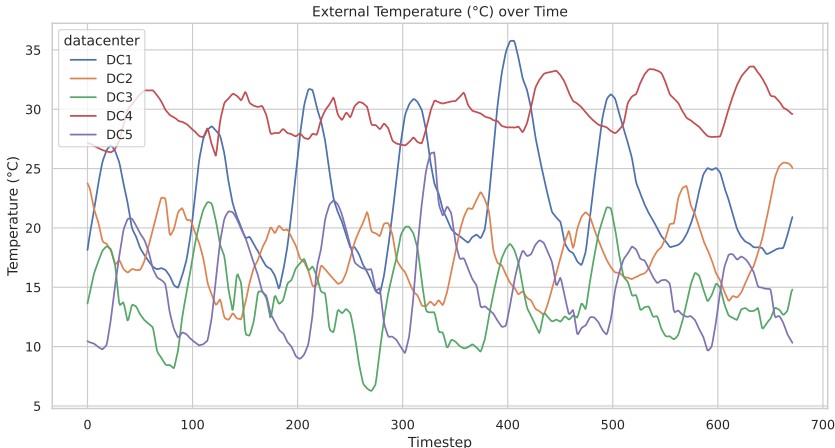

Figure 15: Time-series of External Ambient Temperature (°C) for each data center. This directly impacts HVAC energy consumption.

monetary cost incurred for inter-datacenter data transmissions at each step, a direct consequence of geographical task shifting.

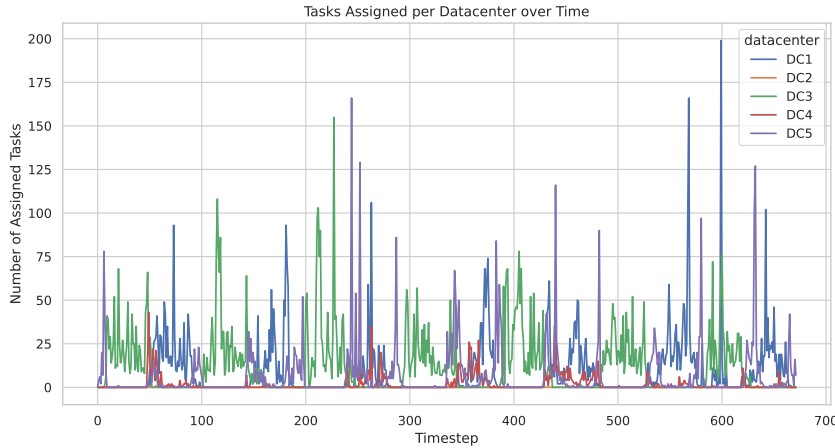

Figure 16: Time-series of the number of new tasks assigned to each data center by the SAC agent at each timestep.

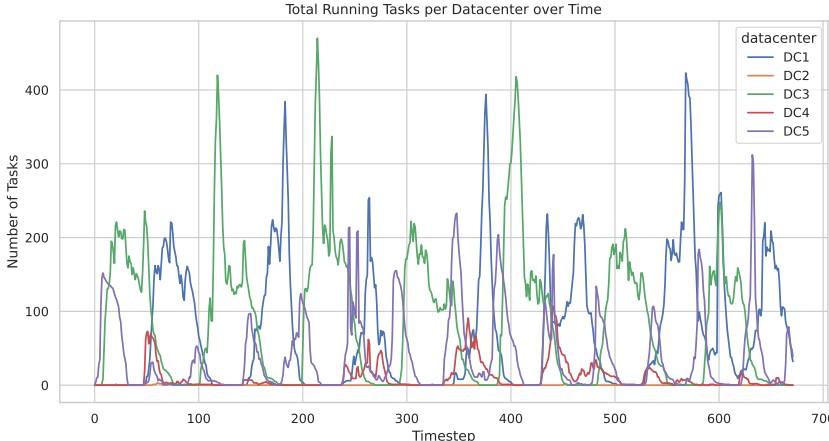

Figure 17: Time-series of the number of tasks actively running in each data center, reflecting the outcome of assignments and task durations.

### H.3 Resulting Datacenter Energy and Carbon Performance Over Time

The agent's scheduling decisions directly impact the sustainability metrics of each data center. Figure 20 shows the energy cost incurred by each data center over time, influenced by both the workload assigned and the prevailing electricity prices. Figure 21 similarly tracks the carbon emissions from each data center, which is a function of its energy consumption and the grid's carbon intensity. These plots allow for analysis of how effectively the agent mitigates costs and emissions at a granular, per-DC level.

### H.4 Resulting Datacenter Operational Metrics Over Time

Beyond sustainability, the agent's actions affect operational efficiency and service quality. Figure 22 provides a multi-panel view of the average CPU, GPU, and Memory utilization percentages for each data center. This indicates how effectively the provisioned hardware resources are being used. High, balanced utilization is often desirable, but sustained maximal utilization can lead to queuing and increased SLA violations.

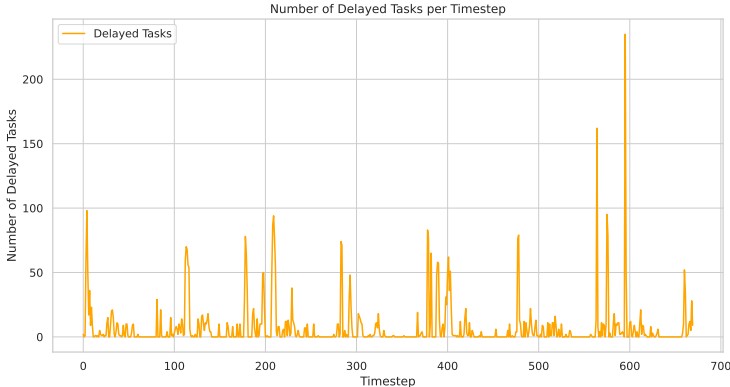

Figure 18: Total number of tasks deferred by the `SAC (Geo+Time)` agent at each timestep across the entire cluster.

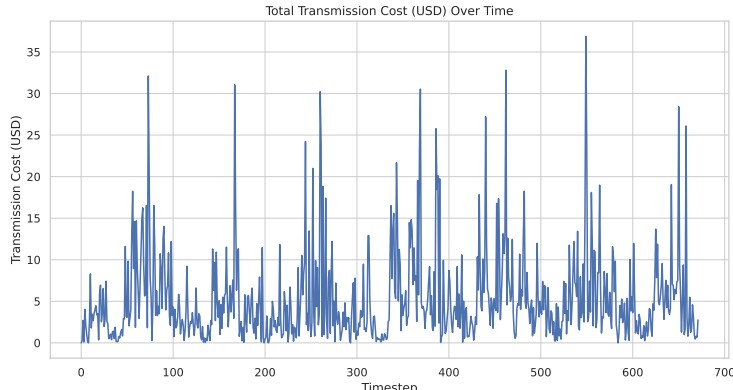

Figure 19: Total inter-datacenter transmission cost ($) incurred by the `SAC (Geo+Time)` agent at each timestep.

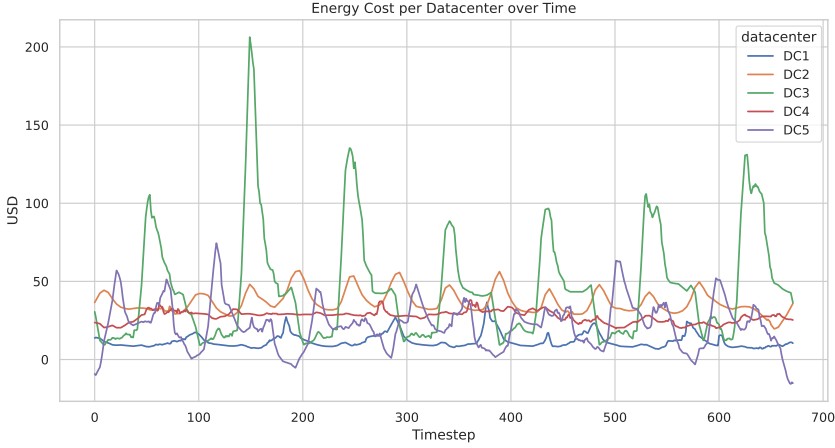

Figure 20: Time-series of Energy Cost ($) incurred by each data center at each timestep.

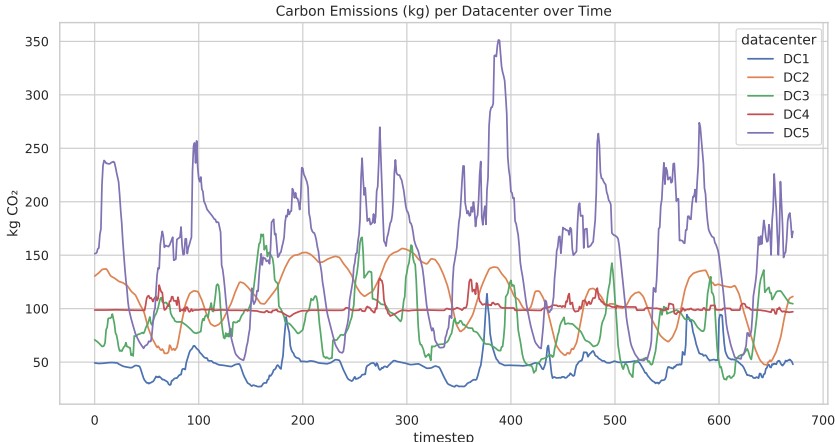

Figure 21: Time-series of Carbon Emissions (kgCO$_2$eq) from each data center at each timestep.

### H.5 Behavioral Comparison: Task Distribution by Local RBC vs. RL Agent

To illustrate the difference in scheduling behavior, we compare the task distribution achieved by the `RBC (Local Only)` controller against our trained `SAC (Geo+Time)` RL agent. Figure 23 first presents the key dynamic environmental factors influencing the RL agent's decisions across the data centers over a representative period (first 5 days) of the simulation.

Figure 24 then shows the corresponding number of tasks concurrently running in each of the five data centers under two distinct scheduling strategies: the `RBC (Local Only)` strategy (where tasks are confined to their origin data center) and the adaptive `SAC (Geo+Time)` RL agent.

As depicted in Figure 24, the `RBC (Local Only)` controller (solid lines) results in a task distribution that directly reflects the probabilistic task origin model for each data center, with load levels fluctuating based on inherent arrival patterns. In contrast, the `SAC (Geo+Time)` RL agent (dashed lines) exhibits significantly different and more dynamic behavior. By correlating these load patterns with the environmental signals shown in Figure 23, it becomes evident that the RL agent actively redistributes tasks. For instance, at timestep 300, one might observe that Chile is the location with the lowest carbon intensity and a low electricity price and external temperature. The RL agent uses this to move tasks to that datacenter. This adaptive scheduling, guided by the multi-objective reward function (detailed in Appendix F.4), demonstrates the agent learning to exploit spatio-temporal variations to optimize the overall objectives defined in the DCcluster-Opt benchmark.

## I   HVAC Control Agent Training Details

The results presented for "SAC (Geo+Time) with RL-Controlled HVAC" and "SAC (Geo+Time) with RL-Controlled HVAC + HRU" in Table 4 (main paper) utilize a local RL agent to dynamically control the HVAC cooling setpoint within each simulated data center. This section details the training setup for this local HVAC controller.

### I.1   Objective and Agent

The local HVAC controller was trained independently for a representative single data center environment (`envs/sustaindc/dc_gym.dc_gymenv`) using the Proximal Policy Optimization (PPO) algorithm [43]. The agent's goal was to learn a policy that minimizes a combination of local data center energy consumption, associated carbon emissions, and energy costs, while maintaining safe operating temperatures (implicitly, by avoiding excessive penalties or by having temperature as part of its observation).

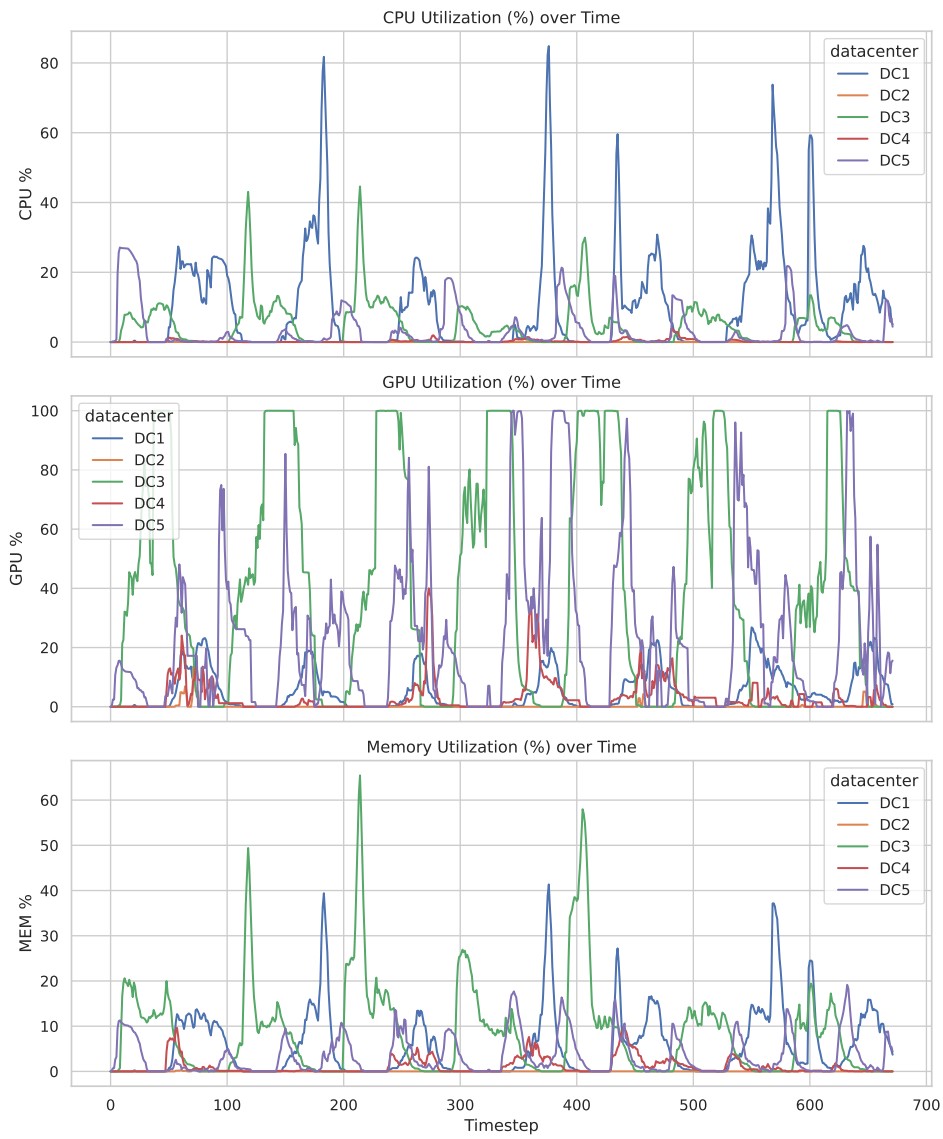

Figure 22: Time-series of average resource utilization per data center: (Top) CPU Utilization (%), (Middle) GPU Utilization (%), (Bottom) Memory Utilization (%).

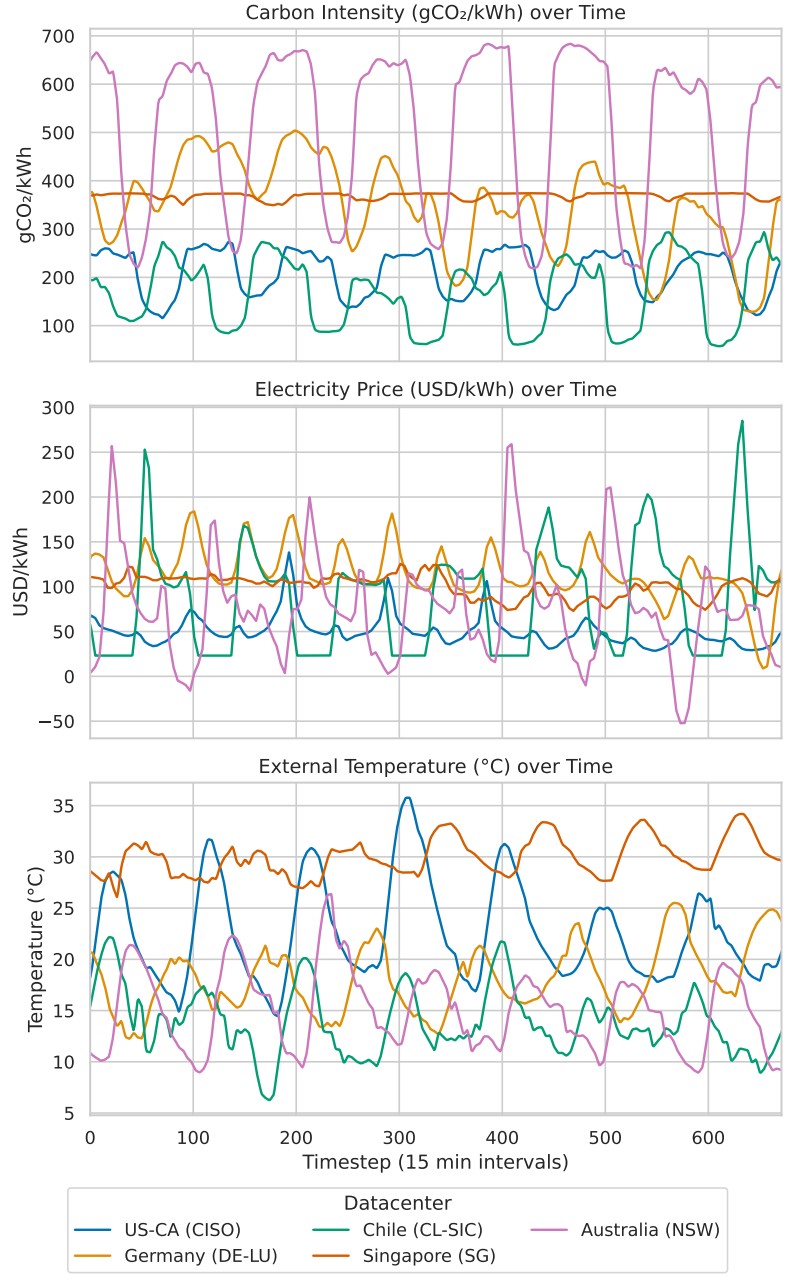

Figure 23: Time-series of key environmental factors for each data center over the first 7 days of simulation: (Top) Grid Carbon Intensity (gCO$_2$eq/kWh), (Middle) Electricity Price ($/kWh), and (Bottom) External Ambient Temperature (°C). These dynamic signals provide context for the RL agent's scheduling decisions.

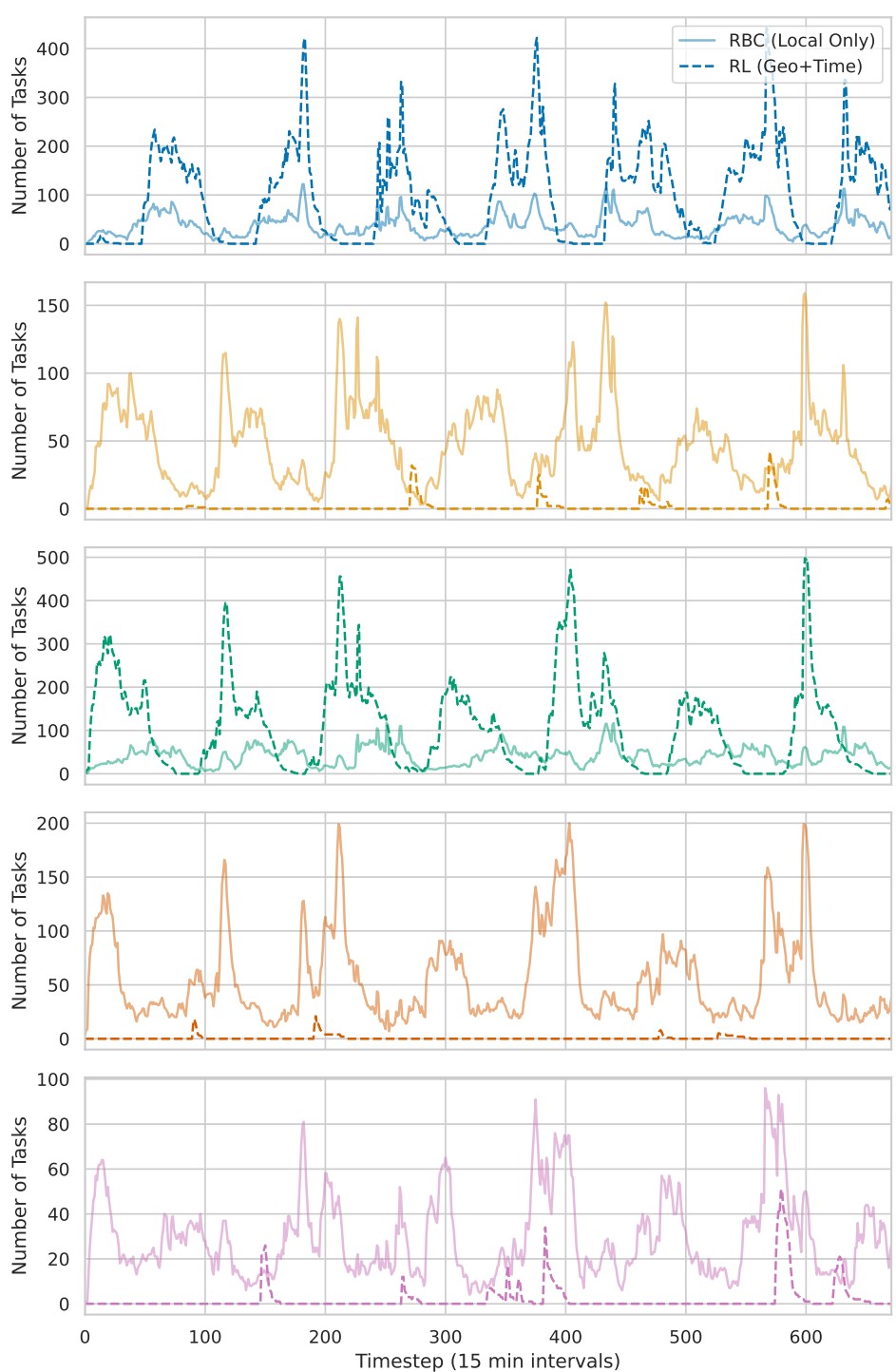

Figure 24: Comparison of the number of actively running tasks (first 5 days) of simulation, shown per data center (California, Germany, Chile, Singapore, and Australia, top to bottom subplot respectively). Within each subplot, the **solid line** represents task execution under the RBC (Local Only) controller, while the **dashed line** represents task execution under the SAC (Geo+Time) RL agent. Differences highlight the RL agent's dynamic load shifting.

## I.2 State Observation Space

The PPO agent for HVAC control observed a fixed-length feature vector representing the local state of its data center and the current time. This observation, constructed at each 15-minute step, consisted of the following 6 features in order:

- **Sine of Hour of Day (sinH):** $\sin(2\pi \times \text{current\_hour}/24.0)$, providing a cyclical representation of the time of day.
- **Cosine of Hour of Day (cosH):** $\cos(2\pi \times \text{current\_hour}/24.0)$, complementing the sine for unambiguous time encoding.
- **Ambient Temperature (ambT):** The current outdoor air temperature (°C) at the data center's location, obtained from the `Weather_Manager`.
- **CPU Load Fraction (cpuL):** The current CPU utilization of the data center, expressed as a fraction (0-1).
- **GPU Load Fraction (gpuL):** The current GPU utilization of the data center, expressed as a fraction (0-1).
- **Previous Setpoint (prevSP):** The CRAC cooling setpoint (°C) that was active in the *previous* timestep (i.e., the outcome of the agent's last action or the initial setpoint).

The resulting feature vector has a dimension of `hvac_obs_dim` = 6. During training, these observations were normalized using running mean and standard deviation statistics (via the `RunningStats` class) before being fed into the policy and value networks. The saved HVAC policy checkpoint includes these normalization statistics to ensure consistent input scaling during deployment within the main DCcluster-Opt simulation.

## I.3 Action Space

The HVAC agent selected one of three discrete actions at each step:

- Action 0: Decrease current CRAC setpoint by 1.0°C.
- Action 1: Maintain current CRAC setpoint (change of 0.0°C).
- Action 2: Increase current CRAC setpoint by 1.0°C.

The resulting absolute setpoint was clipped to a pre-defined valid range (e.g., 18.0°C to 27.0°C).

## I.4 Reward Function

The reward function for training the HVAC PPO agent was designed to penalize high energy consumption. A simplified example is:

$$r_{hvac} = -(P_{TOTAL}/C_{scale}) - P_{boundary} - P_{thermal}$$

where $P_{TOTAL}$ is the total power consumption (kW) of the local DC (extracted from the `info` dictionary), $C_{scale}$ is a scaling factor (e.g., 100). $P_{boundary}$ is a small penalty applied if the chosen setpoint is too close to the operational limits (e.g., 18°C or 27°C). $P_{thermal}$ is an optional larger penalty if internal temperatures (e.g., average rack return air) exceed safe thresholds (e.g., $> 27$°C).

## I.5 Training Environment & Driving Data

The HVAC agent was trained using a single instance of `dc_gymenv`. To ensure generalization, the training environment was driven by:

- **Time-Varying IT Load:** CPU and GPU load profiles used was the same as in the main paper.
- **Real Weather Data:** Ambient temperature and wet-bulb temperature were provided by the `Weather_Manager` using historical data for a representative location (e.g., "US-CAL-CISO" for year 2023, as specified in the HVAC training config).

The training script `train_hvac_ppo_agent.py` orchestrates this setup.

### I.6 PPO Hyperparameters

The key hyperparameters used for training the PPO HVAC agent were specified in `configs/hvac_train_config_ppo.yaml` and are summarized in Table 14.

Table 14: PPO Hyperparameters for Local HVAC Control Agent Training.

| Parameter | Value |
|---|---|
| Algorithm Name | PPO |
| Total Training Steps | 2,000,000 |
| Learning Rate | 3.0e-4 |
| Steps per Rollout (`n_steps`) | 2048 |
| Minibatch Size | 64 |
| PPO Epochs per Rollout | 10] |
| Discount Factor ($\gamma$) | 0.99 |
| GAE Lambda ($\lambda$) | 0.95 |
| Clipping Coefficient (`clip_coef`) | 0.2 |
| Entropy Coefficient (`ent_coef`) | 0.01 |
| Value Function Coefficient (`vf_coef`) | 0.5 |
| Max Gradient Norm | 0.5 |
| Network Hidden Dimension | 64 |
| Optimizer | Adam |
| Observation Dimension (`hvac_obs_dim`) | 6 |

### I.7 Integration

The trained actor network weights, observation dimension, and normalization statistics (if used) from the PPO HVAC agent are saved to a checkpoint file (e.g., `hvac_policy_ppo.pth`). When evaluating the main global scheduler with "RL-Controlled HVAC", each `SustainDC` instance loads this pre-trained policy. At each timestep, it constructs the local HVAC observation, queries the loaded policy for a discrete action (0, 1, or 2), translates this to an absolute CRAC setpoint, and applies it to its internal `dc_gymenv`.

## J Potential Benchmark Extensions and Future Research Directions

### J.1 Extensibility with Additional Control Modules

While the current version of DCcluster-Opt focuses on the core global task scheduling problem coupled with detailed DC physics (IT load, HVAC, and optional HRU), its modular design and underlying `SustainDC` environment (per data center) are architected to support integration with additional local control modules and optimization layers. This opens up avenues for richer research into hierarchical and multi-agent control systems, a paradigm essential for managing the operational complexity of large-scale, federated infrastructure like the American Science Cloud. In such a framework, the global task scheduler acts as a high-level coordinator, making strategic spatio-temporal decisions (e.g., assigning a batch of tasks to a datacenter). In contrast, local, specialized agents within each data center would manage fine-grained, real-time operations (e.g., optimizing the battery discharge rate or reordering the local task queue to minimize power consumption over the next few minutes). This division of responsibility mirrors real-world operational architectures and presents a rich set of research questions in multi-agent coordination and control:

- **Auxiliary Battery Storage Systems:**
  - *Concept:* Integrate a battery model (e.g., `envs/sustaindc/battery_model.py` and `battery_env.py` which exist in the codebase but may need further development for active RL control) within each `SustainDC` instance.
  - *Optimization Problem:* A local RL agent or heuristic controller could manage the battery's charging and discharging cycles.
  - *Objectives for Battery Agent:*
    * *Energy Arbitrage:* Charge during low electricity price periods, discharge during high price periods.

* *Peak Shaving:* Discharge to reduce peak power demand from the grid, potentially lowering demand charges.
* *Renewable Smoothing/Maximization:* Store excess local renewable generation (if modeled) and discharge when renewables are unavailable or grid carbon intensity is high.
* *Grid Services:* (Advanced) Model participation in ancillary grid services.

- *Interaction with Global Scheduler:* The battery's state (SoC, charge/discharge rate) would become part of the local DC state. The global scheduler's decisions (workload placement) would affect the DC's load profile, which in turn impacts optimal battery operation. Conversely, battery availability might influence the global scheduler's perception of a DC's cost or carbon efficiency.
- *Research Questions:* How does coordinated battery control impact overall cost, carbon, and grid stability? What are the optimal MARL or hierarchical control strategies?

- **Local Task Queue Management and Time-Shifting Modules:**

  - *Concept:* While the global scheduler can defer tasks between 15-minute intervals, each `SustainDC` could implement a more sophisticated local queue manager (e.g., `envs/sustaindc/timeloadshifting_env.py` which exists but may need integration/enhancement for active RL control).
  - *Optimization Problem:* A local agent could reorder tasks within its queue, or perform finer-grained intra-DC time-shifting (e.g., delaying a task by a few minutes within the 15-minute global slot) based on short-term local conditions or predictions.
  - *Objectives for Local Queue Agent:*
    * Minimize local energy consumption by aligning execution with micro-fluctuations in cooling efficiency or short-term renewable availability.
    * Improve local resource packing and reduce server idle time.
    * Prioritize tasks closer to their SLA deadlines within the local queue.
  - *Interaction with Global Scheduler:* The global scheduler assigns a batch of tasks to the DC. The local queue manager then optimizes their execution order and precise start times within that global assignment. The efficiency of local management could feed back as part of the DC's observed state.
  - *Research Questions:* How much benefit can local fine-grained time-shifting provide on top of global 15-minute deferral? What are effective local scheduling heuristics or RL policies?

- **Dynamic IT Resource Management (e.g., Power Capping, Server On/Off):**

  - *Concept:* Allow a local agent within each DC to control server power states (e.g., turn servers on/off based on queue length) or apply CPU/GPU power capping.
  - *Objectives:* Reduce idle power, manage thermal load dynamically.
  - *Interaction:* Global scheduler places load; local agent manages the resources to serve that load efficiently.

- **Integration of On-Site Renewable Generation and Forecasting:**

  - *Concept:* While DCcluster-Opt uses grid carbon intensity and price, future extensions could more explicitly model on-site renewable generation (solar, wind) at each DC.
  - *Optimization Problem:* Agents (global or local) would need to consider forecasted renewable generation alongside grid signals.
  - *Interaction:* Would create more complex decision-making, especially when combined with battery storage.

These potential extensions transform DCcluster-Opt into a hierarchical control problem, where the global task scheduler interacts with (or makes assumptions about) local DC controllers. Exploring these interactions is a rich area for future research, and the modularity of the `SustainDC` sub-environment is intended to support such investigations. The codebase already contains stubs or initial versions for some of these local components (e.g., battery, time-load-shifting environments), which can be further developed and integrated for active control.

# K  Code Repository and Maintenance Plan

This section provides details regarding the accessibility of the DCcluster-Opt codebase and our plan for its ongoing maintenance and support, ensuring its continued utility for the research community.

## K.1  Code Repository

The complete source code for the DCcluster-Opt benchmark environment, including all simulation logic, data processing scripts, baseline agent implementations, configuration files, and evaluation notebooks, is publicly available on GitHub under an MIT License:

$$\text{https://github.com/HewlettPackard/sustain-cluster}$$

The repository also includes documentation, which can be found [Specify location, e.g., in the `docs/` folder or hosted on GitHub Pages at `https://hewlettpackard.github.io/sustain-cluster`].

## K.2  Maintenance Plan

We are committed to maintaining and supporting the DCcluster-Opt benchmark to ensure its long-term value and relevance to the research community. Our maintenance plan includes the following aspects:

- **Primary Maintainers and Duration:** The benchmark will be actively maintained by the core authors affiliated with Hewlett Packard Labs. We commit to providing support, addressing issues, and considering updates for at least **three years** following its initial public release, with continued maintenance contingent on ongoing research activities and community engagement.

- **Bug Fixes and Issue Tracking:**
  - Users are encouraged to report bugs, issues, or unexpected behavior via the "Issues" tab on the DCcluster-Opt GitHub repository.
  - We will endeavor to address reported issues in a timely manner. Critical bugs affecting the core functionality or reproducibility of published results will be prioritized.
  - Fixes will be incorporated into the main branch. For significant issues that might alter previously reported baseline results, an errata or changelog will be maintained in the repository (e.g., in the main `README.md` or a dedicated `CHANGELOG.md`).

- **Dataset Updates:**
  - The underlying real-world datasets (electricity prices, carbon intensity, weather data) will be periodically reviewed for updates from their original sources (e.g., Electricity Maps, Open-Meteo).
  - We plan to refresh these datasets (e.g., by adding new historical years) approximately **annually**, subject to the availability and accessibility of updated data from the providers and the stability of their APIs.
  - The core Alibaba workload trace, being historical, will remain static unless significant errors are discovered or a substantially improved public trace becomes available and is deemed suitable for integration.
  - Updates to datasets will be clearly communicated in the repository release notes.

- **Software Compatibility and Dependencies:**
  - The benchmark is developed using Python 3.10 with a defined set of dependencies listed in `requirements.txt`.
  - We will monitor compatibility with major new releases of core dependencies (e.g., Gymnasium, PyTorch, Pandas, NumPy) and aim to address breaking changes through code updates or by specifying compatible version ranges.
  - Major updates to dependencies that necessitate significant code changes might be released as new major versions of DCcluster-Opt.

- **Feature Enhancements and Roadmap:**

- New features and enhancements, as outlined in the roadmap (Section 7 of the main paper), will be considered for implementation based on research priorities, author availability, and community feedback/contributions.
- We welcome suggestions for new features or improvements via GitHub issues.

- **Community Contributions:**
  - We encourage contributions from the research community (e.g., new baseline agents, alternative reward functions, bug fixes, documentation improvements, data for new regions).
  - Contributions can be made via Pull Requests on the GitHub repository.
  - Submitted contributions will be reviewed by the maintainers for quality, relevance, and compatibility before potential integration.

- **Versioning:**
  - DCcluster-Opt will follow a semantic versioning scheme (e.g., v1.0.0, v1.1.0, v2.0.0) for its releases.
  - Major versions may include significant new features or breaking changes to APIs or dataset formats (with clear migration guidance if possible).
  - Minor versions will typically include backward-compatible feature additions or improvements.
  - Patch versions will address bug fixes.
  - Releases will be tagged on GitHub, and release notes will detail changes.

- **Communication:**
  - Primary communication regarding updates, issues, and discussions will occur through the GitHub repository (Issues, Discussions, Pull Requests, Release Notes).
  - Contact information for the corresponding author is provided in the main paper.

This maintenance plan aims to ensure that DCcluster-Opt remains a reliable, up-to-date, and evolving resource for the sustainable computing research community.

# L  License

The DCcluster-Opt codebase is licensed under the MIT License. The full license text is provided below. `MIT License`

`Copyright 2025 Hewlett Packard Enterprise`

`Permission is hereby granted, free of charge, to any person obtaining a`
`copy of this software and associated documentation files (the ``Software''),`
`to deal in the Software without restriction, including without limitation`
`the rights to use, copy, modify, merge, publish, distribute, sublicense,`
`and/or sell copies of the Software, and to permit persons to whom the`
`Software is furnished to do so, subject to the following conditions:`

`The above copyright notice and this permission notice shall be included in`
`all copies or substantial portions of the Software.`

`THE SOFTWARE IS PROVIDED ``AS IS'', WITHOUT WARRANTY OF ANY KIND, EXPRESS OR`
`IMPLIED, INCLUDING BUT NOT LIMITED TO THE WARRANTIES OF MERCHANTABILITY,`
`FITNESS FOR A PARTICULAR PURPOSE AND NONINFRINGEMENT. IN NO EVENT SHALL`
`THE AUTHORS OR COPYRIGHT HOLDERS BE LIABLE FOR ANY CLAIM, DAMAGES OR`
`OTHER LIABILITY, WHETHER IN AN ACTION OF CONTRACT, TORT OR OTHERWISE,`
`ARISING FROM, OUT OF OR IN CONNECTION WITH THE SOFTWARE OR THE USE OR OTHER`
`DEALINGS IN THE SOFTWARE.`

Please ensure attribution to original dataset sources (listed in Section 4 and Appendix D) is maintained when using or distributing derived work.

# M   Ethical Considerations & Broader Impact

The development and release of the DCcluster-Opt benchmark and its associated datasets have been guided by considerations of ethical implications and potential broader impacts on the research community and society.

## M.1   Motivation and Positive Impact

DCcluster-Opt is motivated by the urgent need to address the significant and growing environmental footprint (energy consumption, carbon emissions, water usage) of large-scale AI and data center operations.

- **Facilitating Sustainable AI Research:** By providing a realistic, high-fidelity, and open-source simulation environment, DCcluster-Opt aims to lower the barrier to entry for research into sustainable computing. It enables the development, testing, and fair comparison of novel workload scheduling algorithms, control strategies, and system designs that explicitly optimize for sustainability objectives (e.g., reduced carbon emissions, lower energy costs, minimized water usage) alongside operational performance.
- **Promoting Transparency and Reproducibility:** The open-source nature of the code and the provision of curated real-world (though often anonymized or aggregated) datasets promote transparency and reproducibility in a research area where proprietary systems and data can be barriers.
- **Enabling Sustainable Operation of National-Scale Infrastructure:** The intelligent, multi-objective scheduling agents developed and validated using DCcluster-Opt can serve as foundational control-plane technologies for managing next-generation, federated scientific computing ecosystems. This has a direct application to initiatives like the U.S. Department of Energy's Integrated Research Infrastructure (IRI) and the American Science Cloud (AmSC) [1], helping to ensure these vital national resources are operated in a fiscally and environmentally sustainable manner.
- **Informing Policy and Best Practices:** Insights gained from using DCcluster-Opt could potentially inform best practices for data center operators and cloud providers in managing their infrastructure more sustainably, and could also contribute to discussions on policy measures related to IT energy consumption.
- **Educational Tool:** The benchmark can serve as an educational tool for students and researchers learning about the complexities of data center operations, energy efficiency, and the challenges of sustainable computing.

## M.2   Potential Risks and Negative Impacts (and Mitigations)

- **Misinterpretation or Misuse of Results:**
    - *Risk:* Focusing solely on one objective (e.g., minimizing monetary cost) using DCcluster-Opt could lead to scheduling policies that are detrimental to other sustainability aspects (e.g., increased carbon emissions if clean energy is expensive).
    - *Mitigation:* DCcluster-Opt's design explicitly supports multi-objective optimization through its modular reward system. We encourage and demonstrate evaluation across a comprehensive suite of metrics (cost, carbon, energy, water, SLA) to highlight trade-offs. The documentation and example analyses will emphasize the importance of holistic evaluation.
- **Dataset Bias and Generalizability:**
    - *Risk:* The primary AI workload trace is from a single provider (Alibaba GPU Cluster 2020) and, while extensive, may not perfectly represent all types of AI workloads or user behaviors across all cloud platforms or geographical regions. Policies optimized heavily on this specific trace might not generalize perfectly to entirely different workload profiles. The temporal extension of this trace is also synthetic. Similarly, environmental data (price, carbon, weather) is historical and might not perfectly predict future conditions.
    - *Mitigation:* We are transparent about the sources and preprocessing of all datasets (see Appendix D and datasheets). We provide data for over 20 diverse global regions to

enable studies on regional variations. The benchmark is designed to be extensible, allowing users to integrate their own workload traces or environmental data. We encourage research on robust scheduling and transfer learning using DCcluster-Opt.

- **Computational Cost of Using the Benchmark:**
  - *Risk:* Training complex RL agents on a high-fidelity simulator can be computationally intensive, potentially limiting access for researchers with fewer resources. This also contributes to the carbon footprint of research itself.
  - *Mitigation:* DCcluster-Opt itself is designed to be reasonably efficient for its level of detail. We provide baseline results from simpler rule-based controllers and pre-trained RL agent checkpoints to allow researchers to compare against strong baselines without extensive retraining. The Google Colab notebook aims to improve accessibility. We also document the computational resources used for our experiments (Appendix F.7) and report their estimated carbon footprint.
- **Complexity Barrier:**
  - *Risk:* The inherent complexity of managing geo-distributed DCs with multiple dynamic factors might present a steep learning curve.
  - *Mitigation:* We aim for clear documentation, a well-defined API, and illustrative examples. The provision of simpler rule-based controllers allows users to start with understandable baselines.

## M.3 Data Privacy and Ethics (Source Data)

- **AI Workload Trace:** The core workload is derived from the publicly released Alibaba Cluster Trace 2020 [6]. According to its publishers, this trace was anonymized to protect user privacy and proprietary information. Our processing steps do not attempt to re-identify any individuals or entities.
- **Environmental Data (Price, Carbon, Weather):** This data is sourced from public APIs (Electricity Maps, Open-Meteo) or publicly accessible reports from grid operators and market monitors (GridStatus.io, ISOs). This data describes aggregated system-level characteristics and does not contain personally identifiable information.
- **Transmission Data:** Cost data is based on publicly listed cloud provider rates. Delay parameters are derived from published academic research [16] on aggregate network performance.
- Users of DCcluster-Opt are responsible for adhering to the licenses and terms of use of the original data sources when applicable.

## M.4 Intended Use

DCcluster-Opt is intended for research and educational purposes to advance the field of sustainable computing and intelligent systems management. It is not intended for direct use in production control systems without extensive further validation and adaptation to specific operational environments. Any application of insights or algorithms developed using DCcluster-Opt in real-world scenarios should be done with careful consideration of the specific context, safety, and ethical implications.

By openly discussing these considerations, we hope to encourage responsible use and further development of the DCcluster-Opt benchmark.

# Neurips Check List

1. **Claims**

    Question: Do the main claims made in the abstract and introduction accurately reflect the paper's contributions and scope?

    Answer: [Yes]

    Justification: The abstract and introduction (Section 1) clearly state the claim of proposing SustainCluster as a novel, high-fidelity benchmark environment for sustainable geo-distributed scheduling, and detail its contributions regarding data integration, physics-informed modeling, network awareness, and its utility for research. These claims are substantiated by the descriptions of the benchmark design (Section 3), datasets (Section 4), API, and baseline evaluations (Section 6).

    - The answer NA means that the abstract and introduction do not include the claims made in the paper.
    - The abstract and/or introduction should clearly state the claims made, including the contributions made in the paper and important assumptions and limitations. A No or NA answer to this question will not be perceived well by the reviewers.
    - The claims made should match theoretical and experimental results, and reflect how much the results can be expected to generalize to other settings.
    - It is fine to include aspirational goals as motivation as long as it is clear that these goals are not attained by the paper.

2. **Limitations**

    Question: Does the paper discuss the limitations of the work performed by the authors?

    Answer: [Yes]

    Justification: Section **??** of the main paper includes a subsection dedicated to the current limitations of the SustainCluster benchmark, such as simplifications in network emission modeling, fixed HVAC setpoints by default, and the scope of the initial workload trace.

    Guidelines:

    - The answer NA means that the paper has no limitation while the answer No means that the paper has limitations, but those are not discussed in the paper.
    - The authors are encouraged to create a separate "Limitations" section in their paper.
    - The paper should point out any strong assumptions and how robust the results are to violations of these assumptions (e.g., independence assumptions, noiseless settings, model well-specification, asymptotic approximations only holding locally). The authors should reflect on how these assumptions might be violated in practice and what the implications would be.
    - The authors should reflect on the scope of the claims made, e.g., if the approach was only tested on a few datasets or with a few runs. In general, empirical results often depend on implicit assumptions, which should be articulated.
    - The authors should reflect on the factors that influence the performance of the approach. For example, a facial recognition algorithm may perform poorly when image resolution is low or images are taken in low lighting. Or a speech-to-text system might not be used reliably to provide closed captions for online lectures because it fails to handle technical jargon.
    - The authors should discuss the computational efficiency of the proposed algorithms and how they scale with dataset size.
    - If applicable, the authors should discuss possible limitations of their approach to address problems of privacy and fairness.
    - While the authors might fear that complete honesty about limitations might be used by reviewers as grounds for rejection, a worse outcome might be that reviewers discover limitations that aren't acknowledged in the paper. The authors should use their best judgment and recognize that individual actions in favor of transparency play an important role in developing norms that preserve the integrity of the community. Reviewers will be specifically instructed to not penalize honesty concerning limitations.

3. **Theory assumptions and proofs**

   Question: For each theoretical result, does the paper provide the full set of assumptions and a complete (and correct) proof?

   Answer: [NA]

   Justification: This paper introduces a benchmark environment and associated datasets. It does not present new theoretical results, theorems, or mathematical proofs as its primary contribution. The underlying physics models are based on established literature, which is cited.

   Guidelines:

   - The answer NA means that the paper does not include theoretical results.
   - All the theorems, formulas, and proofs in the paper should be numbered and cross-referenced.
   - All assumptions should be clearly stated or referenced in the statement of any theorems.
   - The proofs can either appear in the main paper or the supplemental material, but if they appear in the supplemental material, the authors are encouraged to provide a short proof sketch to provide intuition.
   - Inversely, any informal proof provided in the core of the paper should be complemented by formal proofs provided in appendix or supplemental material.
   - Theorems and Lemmas that the proof relies upon should be properly referenced.

4. **Experimental result reproducibility**

   Question: Does the paper fully disclose all the information needed to reproduce the main experimental results of the paper to the extent that it affects the main claims and/or conclusions of the paper (regardless of whether the code and data are provided or not)?

   Answer: [Yes]

   Justification: Section 6 and Supplemental document detail the experimental setup, including datacenter configurations, simulation parameters, reward functions used for training RL agents, RL algorithm hyperparameters, and descriptions of the rule-based controllers. The logic for RL ablations is also described, aiming to provide sufficient detail for others to replicate the baseline comparisons.

   Guidelines:

   - The answer NA means that the paper does not include experiments.
   - If the paper includes experiments, a No answer to this question will not be perceived well by the reviewers: Making the paper reproducible is important, regardless of whether the code and data are provided or not.
   - If the contribution is a dataset and/or model, the authors should describe the steps taken to make their results reproducible or verifiable.
   - Depending on the contribution, reproducibility can be accomplished in various ways. For example, if the contribution is a novel architecture, describing the architecture fully might suffice, or if the contribution is a specific model and empirical evaluation, it may be necessary to either make it possible for others to replicate the model with the same dataset, or provide access to the model. In general. releasing code and data is often one good way to accomplish this, but reproducibility can also be provided via detailed instructions for how to replicate the results, access to a hosted model (e.g., in the case of a large language model), releasing of a model checkpoint, or other means that are appropriate to the research performed.
   - While NeurIPS does not require releasing code, the conference does require all submissions to provide some reasonable avenue for reproducibility, which may depend on the nature of the contribution. For example
     (a) If the contribution is primarily a new algorithm, the paper should make it clear how to reproduce that algorithm.
     (b) If the contribution is primarily a new model architecture, the paper should describe the architecture clearly and fully.

(c) If the contribution is a new model (e.g., a large language model), then there should either be a way to access this model for reproducing the results or a way to reproduce the model (e.g., with an open-source dataset or instructions for how to construct the dataset).

(d) We recognize that reproducibility may be tricky in some cases, in which case authors are welcome to describe the particular way they provide for reproducibility. In the case of closed-source models, it may be that access to the model is limited in some way (e.g., to registered users), but it should be possible for other researchers to have some path to reproducing or verifying the results.

5. **Open access to data and code**

Question: Does the paper provide open access to the data and code, with sufficient instructions to faithfully reproduce the main experimental results, as described in supplemental material?

Answer: [Yes]

Justification: The paper provides a URL to the open-source GitHub repository (`https://github.com/HewlettPackard/sustain-cluster`) in Section 1 and Supplemental document. The repository contains the code, configuration files, and processed datasets or scripts to obtain them. The Supplemental document provides installation and setup instructions.

Guidelines:

- The answer NA means that paper does not include experiments requiring code.
- Please see the NeurIPS code and data submission guidelines (`https://nips.cc/public/guides/CodeSubmissionPolicy`) for more details.
- While we encourage the release of code and data, we understand that this might not be possible, so "No" is an acceptable answer. Papers cannot be rejected simply for not including code, unless this is central to the contribution (e.g., for a new open-source benchmark).
- The instructions should contain the exact command and environment needed to run to reproduce the results. See the NeurIPS code and data submission guidelines (`https://nips.cc/public/guides/CodeSubmissionPolicy`) for more details.
- The authors should provide instructions on data access and preparation, including how to access the raw data, preprocessed data, intermediate data, and generated data, etc.
- The authors should provide scripts to reproduce all experimental results for the new proposed method and baselines. If only a subset of experiments are reproducible, they should state which ones are omitted from the script and why.
- At submission time, to preserve anonymity, the authors should release anonymized versions (if applicable).
- Providing as much information as possible in supplemental material (appended to the paper) is recommended, but including URLs to data and code is permitted.

6. **Experimental setting/details**

Question: Does the paper specify all the training and test details (e.g., data splits, hyperparameters, how they were chosen, type of optimizer, etc.) necessary to understand the results?

Answer: [Yes]

Justification: The experimental setup for evaluation is described in Section 6. Detailed RL agent training hyperparameters, optimizer choices, and network architectures are provided in the Supplemental document. The specific reward function used for training the reported RL baselines is also detailed.

Guidelines:

- The answer NA means that the paper does not include experiments.
- The experimental setting should be presented in the core of the paper to a level of detail that is necessary to appreciate the results and make sense of them.
- The full details can be provided either with the code, in Supplemental document, or as supplemental material.

7. **Experiment statistical significance**

   Question: Does the paper report error bars suitably and correctly defined or other appropriate information about the statistical significance of the experiments?

   Answer: [Yes]

   Justification: Tables in Section 6 report results as mean ± standard deviation across multiple (e.g., 10) random seeds for each controller.

   Guidelines:

   - The answer NA means that the paper does not include experiments.
   - The authors should answer "Yes" if the results are accompanied by error bars, confidence intervals, or statistical significance tests, at least for the experiments that support the main claims of the paper.
   - The factors of variability that the error bars are capturing should be clearly stated (for example, train/test split, initialization, random drawing of some parameter, or overall run with given experimental conditions).
   - The method for calculating the error bars should be explained (closed form formula, call to a library function, bootstrap, etc.)
   - The assumptions made should be given (e.g., Normally distributed errors).
   - It should be clear whether the error bar is the standard deviation or the standard error of the mean.
   - It is OK to report 1-sigma error bars, but one should state it. The authors should preferably report a 2-sigma error bar than state that they have a 96% CI, if the hypothesis of Normality of errors is not verified.
   - For asymmetric distributions, the authors should be careful not to show in tables or figures symmetric error bars that would yield results that are out of range (e.g. negative error rates).
   - If error bars are reported in tables or plots, The authors should explain in the text how they were calculated and reference the corresponding figures or tables in the text.

8. **Experiments compute resources**

   Question: For each experiment, does the paper provide sufficient information on the computer resources (type of compute workers, memory, time of execution) needed to reproduce the experiments?

   Answer: [Yes]

   Justification: In the Supplemental document, we provide the details of the compute resources (CPU/GPU types, memory) used for training the baseline RL agents and the approximate training times. Evaluation runs are less computationally intensive and primarily CPU-bound. We also provide the carbon emissions associated with this project..

   Guidelines:

   - The answer NA means that the paper does not include experiments.
   - The paper should indicate the type of compute workers CPU or GPU, internal cluster, or cloud provider, including relevant memory and storage.
   - The paper should provide the amount of compute required for each of the individual experimental runs as well as estimate the total compute.
   - The paper should disclose whether the full research project required more compute than the experiments reported in the paper (e.g., preliminary or failed experiments that didn't make it into the paper).

9. **Code of ethics**

   Question: Does the research conducted in the paper conform, in every respect, with the NeurIPS Code of Ethics https://neurips.cc/public/EthicsGuidelines?

   Answer: [Yes]

   Justification: The research involves the development of a benchmark and datasets using publicly available or permissively licensed data, with the goal of promoting energy-efficient and sustainable AI. No human subjects were involved directly in this benchmark creation, and the work adheres to standard academic integrity.

Guidelines:

- The answer NA means that the authors have not reviewed the NeurIPS Code of Ethics.
- If the authors answer No, they should explain the special circumstances that require a deviation from the Code of Ethics.
- The authors should make sure to preserve anonymity (e.g., if there is a special consideration due to laws or regulations in their jurisdiction).

10. **Broader impacts**

Question: Does the paper discuss both potential positive societal impacts and negative societal impacts of the work performed?

Answer: [Yes]

Justification: The paper (e.g., in Section 1 and Section **??** discusses the positive societal impact of facilitating research into reducing the environmental footprint of AI. Potential negative aspects or limitations, such as dataset biases or the risk of misinterpreting results to justify less optimal practices if not viewed holistically, are implicitly or explicitly discussed.

Guidelines:

- The answer NA means that there is no societal impact of the work performed.
- If the authors answer NA or No, they should explain why their work has no societal impact or why the paper does not address societal impact.
- Examples of negative societal impacts include potential malicious or unintended uses (e.g., disinformation, generating fake profiles, surveillance), fairness considerations (e.g., deployment of technologies that could make decisions that unfairly impact specific groups), privacy considerations, and security considerations.
- The conference expects that many papers will be foundational research and not tied to particular applications, let alone deployments. However, if there is a direct path to any negative applications, the authors should point it out. For example, it is legitimate to point out that an improvement in the quality of generative models could be used to generate deepfakes for disinformation. On the other hand, it is not needed to point out that a generic algorithm for optimizing neural networks could enable people to train models that generate Deepfakes faster.
- The authors should consider possible harms that could arise when the technology is being used as intended and functioning correctly, harms that could arise when the technology is being used as intended but gives incorrect results, and harms following from (intentional or unintentional) misuse of the technology.
- If there are negative societal impacts, the authors could also discuss possible mitigation strategies (e.g., gated release of models, providing defenses in addition to attacks, mechanisms for monitoring misuse, mechanisms to monitor how a system learns from feedback over time, improving the efficiency and accessibility of ML).

11. **Safeguards**

Question: Does the paper describe safeguards that have been put in place for responsible release of data or models that have a high risk for misuse (e.g., pretrained language models, image generators, or scraped datasets)?

Answer: [NA]

Justification: The SustainCluster benchmark itself and the processed datasets (workload traces, environmental data) do not pose a high direct risk for misuse in the manner of generative models or sensitive personal data. The workload trace used is an anonymized, publicly available academic dataset.

Guidelines:

- The answer NA means that the paper poses no such risks.
- Released models that have a high risk for misuse or dual-use should be released with necessary safeguards to allow for controlled use of the model, for example by requiring that users adhere to usage guidelines or restrictions to access the model or implementing safety filters.

- Datasets that have been scraped from the Internet could pose safety risks. The authors should describe how they avoided releasing unsafe images.
- We recognize that providing effective safeguards is challenging, and many papers do not require this, but we encourage authors to take this into account and make a best faith effort.

12. **Licenses for existing assets**

Question: Are the creators or original owners of assets (e.g., code, data, models), used in the paper, properly credited and are the license and terms of use explicitly mentioned and properly respected?

Answer: [Yes] All existing assets used in SustainCluster are properly credited with citations to their original sources throughout the main paper (e.g., Section 1, Section 3, Section 4) and extensively detailed in the Supplementary Material (specifically Section D: Dataset Details, and within individual datasheets in Supplemental D.10).

For each major external dataset component, the Supplementary Material (Supplemental D.10) provides:

- The original source and its citation (e.g., Alibaba Cluster Trace 2020 [6], Electricity Maps [7], Open-Meteo API [9], Persico et al. [16] for network delay parameters, and public pricing information for cloud transmission costs from AWS, GCP, and Azure).
- URLs to the original data sources or relevant documentation where available.
- The specific license or terms of use for each original dataset (e.g., CC BY 4.0 for Open-Meteo and likely the Alibaba trace, ODbL for Electricity Maps database content, API-specific terms for Electricity Maps and GridStatus.io, public information for cloud pricing). We guide users to consult these original terms.
- We specify the versions/timeframes of the data used (e.g., workload trace from 2020, environmental data typically 2020/2021-2024).

The SustainCluster codebase itself, including our data processing scripts and the benchmark framework, is released under the MIT License, as stated in the main paper's footnote, the GitHub repository, and detailed in Supplemental L of the Supplementary Material. The processed/derived datasets distributed as part of SustainCluster are also covered by this MIT license, with the explicit understanding that users must also adhere to the terms of the original underlying data sources. A dedicated section in the Supplementary Material (Supplemental D.11) and the README further outlines licensing considerations for third-party datasets.

Guidelines:

- The answer NA means that the paper does not use existing assets.
- The authors should cite the original paper that produced the code package or dataset.
- The authors should state which version of the asset is used and, if possible, include a URL.
- The name of the license (e.g., CC-BY 4.0) should be included for each asset.
- For scraped data from a particular source (e.g., website), the copyright and terms of service of that source should be provided.
- If assets are released, the license, copyright information, and terms of use in the package should be provided. For popular datasets, `paperswithcode.com/datasets` has curated licenses for some datasets. Their licensing guide can help determine the license of a dataset.
- For existing datasets that are re-packaged, both the original license and the license of the derived asset (if it has changed) should be provided.
- If this information is not available online, the authors are encouraged to reach out to the asset's creators.

13. **New assets**

Question: Are new assets introduced in the paper well documented and is the documentation provided alongside the assets?

Answer: [Yes]

Justification: The primary new asset is the SustainCluster benchmark environment and the integrated, processed forms of the datasets. The paper itself, its Supplemental document (including Datasheets in Supplemental document and the extensive README in the code repository (`https://github.com/HewlettPackard/sustain-cluster`) provide comprehensive documentation.

Guidelines:

- The answer NA means that the paper does not release new assets.
- Researchers should communicate the details of the dataset/code/model as part of their submissions via structured templates. This includes details about training, license, limitations, etc.
- The paper should discuss whether and how consent was obtained from people whose asset is used.
- At submission time, remember to anonymize your assets (if applicable). You can either create an anonymized URL or include an anonymized zip file.

14. **Crowdsourcing and research with human subjects**

Question: For crowdsourcing experiments and research with human subjects, does the paper include the full text of instructions given to participants and screenshots, if applicable, as well as details about compensation (if any)?

Answer: [NA]

Justification: This research does not involve crowdsourcing or direct research with human subjects.

Guidelines:

- The answer NA means that the paper does not involve crowdsourcing nor research with human subjects.
- Including this information in the supplemental material is fine, but if the main contribution of the paper involves human subjects, then as much detail as possible should be included in the main paper.
- According to the NeurIPS Code of Ethics, workers involved in data collection, curation, or other labor should be paid at least the minimum wage in the country of the data collector.

15. **Institutional review board (IRB) approvals or equivalent for research with human subjects**

Question: Does the paper describe potential risks incurred by study participants, whether such risks were disclosed to the subjects, and whether Institutional Review Board (IRB) approvals (or an equivalent approval/review based on the requirements of your country or institution) were obtained?

Answer: [NA]

Justification: This research does not involve direct research with human subjects, and therefore IRB approval was not applicable.

Guidelines:

- The answer NA means that the paper does not involve crowdsourcing nor research with human subjects.
- Depending on the country in which research is conducted, IRB approval (or equivalent) may be required for any human subjects research. If you obtained IRB approval, you should clearly state this in the paper.
- We recognize that the procedures for this may vary significantly between institutions and locations, and we expect authors to adhere to the NeurIPS Code of Ethics and the guidelines for their institution.
- For initial submissions, do not include any information that would break anonymity (if applicable), such as the institution conducting the review.

16. **Declaration of LLM usage**

Question: Does the paper describe the usage of LLMs if it is an important, original, or non-standard component of the core methods in this research? Note that if the LLM is used only for writing, editing, or formatting purposes and does not impact the core methodology, scientific rigorousness, or originality of the research, declaration is not required.

Answer: [No]

Justification: Large Language Models (LLMs) were not used as an important, original, or non-standard component of the core methods or benchmark development in this research.

Guidelines:

- The answer NA means that the core method development in this research does not involve LLMs as any important, original, or non-standard components.
- Please refer to our LLM policy (`https://neurips.cc/Conferences/2025/LLM`) for what should or should not be described.

