# OpenReview forum: "DCcluster-Opt: Benchmarking Dynamic Multi-Objective Optimization for Geo-Distributed Data Center Workloads"
_NeurIPS.cc/2025/Datasets_and_Benchmarks_Track — NeurIPS 2025 Datasets and Benchmarks Track poster_

### Official Review · Reviewer_jzqQ · 2025-06-10

**Rating:** 4
**Confidence:** 4

**Summary:**

Developing and evaluating sustainable scheduling strategies for data centers is practically important.

**Dataset Code Accessibility:**

Yes

**Ethical Considerations:**

No, there are no or only very minor ethics concerns

**Final Justification:**

The authors have addressed my concerns.

**Limitations Weaknesses:**

1. The benchmark adopts fluctuating electricity pricing data. This is an oversimplified model. In many countries (including U.S, China, and Australia), real-time electricity prices are determined by locational marginal pricing and the change in data center load can significantly influence the local electricity price.

2. There lacks discussion or justification on the MDP formulation in Table 1. No detail on the formulated state is provided. It is not clear how the reward can accurately reflect the loss and gain of real-world data centers.

3. The authors are suggested providing more numerical results with various datasets and more SOTA scheduling algorithms, to demonstrate the effectiveness of the developed benchmark.

**Strengths Contributions:**

It is true that there lacks environment or benchmark that realistically captures the complex interplay of time varying environmental factors for data center load scheduling. The potential impact of the developed benchmark is high.

---

> ### Author Rebuttal · Authors · 2025-07-30
>
> We thank you for your time and for providing a thoughtful and detailed review. We are very grateful for your positive assessment of the practical importance of our work and your agreement that there is a lack of realistic benchmarks for sustainable scheduling and that the potential impact of SustainCluster is high.
>
> We have carefully considered the weaknesses you identified. We believe that the core concerns raised are comprehensively addressed within the manuscript and its supplementary materials, and we appreciate this opportunity to clarify and provide direct pointers to the relevant sections. We are confident that these clarifications will demonstrate the technical solidity of our work and show that the reasons for acceptance strongly outweigh the initial concerns.
>
> ### 1. On the Simplification of the Electricity Pricing Model
> We thank you for this expert comment regarding Locational Marginal Prices (LMPs) and the feedback loop between datacenter load and grid prices. While we agree this is a fascinating area of research, we must respectfully assert that our model is not "oversimplified" but rather a deliberate and significant step forward in fidelity for the scheduling domain.
>
> - A Foundation Built on Real-World Data: Our framework integrates actual, time-varying, real-world electricity price data from over 20 global regions (as detailed in Sec. 4.2). This provides an unprecedented level of geo-temporal realism that far exceeds the fixed-price or abstract linear cost models prevalent in related scheduling literature. Our primary goal was to build a benchmark that equips agents to handle the high variance and heterogeneity of existing grid signals.
> - A Deliberate Scoping Choice: Modeling the second-order feedback loop of LMPs, where an agent's actions influence the price, is a distinct and highly complex research problem, often requiring co-simulation with power grid market models. We deliberately scoped our benchmark to first solve the challenge of reacting to dynamic prices before attempting to influence them.
> - Modularity Enables Future Research: Crucially, our benchmark is explicitly designed to enable precisely the kind of advanced research you envision. As described in our architecture (Sec. 3.3 and Appendix D.3), the ElectricityPrice_Manager is a modular component. It can be readily replaced by a dynamic function that models price elasticity based on the datacenter's load, without altering the core scheduling logic or RL agent structure. Therefore, the current design is not a flaw but a foundational strength. To further facilitate this, we will add a new section to our Appendix (and online documentation) explaining exactly how a user can implement their own custom pricing function, using the LMP model as a clear example.
>
> **Action**: We will enhance our discussion in the limitations and future work sections (Sec. 7 and Appendix J.1). We will explicitly acknowledge the LMP feedback loop and frame "price elasticity modeling" as a key research avenue that SustainCluster's modular architecture now makes possible for the community to explore.
>
> ### 2. On the Justification and Details of the MDP Formulation
> We respectfully clarify that the details the reviewer claims are lacking are, in fact, described in significant detail within the manuscript. We apologize if the structure of the paper made this information difficult to locate, and we will take action to make it more prominent.
>
> - State (*s_t*) Formulation: The summary in the introduction is intentionally brief. A complete and explicit breakdown of the state vector provided to the agent is detailed in Section F.2 in the Appendix. This section specifies that the observation for each pending task is a vector concatenated from:
>
>     1. Global Time Features (4 dims): Sine/cosine encodings of the day and hour.
>     2. Task-Specific Features (5 dims): Including origin DC, CPU/GPU requirements, and time-to-SLA.
>     3. Per-Datacenter State Features (5×N dims): For all N datacenters, including available CPU/GPU/Mem %, current carbon intensity, and current electricity price.
>
> The exact vector dimension is also explicitly provided.
>
> - Reward (*r_t*) Function's Realism: The suggestion that it is "not clear how the reward can accurately reflect the loss and gain of real-world data centers" is directly addressed by our methodology. Section F.4, "Reward Function Details," explains that the scalar reward is computed by a configurable function that creates a weighted sum of tangible, real-world Key Performance Indicators (KPIs). These are the exact metrics a datacenter operator uses to measure "loss and gain." The evaluation metrics reported in our results in Table 3, such as Total Operational Cost ($), Total CO2 Emissions (kg), SLA Violation Rate (%), etc., are the direct outputs of the simulation that the reward function synthesizes into a learning signal.
>
> **Action**: To make this critical information impossible to miss, we will replace the brief summary table with a more formal and detailed definition of the MDP in the main body of the paper. This new section will provide the explicit mathematical formulation for the State Space (S), Action Space (A), and Reward Function (R), with direct back-references to the sections where each component is described in full detail.
>
> ### 3. On Providing More Numerical Results and SOTA Algorithms
> The purpose of the experiments in a benchmark paper is not to claim a new state-of-the-art algorithm but to demonstrate the benchmark's utility, discriminatory power, and compatibility with modern tools. Our evaluation was designed to achieve exactly that.
> - Demonstrating Discriminatory Power: Our results in Table 3 clearly show that SustainCluster can differentiate the performance of various strategies. We demonstrate the complex, multi-objective trade-offs the benchmark is designed to explore, showing, for instance, that the "Lowest Price" RBC reduces cost but measurably increases carbon emissions compared to a "Local Only" baseline. This proves the environment is nuanced and presents a meaningful challenge.
> - Compatibility with SOTA Algorithms: The suggestion to include "more SOTA scheduling algorithms" is well-taken, and we have already done so. As stated in Section G.5, "Ray RLlib Additional Results," in addition to our custom SAC implementation, we successfully trained and evaluated agents using the widely adopted Ray RLlib framework, including on-policy SOTA algorithms like PPO, APPO, and IMPALA. These results showcase the benchmark's immediate compatibility and ease of use with standard SOTA libraries.
>
> **Action**: We will make the reference to the successful integration and strong performance of the other SOTA RLlib agents (PPO, APPO, IMPALA) more prominent in the main experimental section. We will also add a new, stronger baseline to our main results table: an advanced greedy heuristic that uses a weighted combination of current carbon, price, and network latency.
>
> We hope these detailed clarifications, with direct pointers to the relevant sections of our manuscript, have resolved your concerns and may help reassess the impact of this benchmark. We believe the points raised are either addressed in detail within the manuscript and supplement or represent valuable future research directions that our benchmark is uniquely positioned to enable. Given your agreement on the high potential impact of this work, we are confident that SustainCluster is a significant, technically solid, and valuable contribution to the community.

---

> > ### Comment · Reviewer_jzqQ · 2025-08-06
> >
> > Thank you for addressing most of my concerns. I am ready to change my score accordingly.

---

> > > ### Author Response · Authors · 2025-08-06
> > >
> > > Thank you again for your positive feedback and for engaging with our rebuttal. We are very pleased that our planned revisions have addressed most of your concerns.
> > >
> > > We just wanted to gently follow up on your comment, as we can still see the initial rating in the system. Could you please let us know if there are any final clarifications we can provide to help you finalize your assessment before the discussion period ends?
> > >
> > > We are, of course, ready to incorporate all the promised changes into the final version to make the paper as strong as possible.

---

### Official Review · Reviewer_gvym · 2025-07-04

**Rating:** 5
**Confidence:** 4

**Summary:**

The authors created a dataset to study AI workload scheduling in data centers and developed a GYM environment simulator that incorporates real AI workload traces, the geographic distribution of data centers, data center physical conditions, weather information, transmission costs, and more. The objectives include minimizing operational costs and energy consumption. The authors demonstrate reinforcement learning training on this simulator. Overall, I enjoyed reading this paper. The construction of this simulator and benchmark will be valuable for future studies on realistic AI workload management.

**Dataset Code Accessibility:**

Yes

**Dataset Code Comments:**

The GitHub repo is very well organized!

**Ethical Considerations:**

No, there are no or only very minor ethics concerns

**Final Justification:**

The authors addressed my concerns, and I kept my score unchanged.

**Limitations Weaknesses:**

1. The AI workload data is from 2020, which misses the opportunity to capture recent trends in LLMs.
2. There is no evaluation of how realistic the simulator is. All simulations are wrong, but it would make sense to develop some evaluations here to examine how good the simulator is.

**Strengths Contributions:**

**Pros:**

1. The simulator encapsulates various realistic considerations, with data calibrated against real-world sources (which is very creative!)
2. The study of RL benchmarks on this simulator is thorough and well-presented, revealing multiple insights, including how much the benefit is from advanced local DC control.
3. The paper is well written and easy to follow
4. I like the paradigm presented in this paper: to study a hard operations problem, first construct a realistic simulator (as much as possible) and then play with it. This pipeline can be adopted by many different fields.

---

> ### Author Rebuttal · Authors · 2025-07-30
>
> We are incredibly grateful for your thoughtful review. Thank you for the highly positive feedback and for noting that you enjoyed reading the paper. We are especially pleased that you recognize the value of the simulator's realistic, data-calibrated components and the insights from our benchmark evaluations. Your encouragement of the overall research paradigm is deeply appreciated.
> We agree with the limitations you've identified and would like to offer the following clarifications and planned improvements to address them.
>
> ## Clarifications and Planned Improvements
> ### 1. AI Workload Data
> We agree that workload characteristics are rapidly evolving with the rise of large-scale LLMs, and this is a fundamental challenge for any benchmark relying on static, public datasets. The Alibaba 2020 trace was chosen as it is one of the most well-cited, large-scale, and detailed publicly available datasets for heterogeneous GPU clusters. Also, as part of the ExaDigiT supercomputing consortium, we have started looking into multiple workloads and will incorporate them as reference options.
>
> The core contribution of our work is that the SustainCluster testbed is designed to be workload-agnostic. The benchmark is not fundamentally tied to this specific trace; its data pipeline can readily ingest any workload data that specifies resource requirements, duration, and dependencies. This modularity is key to the benchmark's long-term utility. As newer, more representative public traces (e.g., of LLM training or inference at scale) become available, they can be readily integrated into the benchmark to ensure its long-term relevance.
>
> **Action**: We will expand our discussion of limitations in Section 7 to more explicitly address this. We will emphasize that the framework is designed as a long-term, extensible tool for the community, capable of incorporating newer and more representative workload traces, such as those from LLM training and inference, as they become publicly available. This ensures SustainCluster can evolve in lockstep with industry trends.
>
> ### 2. Evaluation of the Simulator's Realism.
> This is an excellent and insightful point. We agree that evaluating the fidelity of a simulator is crucial. Direct, end-to-end validation of a simulator against a real, globally distributed cluster of data centers is often logistically and financially intractable, which is precisely the motivation for creating a high-fidelity benchmark like SustainCluster in the first place.
>
> Our approach, therefore, was to ensure realism through component-wise validation and grounding in empirical data. Rather than building a monolithic model and validating it at the end, we constructed SustainCluster from modules that are each independently tied to reality:
>
> -	Physics-Informed DC Models: Our data center models, particularly for HVAC, are not arbitrary. They are based on established engineering principles and performance curves derived from validated tools like EnergyPlus, which is an industry standard for building energy simulation.
>
> -	Real-World Data Integration: The simulator is driven by curated, real-world datasets for every key external factor: a real AI workload trace from Alibaba, real time-series electricity prices and carbon intensity from sources like Electricity Maps, real weather data from Open-Meteo, and network parameters from empirical studies.
>
> The realism of the benchmark emerges from the high-fidelity integration of these individually grounded components. The experimental results, which demonstrate intuitive and complex trade-offs (e.g., how a "Lowest Price" strategy increases carbon emissions), serve as a form of qualitative validation that the simulator captures expected real-world dynamics.
>
> **Action**: We will add a new paragraph to Section 7 (Discussion) to explicitly address our validation strategy. We will clarify that while holistic validation is prohibitively complex, we ensured realism through this rigorous, component-wise grounding in empirical data and established physics models.
>
> Once again, thank you for your strong support and constructive feedback. Your comments will help us significantly improve the clarity and impact of our paper.

---

### Official Review · Reviewer_4Cgu · 2025-07-07

**Rating:** 5
**Confidence:** 3

**Summary:**

The authors introduce SustainCluster a composite discrete time dataset for modeling the workload job allocation across multiple spatially distributed datacenters by combining data from existing: AI workload traces, electricity prices, carbon intensity, weather and transmission datasets. The dataset allows for evaluation of discrete-time controllers which load shift workloads across datacenters to minimize overall cost, measured with multiple target objectives, including: carbon emissions, water use, SLA satisfaction, cost, or hardware/software performance. The dataset provides integrations with existing reinforcement learning environments, and reports the performance of baseline controllers.

**Dataset Code Accessibility:**

Yes

**Dataset Code Comments:**

Available on github. Documentation provided in supplementary.

**Ethical Considerations:**

No, there are no or only very minor ethics concerns

**Final Justification:**

See comment

**Limitations Weaknesses:**

## Weaknesses
1. *Limited comparison with existing datasets and related work*. Authors do not differentiate from existing datacenter workload datasets [1,2,3]
2. *Simplified Model of Datacenter Energy Use (Appx B.1).* Datacenter power and water efficiency (PUE, WUE) is known to vary across facilities. Do the authors have intuition of how these factors may vary across facilities (and how to incorporate that into the dataset), outside of the effects of ambient air temperature on cooling?
3. *Workloads may be out-of-date*. Reliance on Alibaba 2020 traces may not be representative of scaling associated with current larger-scale LLM workloads, or changes in the power efficiency of compute hardware.
4. See additional questions below.

## Questions.
1. How is GPU utilization and the resulting energy use determined? Power draw does not vary directly with GPU-hrs, and varies with compute/memory-intensity of the workload [4].
2. How was continuation of workload traces conducted?  Does exclusion of sub 15m discount on-demand inference workloads? How do authors expect this to vary as LLM workloads change?

## References
1. Samsi, Siddharth, et al. "The mit supercloud dataset." 2021 IEEE High Performance Extreme Computing Conference (HPEC). IEEE, 2021.
2. Cortez, Eli, et al. "Resource central: Understanding and predicting workloads for improved resource management in large cloud platforms." Proceedings of the 26th Symposium on Operating Systems Principles. 2017.
3. https://github.com/Azure/AzurePublicDataset
4. Patel, Pratyush, et al. "Characterizing power management opportunities for llms in the cloud." Proceedings of the 29th ACM International Conference on Architectural Support for Programming Languages and Operating Systems, Volume 3. 2024.

**Strengths Contributions:**

1. *Multi-Objective Evaluation.* The selected metrics for evaluation are representative of considerations of datacenter performance and sustainability considerations (energy use, carbon emissions, costs, and workload performance). The proposed dataset preprocesses existing datasets
2. *Grounded Models of Data Center Usage.* The dataset provides realistic grounding of the effects of geographic and temporal load shifting by constructing  hardware and physics-informed models of resource utilizations.
3.  *Wide variety of evaluated baseline controllers*. The lack of an optimal controller for workload dispatch  (and suboptimal performance of single-objective controllers), suggests that the scheduling task may be a challenging testbed for new algorithms, models.
4. Paper is well written; supplementary material and dataset repository are well documented (e.g. Croissant metadata is provided for dataset.

---

> ### Author Rebuttal · Authors · 2025-07-30
>
> Thank you for your thorough and positive review of our work. We are extremely encouraged by your review and your praise for the paper's contributions, particularly the multi-objective evaluation, grounded datacenter models, and comprehensive documentation. Your insightful feedback and questions will be invaluable in further strengthening the final version of our manuscript.
>
> We provide the following clarifications and planned improvements in response to the weaknesses and questions you raised.
>
> ## Clarifications and Planned Improvements
> ### 1. Comparison with Existing Datasets and Related Work:
> Thank you for pointing out the need for clearer differentiation from other important datacenter workload datasets. You are correct that resources like the MIT Supercloud dataset, the analysis by Cortez et al., and the Azure Public Dataset repository are cornerstones of datacenter research.
>
> Our intention was not to diminish their importance but to highlight SustainCluster's unique contribution. These excellent resources primarily provide workload traces, which are a single (though critical) input to our system. The novelty of SustainCluster lies in its role as an end-to-end, integrated simulation environment designed specifically for the sustainable geo-distributed scheduling problem. Its core contribution is the synthesis of:
>
> -	A real-world AI/GPU workload trace.
> -	Dynamic, real-world, geo-temporal data for carbon intensity, electricity price, and weather.
> -	Physics-informed energy models for both IT and, crucially, HVAC systems.
> -	Realistic network cost and latency models.
>
> **Action**: We will revise Section 2 (Related Work) to explicitly cite the datasets you mentioned. We will clarify that while these resources provide invaluable workload traces, SustainCluster's unique contribution is its comprehensive framework that integrates such traces with a multitude of other dynamic signals to create a high-fidelity testbed for evaluating sustainable scheduling algorithms.
>
> ### 2. Simplified Model of Datacenter Energy Use (PUE/WUE)
> This is an excellent point. Power and Water Usage Effectiveness (PUE/WUE) are indeed influenced by many factors beyond ambient temperature, such as facility design, cooling system technology, and hardware efficiency.
>
> Our model is designed to capture this. Rather than using a single, fixed PUE/WUE value, we employ a physics-informed model where the overall efficiency emerges from the interactions of its components. As noted in Section 3.4, our HVAC model simulates specific components like CRAC units, chillers (using performance curves derived from EnergyPlus), and cooling towers. Crucially, the detailed physical parameters for each data center's components are defined in separate configuration files (dc_config.json).
>
> This modular configuration is precisely how facility-specific efficiencies can be modelled. A user can configure one datacenter with a highly efficient chiller model and another with an older, less efficient one, directly impacting their operational PUE and WUE independent of weather.
>
> **Action**: We will clarify in Section 3.4 that the per-datacenter configuration files allow for modelling facility-specific hardware and design efficiencies, ensuring that variations in PUE/WUE across the cluster can be captured beyond the effects of ambient temperature.
>
> ### 3. Out-of-Date Workloads
> We agree that workload characteristics are rapidly evolving with the rise of large-scale LLMs, and this is a fundamental challenge for any benchmark relying on static, public datasets. The Alibaba 2020 trace was chosen as it is one of the most well-cited, large-scale, and detailed publicly available GPU cluster traces. Also, as part of the ExaDigiT supercomputing consortium, we have started looking into multiple workloads and will incorporate them as reference options.
>
> Also note that the SustainCluster framework is designed to be workload-agnostic. The data processing pipeline can be adapted to ingest any task trace that specifies resource requirements and duration. As newer, more representative public traces (e.g., of LLM training or inference at scale) become available, they can be readily integrated into the benchmark to ensure its long-term relevance.
>
> **Action**: We will expand our discussion of limitations in Section 7 to explicitly state that the framework is trace-agnostic and designed to support future, more current workload datasets as they are released by the community.
>
> ## Responses to Your Questions
> ### 1. How is GPU utilization and the resulting energy use determined?
> This is a critical question about the fidelity of our IT power model. You are correct that power draw is complex and not simply linear with GPU-hours. Our model accounts for this by calculating IT power based on both static power consumption and a dynamic component that varies with load. The models for IT components (CPU, GPU, Memory) are based on established approaches in the literature. Specifically, the CPU/GPU power draw is based on work by [1, 2, 6], memory power is informed by [7], and the HVAC/cooling dynamics are derived from foundational models in [3, 4, 5]. We will add these key citations directly into Section 3.4 to make this grounding explicit. These models capture the non-linear relationship between utilization and power draw.
> ### References used to model different parts of the datacenters and included in the datacenter model:
>
> [1]: Postema, Björn Frits. "Energy-efficient data centres: model-based analysis of power-performance trade-offs." (2018).
>
> [2]: Raghunathan, S., & Vk, M. (2014). Power management using dynamic power state transitions and dynamic voltage frequency scaling controls in virtualized server clusters. Turkish Journal of Electrical Engineering and Computer Sciences, 24(4). doi: 10.3906/elk-1403-264
>
> [3]: Sun, Kaiyu, et al. "Prototype energy models for data centers." Energy and Buildings 231 (2021): 110603.
>
> [4]: Breen, Thomas J., et al. "From chip to cooling tower data center modeling: Part I influence of server inlet temperature and temperature rise across cabinet." 2010 12th IEEE Intersociety Conference on Thermal and Thermomechanical Phenomena in Electronic Systems. IEEE, 2010.
>
> [5]: https://h2ocooling.com/blog/look-cooling-tower-fan-efficiences/#:~:text=The%20tower%20has%20been%20designed,of%200.42%20inches%20of%20water.
>
> [6]: X. Tang and Z. Fu, "CPU–GPU Utilization Aware Energy-Efficient Scheduling Algorithm on Heterogeneous Computing Systems," in IEEE Access, vol. 8, pp. 58948-58958, 2020, doi: 10.1109/ACCESS.2020.2982956.
>
> [7]: Seunghak Lee, Ki-Dong Kang, Hwanjun Lee, Hyungwon Park, Younghoon Son, Nam Sung Kim, and Daehoon Kim. 2021. GreenDIMM: OS-assisted DRAM Power Management for DRAM with a Sub-array Granularity Power-Down State. In MICRO-54: 54th Annual IEEE/ACM International Symposium on Microarchitecture (MICRO '21). Association for Computing Machinery, New York, NY, USA, 131–142. https://doi.org/10.1145/3466752.3480089.
>
> **Action**: In Section 3.4, we will add clarification for this modeling approach for IT power and explicitly cite the relevant literature to provide clearer context on the power consumption models used.
>
> ### 2. How was continuation of workload traces conducted? Does the exclusion of sub-15m jobs discount on-demand inference workloads?
>
> The 15-minute timestep is a deliberate design choice for the benchmark, as it aligns with two real-world constraints:
>
> -	Data Availability: Key environmental signals, such as electricity prices and grid carbon intensity, are often reported at 5, 15, or 60-minute intervals.
> -	Physical Inertia: Large-scale data center thermal systems (i.e., HVAC) have significant inertia and do not respond to load changes instantaneously. A 15-minute operational cadence is realistic for strategic, system-level control.
>
> Excluding sub-15-minute jobs is a direct consequence of this chosen temporal resolution. We agree this might not fully capture the behavior of very short, on-demand inference workloads. The continuation of the trace to a full year was performed using statistical methods to preserve the key characteristics of the original trace, as detailed in the Supplementary Material (Sec. D.2). As for LLM workloads, if the trend moves towards sustained training or batch inference, our model remains highly relevant. If it moves towards high-frequency, short-duration tasks, a finer-grained simulation would be needed, which would in turn require higher-frequency data for all environmental inputs.
>
> **Action**: We will enhance the justification for the 15-minute timestep in Section 3.3. In our future work discussion (Section 7) and in our benchmark, we will add the enablement of finer-grained control options and support high-frequency datasets for all input streams as they become more widely available.
>
> Once again, we thank you for your supportive review and the valuable suggestions for improvement. We are confident that incorporating these changes will make our paper even stronger.

---

> > ### Comment · Reviewer_4Cgu · 2025-08-06
> >
> > Thanks to the authors for their comprehensive response and clarifications, which address most of my stated concerns. While I would like to see the proposed dataset extended to current large-scale ML workloads, the work provides more than sufficient contribution in its current form.  I will be maintaining my existing score.

---

### Note · Authors · 2025-08-13

Dear Area Chair,

Through our work on SustainCluster, we have laid a foundation for a multi-objective sustainable data center cluster benchmark to advance ML solutions that reduce the environmental impact of AI.

`SustainCluster` introduces the first high-fidelity, open-source benchmark to holistically model the global **geographical and temporal** datacenter scheduling problem. Its primary contribution, appreciated by the reviewers, is the unique and complex integration of four crucial elements at a global scale:
*   Real-world AI/GPU workload traces.
*   Dynamic, geo-temporal environmental data (carbon intensity, electricity price, weather) for over 20 international regions.
*   Physics-informed energy models for both IT components and, critically, the complex HVAC systems they depend on.
*   Realistic network models capturing both monetary cost and latency.

This work moves the community beyond simplified or siloed optimizations into a new era of realistic, multi-objective research. As Reviewer `gvym` aptly noted, our work presents a powerful paradigm: "to study a hard operations problem, first construct a realistic simulator... and then play with it." Our benchmark provides the robust, standardized sandbox that the field has been missing, enabling fair comparison and accelerating the development of intelligent schedulers that can meaningfully reduce AI's environmental footprint.

We were very encouraged by the positive outcome of the discussion phase, where our detailed rebuttal successfully addressed the initial concerns of all reviewers. We demonstrated that the simulator's realism is grounded in component-wise validation against empirical data and established models (e.g., EnergyPlus). Furthermore, the project's accessibility ensures it will have immediate and broad impact: it is fully open-source, **praised by reviewers for being "well documented" and “very well organized” (including modern standards like Croissant metadata for discoverability)**, and has proven compatibility with major SOTA RL libraries and frameworks like **Ray RLlib** for training a suite of agents (SAC, PPO, etc.).

`SustainCluster` provides a foundational resource that will catalyze new research avenues for years to come. We believe its scope, technical depth, and potential to drive progress on a critical global challenge make it an ideal candidate for acceptance with consideration for a spotlight selection.

Thank you for your time and consideration.

---

### Decision · Program_Chairs · 2025-09-18

**Decision:**

Accept (poster)

**Comment:**

Strengths & Contributions
- Novel dataset & simulator: SustainCluster combines AI workload traces, electricity prices, carbon intensity, weather, and transmission data to model spatially distributed datacenter scheduling.
- Multi-objective focus: Supports evaluation across carbon emissions, energy/water use, SLA satisfaction, costs, and performance, reflecting realistic sustainability concerns.
- Physics- & hardware-informed grounding: Models datacenter usage with geographic and temporal load-shifting impacts tied to real-world conditions.
- Benchmark integration: Provides reinforcement learning environments and well-documented supplementary material (e.g., Croissant metadata).
- Baseline evaluations: Includes a variety of controllers; results show that single-objective approaches are suboptimal, highlighting the challenge and relevance of the benchmark.

Clarity & presentation: Paper is well-written, easy to follow, and presents an attractive paradigm: build realistic simulators, then benchmark algorithms on them.

Weaknesses & Limitations
- Limited comparison with existing datasets
- Does not clearly differentiate from prior datacenter workload datasets [1,2,3].
- Simplified modeling assumptions
- Relies on Alibaba 2020 traces, which may not capture modern scaling (e.g., LLM workloads) or improvements in compute hardware efficiency.

Lack of realism evaluation
- While realistic data sources are integrated, there is no systematic validation of how closely the simulator reflects actual datacenter operations (“all simulations are wrong, but how wrong?”).

Reviewer Consensus
- Overall impression: Strong, creative contribution with high potential impact as a benchmark for multi-objective datacenter sustainability optimization.
- Main concerns: Need for clearer differentiation from prior datasets, more realistic modeling of datacenter heterogeneity, and up-to-date workload traces.

===== FINAL UPDATE FROM DB Track PCs ====

The final decision for this paper has been taken by the program chairs after consultation with the SACs. All Senior Area Chairs have ranked papers according to the feedback from the AC during the review process. We decided to leave the original meta-review to reflect the opinion of the AC in light of the initial discussions with reviewers and SAC.